# Hardness in Markov Decision Processes:
# Theory and Practice

**Michelangelo Conserva**
Queen Mary University of London
London, United Kingdom
m.conserva@qmul.ac.uk

**Paulo Rauber**
Queen Mary University of London
London, United Kingdom
p.rauber@qmul.ac.uk

## Abstract

Meticulously analysing the empirical strengths and weaknesses of reinforcement learning methods in hard (challenging) environments is essential to inspire innovations and assess progress in the field. In tabular reinforcement learning, there is no well-established standard selection of environments to conduct such analysis, which is partially due to the lack of a widespread understanding of the rich theory of hardness of environments. The goal of this paper is to unlock the practical usefulness of this theory through four main contributions. First, we present a systematic survey of the theory of hardness, which also identifies promising research directions. Second, we introduce Colosseum, a pioneering package that enables empirical hardness analysis and implements a principled benchmark composed of environments that are diverse with respect to different measures of hardness. Third, we present an empirical analysis that provides new insights into computable measures. Finally, we benchmark five tabular agents in our newly proposed benchmark. While advancing the theoretical understanding of hardness in non-tabular reinforcement learning remains essential, our contributions in the tabular setting are intended as solid steps towards a principled non-tabular benchmark. Accordingly, we benchmark four agents in non-tabular versions of Colosseum environments, obtaining results that demonstrate the generality of tabular hardness measures.

## 1 Introduction

Reinforcement learning studies a setting where an agent interacts with an environment by observing states, receiving rewards, and selecting actions with the objective of optimizing a reward-based criterion. The field has attracted significant interest in recent years after striking performances obtained in board games [1] and video games [2, 3]. Solving these grand challenges constitutes a pivotal milestone in the field. However, the corresponding agents require efficient simulators due to their high sample complexity, i.e., the number of observations that they require to optimize a reward-based criterion in an unknown environment. Outside of games, many important applications in healthcare, robotics, logistics, finance, and advertising can also be naturally formulated as reinforcement learning problems. However, simulators for these scenarios may not be available, reliable, or efficient.

The development of reinforcement learning methods that explore efficiently has long been considered one of the most crucial efforts to reduce sample complexity. Meticulously evaluating the strengths and weaknesses of such methods is essential to assess progress and inspire new developments in the field. Such empirical evaluations are performed through benchmarks composed of a selection of environments and evaluation criteria. Ideally, this selection should be based on theoretically principled reasoning that considers the *hardness* of the environments and the *soundness* of the evaluation criteria.

In non-tabular reinforcement learning, where the number of states is large (and potentially infinite), there is no theory of hardness except for a few restricted settings. Consequently, the selection of

environments in current benchmarks [4, 5] relies solely on the experience of their authors. Although such benchmarks are certainly valuable, there is no guarantee that they contain a sufficiently diverse range of environments and that they are effectively able to quantify agent capabilities. In contrast, in tabular reinforcement learning, where the number of states and actions is finite, a rich theory of hardness of environments is available. Perhaps surprisingly, a principled benchmark based on this theory has so far been absent. There are at least two reasons for this absence. First, these hardness measures have been developed to provide theoretical guarantees for reinforcement learning methods, and not considered as directly useful for practical purposes. Second, the lack of a unifying presentation and critical comparison between different measures has limited their potential impact.

The goal of this paper is to establish the importance of hardness measures outside the context of theoretical reinforcement learning and unlock their practical usefulness with four main contributions. In Section 2, we present a systematic survey of the theory of hardness for Markov decision processes. This survey serves as an introduction, tutorial, and, equally importantly, identifies gaps in the current theoretical landscape that suggest promising directions for future work. In Section 3.1, we introduce `Colosseum`, a pioneering Python package that enables the empirical investigation of hardness and implements a principled benchmark for the four most widely studied tabular reinforcement learning settings. The selected environments aim to maximize diversity with respect to two important measures of hardness, thus providing a varied set of challenges for which a *precise* characterization of hardness is available. In Section 3.2, we present an empirical comparison between three theoretical, yet efficiently computable, measures of hardness. Our analysis provides insights into which aspects of hardness are best captured by each of the measures and identifies desirable qualities for future measures. Finally, in Section 3.3, we report the results of five agents with theoretical guarantees in our novel (principled) benchmark, which allows us to empirically validate the quality of the selection methodology by demonstrating that harder environments effectively lead to worse performances.

Although this paper is concerned with the *tabular* setting, for which principled measures of hardness are available, we intend our contributions as milestones towards the future development of theoretical and empirical measures of hardness for non-tabular reinforcement learning. Accordingly, Section 3.1 shows how tabular hardness measures can be used in a widespread class of environments that includes non-tabular versions of the environments in the `Colosseum` benchmark, which enables studying how these measures relate to the performance of four agents from the non-tabular `bsuite` benchmark [4].

## 2 Hardness in Theory

Section 2.1 introduces our notation and important definitions. Section 2.2 presents our survey of the theoretical landscape of measures of hardness, which includes a novel categorization of existing measures. Section 2.2.3 highlights the weaknesses of existing measures and introduces the concept of a *complete* measure of hardness, which we believe should be the focus of future developments.

### 2.1 Preliminaries

Let $\Delta(\mathcal{X})$ denote the set of probability distributions over a set $\mathcal{X}$. A finite Markov decision process (MDP) is a tuple $M = (\mathcal{S}, \mathcal{A}, P, P_0, R)$, where $\mathcal{S}$ is the finite set of states, $\mathcal{A}$ is the finite set of actions, $P : \mathcal{S} \times \mathcal{A} \to \Delta(\mathcal{S})$ is the transition kernel, $P_0 \in \Delta(\mathcal{S})$ is the initial state distribution, and $R : \mathcal{S} \times \mathcal{A} \to \Delta([0, 1])$ is the reward kernel [6]. Given an optimization horizon $T$, a reinforcement learning agent aims to find a (possibly stochastic) policy $\pi : \mathcal{S} \to \Delta(\mathcal{A})$ that optimizes a reward-based criterion. The MDP M is unknown to the agent but can be learned through experience. The interaction between the agent and the environment starts when the initial state $s_0 \sim P_0$ is drawn. For any $0 \leq t < T$, the agent samples an action $a_t \sim \pi_t(s_t)$ from its current policy $\pi_t$, and the environment draws the next state $s_{t+1} \sim P(s_t, a_t)$ and reward $r_{t+1} \sim R(s_t, a_t)$. An MDP is *episodic* with time horizon $H$ when the state $s_{t+1}$ is drawn from the initial state distribution whenever $t + 1 \equiv 0 \pmod{H}$ and *continuous* otherwise. In the continuous setting, a factor $\gamma \in (0, 1)$ is used to discount future rewards in the *discounted* setting, and future rewards are averaged across time in the *undiscounted* setting. For a policy $\pi$, an MDP M induces a Markov chain [7] with transition probabilities $P_{s \to s'}^{\pi}(M) = \sum_{a \in \mathcal{A}} \pi(a \mid s) P(s' \mid s, a)$ whose stationary distribution is denoted by $\mu^{\pi}$. MDPs can be classified into three communication classes [8]. An MDP is *ergodic* if the Markov chain induced by any deterministic policy is ergodic. An MDP is *communicating* if, for every two states $s, s' \in \mathcal{S}$, there is a deterministic policy that induces a Markov chain where $s$ is accessible

from $s'$ and vice versa. An MDP is *weakly communicating* when there is a partition $(\mathcal{C}, \mathcal{S} \setminus \mathcal{C})$ of the set of states $\mathcal{S}$ such that, for every two states $s, s' \in \mathcal{C}$, there is a deterministic policy that induces a Markov chain where $s$ is accessible from $s'$ and vice versa, and every state $s \in \mathcal{S} \setminus \mathcal{C}$ is transient in every Markov chain induced by any deterministic policy. Every ergodic MDP is communicating, and every communicating MDP is weakly communicating.

The episodic state-action value function is given by $Q_{h,\texttt{epi}}^{\pi}(s,a) := \mathbb{E}\left[\sum_{t=h+1}^{H} r_t | s_h = s, a_h = a\right]$, and the episodic state value function is given by $V_{h,\texttt{epi}}^{\pi}(s) := \sum_a \pi(a \mid s) Q_{h,\texttt{epi}}^{\pi}(s,a)$. The discounted state-action value function is given by $Q_{\gamma}^{\pi}(s,a) := \mathbb{E}\left[\sum_{t=1}^{\infty} \gamma^{t-1} r_t \mid s_0 = s, a_0 = a\right]$, and the discounted state value function is given by $V_{\gamma}^{\pi}(s) := \sum_a \pi(a \mid s) Q_{\gamma}^{\pi}(s,a)$. These expectations are taken w.r.t. the policy $\pi$, the transition kernel $P$, and the reward kernel $R$. In the undiscounted setting, the value of every state(-action) is the same, since rewards are averaged across infinite time steps. In that case, we define the expected average reward as $\rho^{\pi} := \lim_{T \to \infty} \frac{1}{T} \sum_{t=1}^{T} \mathbb{E}[r_t]$, which is also given by $\rho^{\pi} = \langle \mu^{\pi}, \mathrm{R}^{\pi} \rangle$, where $\mathrm{R}^{\pi}$ is a vector such that $\mathrm{R}_s^{\pi} = \mathbb{E}_{a \sim \pi(s)} R(s,a)$ is the expected reward obtained when following $\pi$ from state $s$. An optimal policy $\pi^*$ obtains maximum value for every state. We assume with little loss of generality that the optimal policy is unique. We drop the subscripts when the setting is clear from the context, and write $*$ to denote $\pi^*$ in superscripts.

The most widely studied performance criteria are the *expected cumulative regret*, which yields the *regret minimization* setting [9], and the *sample efficiency of exploration*, which yields the Probably Approximately Correct Reinforcement Learning (PAC-RL) setting [10]. In simple terms, the regret measures the loss in reward due to the execution of a sub-optimal policy. In contrast, the sample efficiency measures how many interactions the agent requires to approximately learn $\pi^*$ with high probability. The main difference between the two criteria is that the rewards obtained during the interactions with the MDP are not considered in PAC-RL, whereas all rewards contribute to the cumulative regret. Therefore, PAC-RL agents can generally afford more aggressive exploration.

## 2.2 Characterization of hardness

We distinguish the hardness of MDPs into two kinds of complexity, the *visitation* complexity and the *estimation* complexity. The visitation complexity relates to the difficulty of visiting all the states, and the estimation complexity relates to the discrepancy between the optimal policy and the best policy an agent can derive from a given estimate of the transition and reward kernels. The two complexities are complementary in the sense that the former quantifies the hardness of gathering samples from the state space while the latter quantifies the hardness of producing a highly rewarding policy given the samples. In the literature, we identify two approaches that aim to capture the mentioned complexities. The first approach, which we call *Markov chain-based*, considers that an MDP is an extension of a Markov chain where transition probabilities can be changed based on direct interventions by an agent [11]. This approach is well suited to capture the hardness that comes from the visitation complexity since it considers the properties of transition kernels. The second approach, which we call *value-based*, considers the discrepancy between the optimal policy and the best policy an agent can derive from a value function point estimate. Note that the information contained in such point estimate is lower than the one in kernels estimate and that the difficulty of obtaining an accurate estimate of the value function is not considered in this approach. Therefore, the value-based approach is only able to partially capture the estimation complexity. A striking fact that highlights this shortcoming is that almost every value-based measure of hardness is independent of the variability of the reward kernel. Therefore, given an MDP M$'$ that is obtained by increasing the reward kernel variability of an MDP M, value-based measures of hardness assign the same level of hardness to M and M$'$.

Markov chain properties and value functions depend on a fixed policy, which presents two natural choices to derive measures of hardness. The first choice considers the optimal policy, which typically leads to a measure that considers a best-case scenario. The second choice considers a policy that maximizes a criterion that characterizes a worst-case scenario. For instance, a policy that maximizes such criterion may spend its time in a region of the state space that is not relevant for learning $\pi^*$.

### 2.2.1 Markov chain-based measures of hardness

**Mixing time.** The mixing time of a Markov chain with stationary distribution $\mu$ is defined as

$$t_\mu := \inf\{n \mid \sup_{s \in \mathcal{S}} d_{\mathrm{TV}}\left(p_s^n, \mu\right) \leq 0.25\}, \tag{1}$$

where $d_{\mathrm{TV}}$ is the total variation distance between distributions and the vector $p_s^n$ represents the distribution over states after $n$ steps starting from state $s$. The value $0.25$ is conventionally established in the Markov chain literature for the definition of the mixing time. For ergodic and aperiodic Markov chains, $\lim_{n \to \infty} p_s^n = \mu$ for every state $s$, so the mixing time is the number of steps a Markov chain takes to produce samples that are *close* to being distributed according to the stationary distribution $\mu$.

For instance, a Markov chain with a non-negligible probability of transitioning from every state to every state is quickly mixing. In contrast, the mixing time can be very long in chains where the state space has several distinct regions each of which is well connected but where transitions between regions have low probability [12]. Kearns and Singh [13] propose an extension of the mixing time to MDPs that considers the maximum mixing time across policies $t := \sup_\pi t_{\mu^\pi}$ in the undiscounted setting. Although the mixing time plays an important role in that setting, since the average reward obtained by policy $\pi$ is given by $\rho^\pi = \langle \mu^\pi, \mathbf{R}^\pi \rangle$, it is not a generally good measure of hardness. First, it does not capture the visitation complexity, since it neglects the fact that the agent may direct its exploration through a choice of policy. Second, it does not capture any significant aspect of optimal policy estimation, which does not require stationary distribution samples from every policy.

**Diameter.** The diameter is fundamentally related to the number of time steps required to transition between states. In the continuous setting, the diameter $D$ is most commonly defined as

$$D := \sup_{s_1 \neq s_2} \inf_\pi T_{s_1 \to s_2}^\pi, \tag{2}$$

where $T_{s_1 \to s_2}^\pi$ is the expected number of time steps required to reach state $s_2$ from state $s_1$ when following policy $\pi$ [14]. Intuitively, $D$ is the worst-case expected number of time steps required to transition between two states when following the best policy for that purpose. Related definitions are $D_{\mathrm{worst}} := \sup_\pi \sup_{s_1 \neq s_2} T_{s_1 \to s_2}^\pi$ and $D_{\mathrm{opt}} := \inf_\pi \sup_{s_1 \neq s_2} T_{s_1 \to s_2}^\pi$, which imply $D \leq D_{\mathrm{opt}} \leq D_{\mathrm{worst}}$ [15].

In the episodic setting, we define the diameter by augmenting each state $s$ with the current in-episode time step $h$ and considering the diameter of this augmented MDP in the continuous setting. Note that $T_{(s_1,h_1) \to (s_2,h_2)}^\pi$ can be larger than the episode length $H$, which means that (on average) more than one episode may be required to transition from state $(s_1, h_1)$ to state $(s_2, h_2)$. This happens whenever $h_2 < h_1$, which may be undesirable if the intent is to focus on the expected number of time steps required to transition between states from the same episode. The diameter is always infinite in weakly-communicating MDPs if the supremum is not restricted to states in the recurrent class $\mathcal{C}$ and is always finite in the episodic setting (where every state is reachable within an episode). A large diameter can be caused by high stochasticity. The diameter is very apt at measuring visitation complexity, since it captures the effort required to deliberately move between states. However, it neglects the reward kernel, and so has limited capacity to measure the estimation complexity.

**Distribution mismatch coefficient.** The *distribution mismatch coefficient* (DMC) has been defined for the continuous undiscounted [16] and continuous discounted [17] cases respectively as

$$\mathrm{DMC} := \sup_\pi \sum_{s \in \mathcal{S}} \frac{\mu_s^*}{\mu_s^\pi} \qquad \text{and} \qquad \mathrm{DMC}_{s_0} := \sup_{s \in \mathcal{S}} \frac{d_{s_0}^*(s)}{P_0(s)},$$

where $d_{s_0}^*(s) = (1 - \gamma) \sum_{t=0}^\infty \gamma^t \mathrm{Pr}(s_t = s \mid s_0)$ is the discounted state visitation distribution of the optimal policy given an initial state $s_0$. Note that the DMC is guaranteed to be finite only for ergodic MDPs. In communicating MDPs, there is at least one policy $\pi$ whose stationary distribution assigns probability zero to some states. MDPs whose optimal stationary distribution $\mu^*$ has its probability mass concentrated on a few states tend to have a large DMC. In contrast, when $\mu^*$ is closer to being uniformly distributed across states, the DMC tends to be small. For small values of DMC, as every $\mu^\pi$ is *close* to $\mu^*$, the agent will gather samples from the optimal stationary distribution regardless of its current policy, which may enable quick learning. In contrast, for large values of DMC, the agent needs to actively seek a policy that gathers such samples. The DMC is not well suited to quantify

the visitation complexity, since it fails to capture the difficulty of visiting all states. The DMC also does not capture the estimation complexity, since it does not account for the stochasticity of the environment, which is related to the number of samples required to make accurate estimations.

### 2.2.2 Value-based measures of hardness

**Action-gap regularity.** Given an estimate $\hat{Q}^*$ of the optimal state-action value function $Q^*$, the *greedy policy* with respect to $\hat{Q}^*$ always chooses an action associated with the highest state-action value. Whether or not such a policy is optimal depends exclusively on the ordering of the estimates for a given state rather than their accuracy. For instance, assuming that $a^*$ is the optimal action for every state $s$, a greedy agent would act optimally if $\hat{Q}(s, a^*) > \hat{Q}(s, a')$ for every action $a' \neq a^*$, even if $|Q^*(s, a) - \hat{Q}(s, a)| \gg 0$ for every action $a$. The action-gap regularity $\zeta$ is a measure of hardness that leverages this principle through the theory of hardness for classification algorithms [18]. However, this measure is only defined for two actions, and so has exceptionally limited applicability.

**Environmental value norm.** The (discounted) environmental value norm $C_\gamma^\pi$ is defined as

$$C_\gamma^\pi := \sup_{(s,a)} \sqrt{\operatorname*{Var}_{s' \sim P(s,a)} V_\gamma^\pi(s')}.$$

This quantity can be similarly defined in the undiscounted setting [19]. In words, the environmental value norm captures the one-step variance of the value function $V^\pi$ for a given policy $\pi$. In the episodic setting, a closely related measure called *maximum per-step conditional variance $C_H^\pi$* [20] is defined as

$$C_H^\pi := \sup_{(s,a,h)} \left( \operatorname{Var} R(s,a) + \operatorname*{Var}_{s' \sim P(s,a)} V_{h+1}^\pi(s') \right).$$

Alternatively, as with the diameter, it is also possible to define this quantity by augmenting each state $s$ with the current in-episode time step $h$ and considering the environmental value norm of this augmented MDP in the continuous setting. For every policy, the environmental value norm is equal to zero when the transition kernel is deterministic. However, a highly stochastic MDP may still have a small environmental value norm, since this norm captures the variance of the state value function rather than the stochasticity of the transition kernel. Maillard et al. [19] suggest using the environmental value norm of the optimal policy $\pi^*$ as a measure of hardness. The main strength of such measure is that the variability of the optimal value function captures an important aspect of the estimation complexity. However, this measure of hardness neglects the visitation complexity.

**Sub-optimality gap.** The sub-optimality gap is defined in the continuous and episodic settings as

$$\Delta(s,a) := V^*(s) - Q^*(s,a) \quad \text{and} \quad \Delta_h(s,a) := V_h^*(s) - Q_h^*(s,a),$$

respectively. Since $V^*(s) = \max_a Q^*(s,a)$ for every state $s$, the sub-optimality gap $\Delta(s,a)$ measures the difference in expected return between selecting the optimal action for state $s$ and selecting the action $a$. Intuitively, identifying a suboptimal action $a'$ in a given state $s$ is easier if the gap $\Delta(s, a')$ is large. Simchowitz and Jamieson [21] identifies the sum of the reciprocals of the sub-optimality gaps $\sum_{(s,a)|\Delta(s,a) \neq 0} \frac{1}{\Delta(s,a)}$ as a measure of hardness. They also demonstrate the importance of the sub-optimality gaps by showing that recent optimistic algorithms necessarily incur in a cumulative regret proportional to the smallest nonzero sub-optimality gap in the episodic setting. However, note that approximating the optimal value function (and identifying a near-optimal policy) is particularly easy when every sub-optimality gap is small. Consequently, the (PAC-RL) sample complexity is likely to decrease when the sum of the reciprocals of the sub-optimality gaps increases. Furthermore, this measure does not explicitly capture visitation complexity and is prone to severe numerical issues.

Table 1: Computational complexity of generally applicable measures up to logarithmic factors.

| | Markov chain-based measures | | Value-based measures | |
|---|---|---|---|---|
| Mixing time | Diameter | Distribution mismatch coefficient | Environmental value norm | Sub-optimality gaps |
| ? | $\tilde{O}(|\mathcal{S}|^{3.5}|\mathcal{A}|)$ | ? | $\tilde{O}(|\mathcal{S}|^2|\mathcal{A}|(1-\gamma)^{-1})$ | $\tilde{O}(|\mathcal{S}|^2|\mathcal{A}|(1-\gamma)^{-1})$ |

### 2.2.3 Future directions

Current hardness measures suffer from three principal issues. They are not designed to be efficiently computable (see Table 1 and Appendix B), they are limited in their ability to simultaneously capture visitation complexity and estimation complexity, and they are oblivious to the distinct challenges presented by different performance criteria. For instance, while regret minimizing agents must be cautious not to incur in large regret during learning (for example, by not revisiting lowly rewarding states that are not followed by highly rewarding states), PAC-RL agents have the flexibility to incur in large regret as long as they end up with a near-optimal policy. Therefore, the visitation complexity should differ across settings. The fact that current measures disregard this distinction is concerning, since they should account for the specific difficulty of the optimization task. These issues are not discussed in previous work, since measures of hardness have not been considered relevant outside the context of deriving theoretical performance guarantees for reinforcement learning agents.

In order to address these issues, we believe that future work should focus on developing efficiently computable (non-trivial) hardness measures that (approximately) meet the following novel definition.

**Definition 2.1 (Complete measure of hardness)** A measure $\theta : \mathcal{M} \to \mathbb{R}^+$ is *complete* for an MDP class $\mathcal{M}$ and criterion $\psi$ (sample complexity or cumulative regret) if, for every pair $M_1, M_2 \in \mathcal{M}$ and *near-optimal agent* $A^*$ that achieves the criterion lower bound of class $\mathcal{M}$ up to logarithmic factors,[1] $\theta(M_1) > \theta(M_2)$ implies $\tilde{\psi}(M_1, A^*) > \tilde{\psi}(M_2, A^*)$, where $\tilde{\psi}$ hides logarithmic factors.

Combining existing measures that capture visitation complexity and estimation complexity is a viable first step in that direction. Recently, Wagenmaker et al. [23] have pioneered this approach in the episodic setting by proposing the *gap-visitation complexity*,

$$\text{GVP}(\epsilon) := \sum_{h=0}^{H} \inf_{\pi} \sup_{s,a} \inf \left( \frac{1}{w_h^\pi(s,a)\Delta_h(s,a)^2}, \frac{W_h(s)^2}{w_h^\pi(s,a)\epsilon^2} \right), \tag{3}$$

where $w_h^\pi(s,a) := P_\pi(s_h = s, a_h = a)$ is the probability of visiting state-action pairs $(s,a)$ at in-episode time step $h$ when following policy $\pi$, $W_h(s) := \sup_\pi P_\pi(s_h = s)$ is the maximum reachability of state $s$ at in-episode time step $h$, and $\epsilon$ is a parameter related to the optimality of the output policy in the PAC-RL setting. The strength of this measure is that it weights the sub-optimality action gaps with measures of visitation complexity, $w_h^\pi$ and $W_h$. This captures the difficulty induced by the critical states that are both hard to reach and for which it is hard to estimate the best action. However, the gap-visitation complexity fails to be a generally applicable hardness measure. It depends on the PAC-RL setting-specific parameter $\epsilon$, it is restricted to the finite horizon setting (and can not be extended to the continuous setting), and is not efficiently computable.

## 3 Hardness in Practice

### 3.1 Colosseum

This section briefly introduces `Colosseum`, a pioneering Python package that bridges theory and practice in tabular reinforcement learning while also being applicable in the non-tabular setting. More details about the package can be found in Appendix A and in the project website.[2]

As a hardness analysis tool, `Colosseum` identifies the communication class of MDPs, assembles insightful visualizations and logs of interactions between agents and MDPs, computes three measures of hardness (environmental value norm, sum of the reciprocals of the sub-optimality gaps, and diameter, whose computation requires a novel solution described in App. A.4). Eight MDP families are available for experimentation. Some are traditional families (`RiverSwim` [24], `Taxi` [25], and `FrozenLake`) while others are more recent (`MiniGid` environments [26]). Additionally, `DeepSea` [27] was included as a hard exploration family of problems, and the `SimpleGrid` family is composed of simplified versions of the `MG-Empty` environment. By controlling the parameters of MDPs from each family (further detailed in App. A.3), it is easy to create an MDP with any desired hardness.

As a benchmarking tool, `Colosseum` is unique in its strong connection with theory. For instance, in contrast to non-tabular benchmarks, `Colosseum` computes theoretical evaluation criteria such

---

[1]For example, the $\Omega(|\mathcal{S}||\mathcal{A}|H^2\epsilon^{-2})$ sample complexity bound for the episodic communicating setting [22].

[2]Available at `https://michelangeloconserva.github.io/Colosseum`.

as the expected cumulative regret and the expected average future reward, which can be used to exactly evaluate the performance criterion of regret minimizing agents. The benchmark covers the most commonly studied reinforcement learning settings: *episodic ergodic*, *episodic communicating*, *continuous ergodic*, and *continuous communicating*. For each setting, we have selected twenty MDPs that are diverse with respect to their diameters and environmental value norms as proxies for different combinations of visitation complexity and estimation complexity. Figure 17 in Appendix E shows how each of these MDPs varies according to these measures, and Section 3.3 empirically validates this selection by showing that harder MDPs correspond to worse agent performance. Notably, the theoretically backed selection of MDPs and the rigorous evaluation criteria make the `Colosseum` benchmark the most exhaustive in tabular reinforcement learning, since previous evaluations were conducted empirically in a few MDPs (such as Taxi or RiverSwim).

`Colosseum` also allows testing of non-tabular agents by leveraging the BlockMDP model [28]. BlockMDPs equip tabular MDPs with an *emission map* that is a (possibly stochastic) mapping $q : \mathcal{S} \to \Delta(\mathcal{O})$ from the finite state space $\mathcal{S}$ to a (possibly infinite) *observation space* $\mathcal{O}$. Agents interacting with BlockMDPs are only provided with observations, so non-tabular methods are generally required. Many commonly used non-tabular MDPs (such as Minecraft [29]) can be straightforwardly encoded as BlockMDPs using the `Colosseum` MDP families. `Colosseum` implements a diverse set of deterministic emission maps and allows combining them with different sources of noise. Appendix A.2 further details BlockMDPs and the available emission maps.

## 3.2 Empirical analysis of hardness measures

For brevity, this section only presents results of hardness measures in the `MiniGridEmpty` family of environments in the episodic setting. Appendix D presents the full outcome of the empirical analysis.

A `MiniGridEmpty` MDP is a grid world where an agent has three available actions: moving forward, rotating left, and rotating right. An agent is rewarded for being in a few specific states and receives no reward in every other state. Appendix A.3.4 provides more details about this family of environments.

In our investigation, we consider four scenarios that highlight the different aspects of MDP hardness.

**Scenario 1.** We vary the probability `p_rand` that an MDP executes a random action instead of the action selected by an agent. As `p_rand` approaches one, value estimation becomes easier, since outcomes depend less on agent choices. However, intentionally visiting states becomes harder.

**Scenario 2.** We vary the probability `p_lazy` that an MDP stays in the same state instead of executing the action selected by an agent. Contrary to increasing `p_rand`, increasing `p_lazy` never benefits exploration. Increasing `p_lazy` decreases estimation complexity and increases visitation complexity.

**Scenario 3 and 4.** We vary the number of states across MDPs from the same family. In scenario 4, we also let `p_rand` = 0.1 to study the impact of stochasticity. In these scenarios, increasing the number of states simultaneously increases the estimation complexity and the visitation complexity.

In every scenario, hardness measures are compared with the cumulative regret of a *near-optimal* agent tuned for each specific MDP (see App. D). This regret serves as an optimistic measure of hardness. Appendix C describes how these measures are normalized. Note that, due to normalization, the plots should only be compared in terms of trends (growth rates) rather than absolute values.

**Analysis.** Figure 1 presents the empirical results for the episodic `MiniGridEmpty` family in the four scenarios with 95% bootstrapped confidence intervals over twelve random seeds.

The experiments confirm our claim that the diameter captures visitation rather than estimation complexity. This measure of hardness grows superlinearly with both `p_rand` and `p_lazy` (Figures 1a and 1b) since deliberate movement between states requires an exponentially increasing number of time steps. Although the diameter highlights the sharply increasing visitation complexity, its trend overestimates the increase in cumulative regret of the tuned near-optimal agent, which is explained by the unaccounted decrease in estimation complexity. The diameter also increases almost linearly with the number of states (Figures 1c and 1d). For the small `p_rand` (scenario 4), the relation is still approximately linear. This linear trend underestimates the evident non-linear growth in hardness in the regret of the tuned near-optimal agent but is in line with the mild increase in visitation complexity.

The empirical evidence indicates that the environmental value norm can only capture estimation complexity. It decreases as `p_lazy` and `p_rand` increase (Figures 1a and 1b) because the optimal

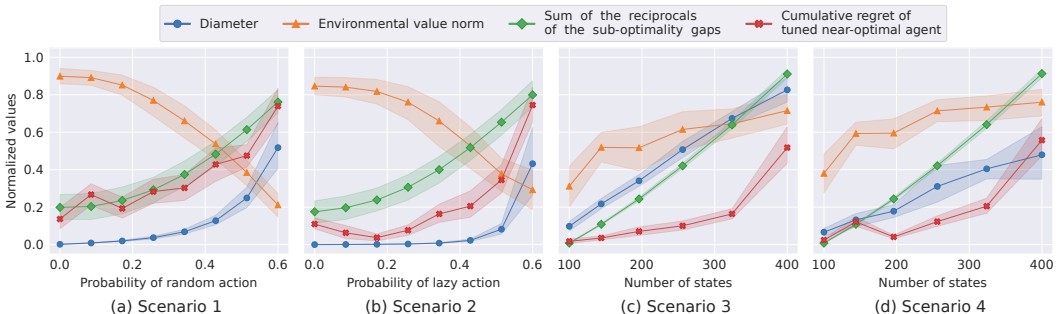

Figure 1: The `Colosseum` hardness analysis for the episodic `MiniGridEmpty` family.

value of neighboring states becomes closer, which decreases the per-step variability of the optimal value function. When the number of states increases but the transition and reward structures remain the same (Figures 1c and 1d), the small increase in this variability only generates a sublinear growth.

We empirically observe that the sum of the reciprocals of the sub-optimality gaps is not particularly apt at capturing estimation complexity, due to its exclusive focus on optimal policy identification, and it also underestimates the increase in hardness induced by an increase in visitation complexity. This measure increases weakly superlinearly in scenarios 1 and 2 (Figures 1a and 1b). The probability of executing the action selected by the agent decreases when `p_lazy` and `p_rand` increase, so the difference between the state and the state-action optimal value functions decreases sharply. The measure increases almost linearly with the number of states (Figures 1c and 1d). This is explained by the fact that the average value of the additional terms in the summation is often similar to the average value of the existing terms when MDPs have the same structure of reward and transition kernels.

### 3.3 `Colosseum` benchmarking

In this section, we benchmark five tabular agents with theoretical guarantees and four non-tabular agents. Besides being valuable on their own, these results help to empirically validate our benchmark.

**Agents.** The tabular agents are posterior sampling for reinforcement learning (PSRL) for the episodic and continuous settings [30, 31], Q-learning with UCB exploration for the episodic setting [32], Q-learning with optimism for the continuous setting [16], and UCRL2 for the continuous setting [14]. The non-tabular agents (from `bsuite`) are ActorCritic, ActorCriticRNN, BootDQN, and DQN.

**Experimental procedure.** We set the total number of time steps to 500 000 with a maximum training time of 10 minutes for the tabular setting and 40 minutes for the non-tabular setting. If an agent does not reach the maximum number of time steps before this time limit, learning is interrupted, and the agent continues interacting using its last best policy. This guarantees a fair comparison between agents with different computational costs. The performance indicators are computed every 100 time steps. Each interaction between an agent and an MDP is repeated for 20 seeds. The agents' hyperparameters have been chosen by random search to minimize the average regret across MDPs with randomly sampled parameters (see Appendix E). We use a deterministic emission map that assigns a uniquely identifying vector to each state (for example, a gridworld coordinate) to derive the non-tabular benchmark MDPs. In Table 2, we report the per-step normalized cumulative regrets divided by the total number of time steps (defined in Appendix C), which allows comparisons across different MDPs. We summarize the main findings here and refer to Appendix E for further details.

**Analysis.** Table 2 often shows high variability in the performance of the same agent across MDPs of the same family. Therefore, maximising the diversity across diameters and value norms effectively produces diverse challenges even for MDPs with similar transition and reward structures. For example, in the continuous communicating case (Table 2d), Q-learning performs well only in some MDPs of the `MiniGridEmpty` family. This also happens for UCRL2 for the `SimpleGrid` family.

The average normalized cumulative regret is lower in ergodic environments compared to communicating environments. This indicates that the ergodic setting is generally slightly easier than the communicating settings. Notably, in the continuous setting, the ergodic setting is more challenging than the communicating setting for Q-learning (Tables 2c and 2d). Designing a naturally ergodic

Table 2: Normalized cumulative regrets of selected agents on the `Colosseum` benchmark. (a) Episodic ergodic. (b) Episodic communicating. (c) Continuous ergodic. (d) Continuous communicating.

(a)

| MDP | Q-learning | PSRL |
|---|---|---|
| DeepSea | .64 ± .00 | **.01** ± .00 |
|  | .52 ± .01 | **.00** ± .00 |
| FrozenLake | .90 ± .01 | **.01** ± .00 |
| MG-Empty | 1.00 ± .00 | **.86** ± .16 |
|  | 1.00 ± .00 | **.94** ± .07 |
|  | 1.00 ± .00 | **.91** ± .09 |
|  | 1.00 ± .00 | **.35** ± .10 |
|  | 1.00 ± .00 | **.44** ± .12 |
|  | .92 ± .04 | **.14** ± .08 |
|  | .91 ± .03 | **.04** ± .03 |
| MG-Rooms | .90 ± .04 | **.05** ± .04 |
|  | 1.00 ± .00 | **.54** ± .36 |
|  | .99 ± .01 | **.24** ± .29 |
| RiverSwim | .07 ± .02 | **.00** ± .00 |
|  | .91 ± .01 | **.00** ± .00 |
| SimpleGrid | .78 ± .03 | **.05** ± .01 |
|  | .79 ± .03 | .79 ± .03 |
|  | .50 ± .03 | .50 ± .03 |
| Taxi | .84 ± .01 | **.08** ± .01 |
|  | .56 ± .02 | **.05** ± .00 |
| *Average* | .81 ± .24 | **.30** ± .33 |

(b)

| MDP | Q-learning | PSRL |
|---|---|---|
| DeepSea | .01 ± .01 | **.00** ± .00 |
|  | .83 ± .02 | **.54** ± .11 |
| FrozenLake | .78 ± .04 | **.03** ± .11 |
| MG-Empty | .59 ± .07 | **.09** ± .05 |
|  | .99 ± .00 | **.24** ± .15 |
|  | .99 ± .01 | **.23** ± .12 |
|  | 1.00 ± .00 | **.91** ± .09 |
|  | 1.00 ± .00 | **.93** ± .09 |
| MG-Rooms | .99 ± .01 | **.21** ± .29 |
|  | 1.00 ± .00 | **.44** ± .39 |
|  | 1.00 ± .00 | **.43** ± .39 |
|  | .94 ± .05 | **.04** ± .04 |
| RiverSwim | .87 ± .00 | **.00** ± .00 |
|  | .96 ± .01 | **.80** ± .00 |
| SimpleGrid | .78 ± .10 | **.20** ± .15 |
|  | .80 ± .00 | **.55** ± .15 |
|  | .50 ± .00 | **.11** ± .01 |
|  | **.79** ± .04 | **.79** ± .04 |
| Taxi | .94 ± .00 | **.09** ± .01 |
|  | .91 ± .01 | **.36** ± .06 |
| *Average* | .83 ± .23 | **.35** ± .30 |

(c)

| MDP | Q-learning | PSRL | UCRL2 |
|---|---|---|---|
| DeepSea | .94 ± .00 | **.06** ± .01 | .23 ± .05 |
| FrozenLake | .83 ± .03 | **.01** ± .03 | **.01** ± .02 |
| MG-Empty | .98 ± .02 | .99 ± .01 | **.05** ± .06 |
|  | .98 ± .02 | .98 ± .04 | **.03** ± .05 |
|  | .97 ± .00 | .95 ± .03 | **.04** ± .01 |
|  | .98 ± .01 | .99 ± .01 | **.54** ± .26 |
|  | .96 ± .01 | .83 ± .31 | **.01** ± .00 |
|  | .98 ± .02 | .99 ± .02 | **.45** ± .35 |
|  | .98 ± .03 | .99 ± .01 | **.27** ± .33 |
|  | .98 ± .01 | .99 ± .01 | **.93** ± .09 |
| MG-Rooms | .98 ± .03 | .99 ± .02 | **.18** ± .29 |
|  | .98 ± .02 | 1.00 ± .00 | **.62** ± .36 |
| RiverSwim | .73 ± .19 | **.00** ± .00 | **.00** ± .00 |
|  | .71 ± .22 | **.00** ± .00 | **.01** ± .00 |
|  | .90 ± .06 | .02 ± .04 | **.01** ± .01 |
|  | .50 ± .25 | **.01** ± .00 | **.01** ± .01 |
| SimpleGrid | .78 ± .00 | .70 ± .19 | **.01** ± .01 |
|  | .46 ± .08 | .01 ± .02 | **.00** ± .00 |
|  | .49 ± .01 | .43 ± .16 | **.00** ± .00 |
| Taxi | .87 ± .01 | .89 ± .08 | **.09** ± .01 |
| *Average* | .85 ± .18 | .59 ± .44 | **.17** ± .26 |

(d)

| MDP | Q-learning | PSRL | UCRL2 |
|---|---|---|---|
| DeepSea | **.78** ± .00 | **.78** ± .05 | .90 ± .01 |
|  | .99 ± .00 | .99 ± .00 | .99 ± .00 |
|  | **.79** ± .00 | **.79** ± .04 | .92 ± .01 |
| FrozenLake | .77 ± .04 | **.01** ± .04 | **.01** ± .01 |
|  | .84 ± .04 | **.01** ± .02 | .04 ± .06 |
| MG-Empty | .51 ± .23 | .95 ± .22 | **.02** ± .00 |
|  | .01 ± .00 | 1.00 ± .00 | **.02** ± .00 |
|  | **.00** ± .00 | .60 ± .50 | **.01** ± .00 |
|  | .35 ± .17 | 1.00 ± .00 | **.01** ± .00 |
|  | .75 ± .21 | 1.00 ± .00 | **.08** ± .20 |
| MG-Rooms | **.01** ± .01 | 1.00 ± .00 | .78 ± .40 |
|  | **.01** ± .01 | 1.00 ± .00 | .02 ± .01 |
|  | **.02** ± .02 | 1.00 ± .00 | .66 ± .47 |
| RiverSwim | .16 ± .03 | **.00** ± .01 | **.00** ± .00 |
|  | .34 ± .14 | **.01** ± .00 | .02 ± .01 |
| SimpleGrid | .11 ± .01 | .93 ± .00 | **.01** ± .00 |
|  | **.01** ± .00 | .45 ± .15 | **.01** ± .00 |
|  | .15 ± .01 | .93 ± .00 | .70 ± .40 |
|  | **.01** ± .00 | .50 ± .00 | .33 ± .24 |
| Taxi | .95 ± .00 | .94 ± .04 | **.12** ± .01 |
| *Average* | .38 ± .37 | .69 ± .38 | **.28** ± .37 |

MDP is not straightforward. In fact, the majority of MDPs in the literature are communicating. In `Colosseum`, ergodicity is induced by setting `p_rand` $> 0$ in otherwise communicating MDPs. Model-free agents struggle with the resulting increase in variability of the state-action value function.

In the episodic settings (Tables 2a and 2b), PSRL obtains excellent performances with low variability. Q-learning instead performs well in a few MDPs. This often happens since, when the action selected by the agent is randomly substituted (due to `p_rand` $> 0$) with one with a large sub-optimality gap, the resulting $Q$-value update introduces a critical error that requires many samples to be corrected.

In the continuous settings (Tables 2c and 2d), UCRL2 performs best in the ergodic cases when Q-learning suffers from the issue caused by `p_rand` $> 0$ but is only slightly better than Q-learning in the communicating ones. PSRL instead struggles with most MDPs. The reason for its weak performance in this setting is the computationally expensive optimistic sampling procedure required for its worst-case theoretical guarantees. It often breaks the time limit before reaching the first quarter of available time steps, meaning that it lacks sufficient samples to estimate the optimal policy.

Figure 2 places the regret of the agents in the continuous ergodic setting (Table 2c) on a position corresponding to the diameter and value norm of the benchmark environments. PSRL and Q-learning (Figures 2a and 2b), appear to be impacted more by the value norm than the diameter. This is in line with the lack of sufficient samples for PSRL and the aforementioned issue related to high q estimates variability for Q-learning, which is exacerbated when the estimation complexity is higher. In the case of UCRL2 (Figure 2c), which provides more reliable evidence since it performs well across the MDPs, higher regret effectively corresponds to higher diameter and value norm.

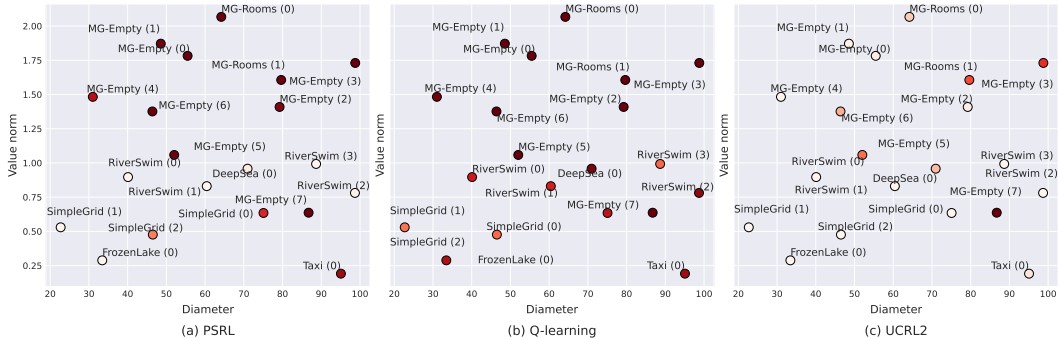

Figure 2: Average cumulative regret obtained by the tabular agents with guarantees in the continuous ergodic setting placed according to the diameter and the value norm associated to the MDPs.

**Non-tabular benchmarking.** The performance of the agents is in line with the results reported by Osband et al. [4], with the exception of BootDQN. Being the most computationally intensive, this agent often breaks the time limit, which consequently worsens its overall performance. Figure 3 places the regret of the agents in the continuous ergodic setting on a position corresponding to the diameter and value norm of the benchmark environments. Interestingly, and similarly to the tabular case (Figure 2), while the best performing agent (DQN) is evidently impacted more by the diameter, the opposite holds for the other agents. Regardless of the visitation complexity, this suggests that an agent that fails to handle the estimation complexity of an environment is bound to perform badly both in the tabular and non-tabular settings.

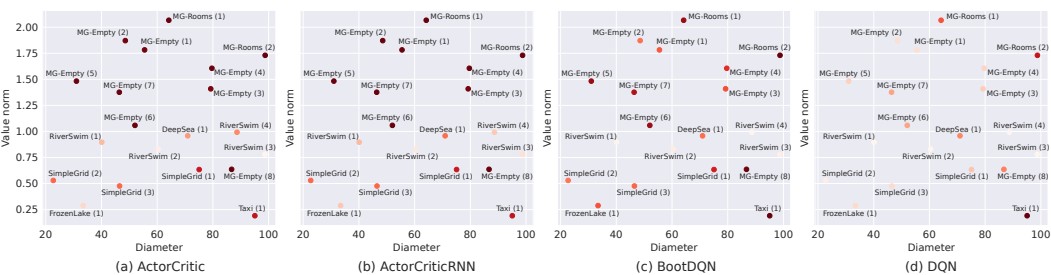

Figure 3: Average cumulative regret obtained by the non-tabular baseline agents in the continuous ergodic setting placed according to the diameter and the value norm associated to the MDPs.

# 4 Conclusion

We established the usefulness of the theory of hardness in empirical reinforcement learning. Prior to our work, hardness measures were limited to providing theoretical guarantees for agents. In order to promote a wider understanding of these measures, we presented a systematic survey that newly identified two major approaches for characterizing hardness: Markov chain-based and value-based. These approaches aim to capture complementary aspects of hardness: visitation complexity and estimation complexity. Our survey also exposed a relative lack of measures that capture both aspects, which motivates our definition of complete measures of hardness. Their development is important theoretically, elucidating what makes a problem hard for a specific performance criterion, and empirically, allowing the creation of principled benchmarks for a specific performance criterion.

We presented the first empirical study of (efficiently computable) hardness measures. This study revealed which aspects of hardness current measures capture and clarified their relationship with the behavior of near-optimal agents. Based on these results, we proposed a benchmark for the most widely studied tabular reinforcement learning settings that contains environments that maximize diversity with respect to two highly distinct measures. Such a principled benchmark is valuable to gauge progress in the field. The new benchmark allowed conducting the most exhaustive empirical comparison between theoretically principled tabular reinforcement learning agents to date, which revealed undocumented weaknesses of these agents and further validated our choices of environments.

As a first step towards principled non-tabular benchmarking, we argued that many commonly used environments can be encoded as BlockMDPs, which are non-tabular versions of tabular MDPs for which a partial characterization of hardness is already possible. We observed a clear empirical relation between two tabular hardness measures and the performance of four non-tabular agents. BlockMDPs represent a promising starting point for the future development of non-tabular hardness measures while already being useful to provide relevant insights into the performance of non-tabular agents.

Our work has led to the development of `Colosseum`, a pioneering tool for empirical but theoretically principled study of tabular reinforcement learning with experimental non-tabular benchmarking capabilities. Besides implementing the aforementioned tabular benchmark, `Colosseum` provides valuable analysis tools: regret and hardness computations, communication class identification, logging, and visualizations. `Colosseum` can also be easily extended and integrated with new agents and environments, for which we will actively seek contributions from the community. We strongly believe that `Colosseum` has the potential to become a fundamental tool in reinforcement learning.

## Acknowledgments and Disclosure of Funding

This research was financially supported by the Intelligent Games and Games Intelligence CDT (IGGI; EP/S022325/1) and used Queen Mary University of London Apocrita HPC facility. The authors would like to thank Sjoerd van Steenkiste and Tabish Rashid for their valuable feedback.

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

# A  Colosseum

Colosseum is a pioneering Python package that creates a bridge between theory and practice in tabular reinforcement learning with an eye on the non-tabular setting. It allows to empirically, and efficiently, investigate the hardness of MDPs, and it implements the first principled benchmark for tabular reinforcement learning algorithms. In the following sections, we report some additional details on the capabilities of `Colosseum`. However, we invite the reader to check the latest online documentation along with the tutorials that cover in detail every aspect of the package.[3]

## A.1  Expected performance indicators

Each agent in `Colosseum` is required to implement a function that returns its current best policy estimate $\hat{\pi}_t^*$ for any time step $t$. Using an efficient implementation of the policy evaluation algorithm, `Colosseum` can compute the corresponding expected regret and expected average reward, which, summed across time steps, amounts to the expected cumulative reward and expected cumulative regret. Although it is possible to perform this operation at every time step of the agent/MDP interaction, we leave the option to approximate the expected cumulative regret by calculating the expected regret every $n$ time steps and assuming that the policy of the agent in the previous $n-1$ time steps would have yielded a similar expected regret. For instance, for $n = 100$, the expected cumulative regret at time step $T = 500$ would be approximated as the sum of the expected regrets calculated at time steps $t = 100, 200, \ldots, 500$ multiplied by 100.

## A.2  Non-tabular capabilities

`Colosseum` is primarily aimed at the tabular reinforcement learning setting. However, as our ultimate goal is to develop principled non-tabular benchmarks, we offer a way to test non-tabular reinforcement learning algorithms on the `Colosseum` benchmark. Although our benchmark defines a challenge that is well characterized for tabular agents, we believe that it can provide valuable insights into the performance of non-tabular algorithms. In order to do so, we adopt the *BlockMDP* formalism proposed by Du et al. [28]. A BlockMDP is a tuple $(\mathcal{S}, \mathcal{A}, P, P_0, R, \mathcal{O}, q)$, where $\mathcal{O}$ and $q : \mathcal{S} \to \Delta(\mathcal{O})$ are respectively the non-tabular observation space that the agent observes and the (possibly stochastic) *emission map* that associates a distribution over the observation space to each state in the MDP. Note that the agent is not provided with any information on the state space $\mathcal{S}$. `Colosseum` implements six deterministic emission maps with different properties and four kinds of noise to make the emission maps stochastic, which we describe below. Examples of the emission maps with distinguishable characteristics for each MDP family will be presented in the corresponding sections.

**Emission maps:**

- *One-hot encoding*. This emission map assigns to each state a feature vector that is filled with zeros with the exception of an index that uniquely corresponds to the state.

- *Linear optimal value*. This emission map assigns to each state a feature vector $\phi(s)$ that enables linear representation of the optimal value function. In other words, there is a $\theta$ such that $V^*(s) = \theta^T \phi(s)$.

- *Linear random value*. This emission map assigns to each state a feature vector $\phi(s)$ that enables linear representation of the value function of the randomly acting policy. In other words, there is a $\theta$ such that $V^\pi(s) = \theta^T \phi(s)$, where $\pi$ is the randomly acting policy.

- *State information*. This emission map assigns to each state a feature vector that contains uniquely identifying information about the state (e.g., coordinates for the DeepSea family).

- *Image encoding*. This emission map assigns to each state a feature matrix that encodes the visual representation of the MDP as a grayscale image.

- *Tensor encoding*. This emission map assigns to each state a tensor composed of the concatenation of matrices that one-hot encode the presence of a symbol in the corresponding indices. For example, for the DeepSea family, the tensor is composed of a matrix that encodes the position of the agent and a matrix that encodes the positions of white spaces.

**Noise:**

---

[3]Available at `https://michelangeloconserva.github.io/Colosseum`.

