# A Colosseum

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

**Noise:**

---

[3]Available at `https://michelangeloconserva.github.io/Colosseum`.

- *Uncorrelated light-tailed noise*. The output of the emission map is corrupted with element-wise uncorrelated Gaussian noise.

- *Correlated light-tailed noise*. The output of the emission map is corrupted with multivariate correlated Gaussian noise with a covariance matrix sampled from a Wishart distribution with a pre-specified scale level when the MDP is created. In other words, the correlation structure of the noise remains unchanged while the agent interacts with the MDP.

- *Uncorrelated heavy-tailed noise*. The output of the emission map is corrupted with element-wise uncorrelated Student's t noise.

- *Correlated heavy-tailed noise*. The output of the emission map is corrupted with multivariate correlated Student's t noise with covariance matrix sampled from a Wishart distribution with a pre-specified scale level when the MDP is created. In other words, the correlation structure of the noise remains unchanged while the agent interacts with the MDP.

## A.3 `Colosseum` **MDP families**

`Colosseum` implements eight families of MDPs. When selecting which families to include in Colosseum, we aimed to balance between traditional environment families (`RiverSwim` [24], `Taxi` [25], and `FrozenLake`) and unconventional ones (`MiniGid` environments [26]). The `DeepSea` family [27] was included since it was proposed as an example of a hard exploration problem. The `SimpleGrid` family acts as a simplified version of the `MiniGrid-Empty` environment.

Each MDP family requires a set of parameters to instantiate an MDP. In addition to individual parameters, all MDP families share the following:

- The `size` $\in \mathbb{N}$ parameter controls the number of states through geometrical properties of the MDP family. For example, in a grid world, it controls the size of the grid. This parameter allows increasing the difficulty of an MDP instance without altering the fundamental structure of the MDP family.

- The `p_rand` $\in [0,1)$ parameter controls the probability $r$ that an MDP executes an action at random instead of the one selected by the agent. Concretely, the new transition kernel is given by $P'(s_{t+1} \mid s_t, a_t) = (1-r)P(s_{t+1} \mid s_t, a_t) + \frac{r}{|\mathcal{A}|} \sum_a P(s_{t+1} \mid s_t, a)$. Setting this parameter to a non-zero value can make a communicating MDP ergodic.

- The `lazy` $\in [0,1)$ parameter controls the probability $l$ of an action not being executed. Concretely, the new transition kernel is given by $P'(s_{t+1} \mid s_t, a_t) = (1-l)P(s_{t+1} \mid s_t, a_t) + l\mathbb{1}(s_{t+1} = s_t)$. This parameter can render a deterministic MDP stochastic without changing the communication class.

- `make_reward_stochastic` is a boolean parameter to render the rewards stochastic instead of deterministic (the default). We opted for a Beta distribution to guarantee rewards bounded in a specific range. However, it is possible to specify custom reward distributions using `scipy` random variables.

- `r_min` and `r_max` scale the rewards. The default values are 0 and 1, respectively.

The hardness analysis presented in Appendix D shows the relationships between the measures of hardness and some of these parameters for the Colosseum MDP families. Such relationships can be easily exploited to create MDP instances with specific hardness characterization. For example, in order to create an MDP instance with low estimation complexity and high visitation complexity, one can force the MDP instance to be deterministic by setting `p_rand` and `p_lazy` to zero and the size to a high value. If instead one wants to increase the estimation complexity while keeping the visitation complexity fixed, the mean reward of a subset of states can be increased. The scale of the increase depends on the variability of the next state distributions of the selected states. The more variable such distributions are the higher the increase in estimation complexity.

### A.3.1 `RiverSwim`

The `RiverSwim` MDP has been introduced by Strehl and Littman [24] as a simple but challenging MDP. This MDP is a chain of states where the agent can only move between adjacent states. The agent starts in the leftmost state. We have removed the *current* mechanism proposed by Strehl and

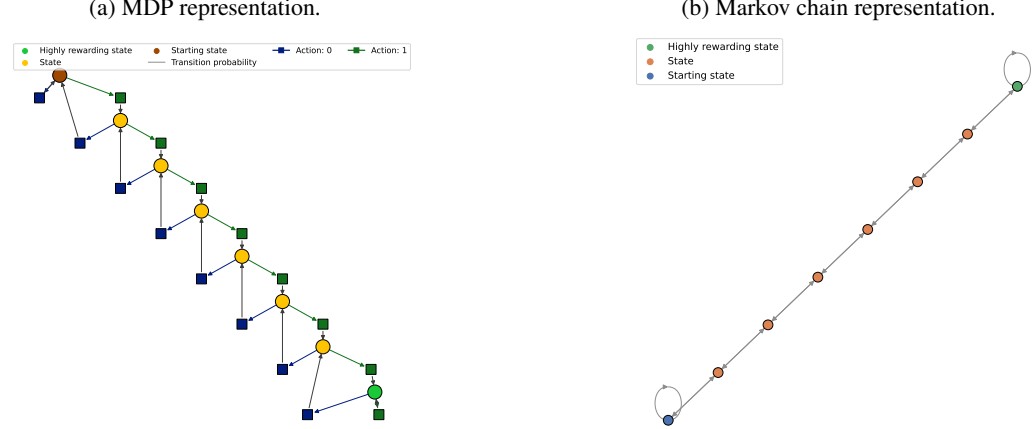

(a) MDP representation.

(b) Markov chain representation.

Figure 4: `RiverSwim` MDP with length eight.

Littman [24], which increases the difficulty of moving right since we provide more general controls (namely, `p_lazy` and `p_rand`). The agent is given a small reward for staying in the initial state, but it can obtain a large reward in the rightmost state. The challenge of `RiverSwim` is that the agent has to travel all the states in the chain in order to discover the highly rewarding state. The chain structure is evident in the visual representations in Fig. 4. In the textual representation for the `RiverSwim` MDP, the letter `A` encodes the position of the agent, the letter `S` represents the starting state, and the letter `G` represents the position of the highly rewarding state.

Table 3: RiverSwim emission map examples for a given state.

| (a) | Textual state representation. | (b) | State information emission map. | (c) | Image encoding emission map. | (d) | Tensor encoding emission map. |
|-----|-------------------------------|-----|--------------------------------|-----|------------------------------|-----|-------------------------------|

'S' ' ' ' ' 'A' ' ' ' ' ' ' 'G'        [2.]

### A.3.2 `DeepSea`

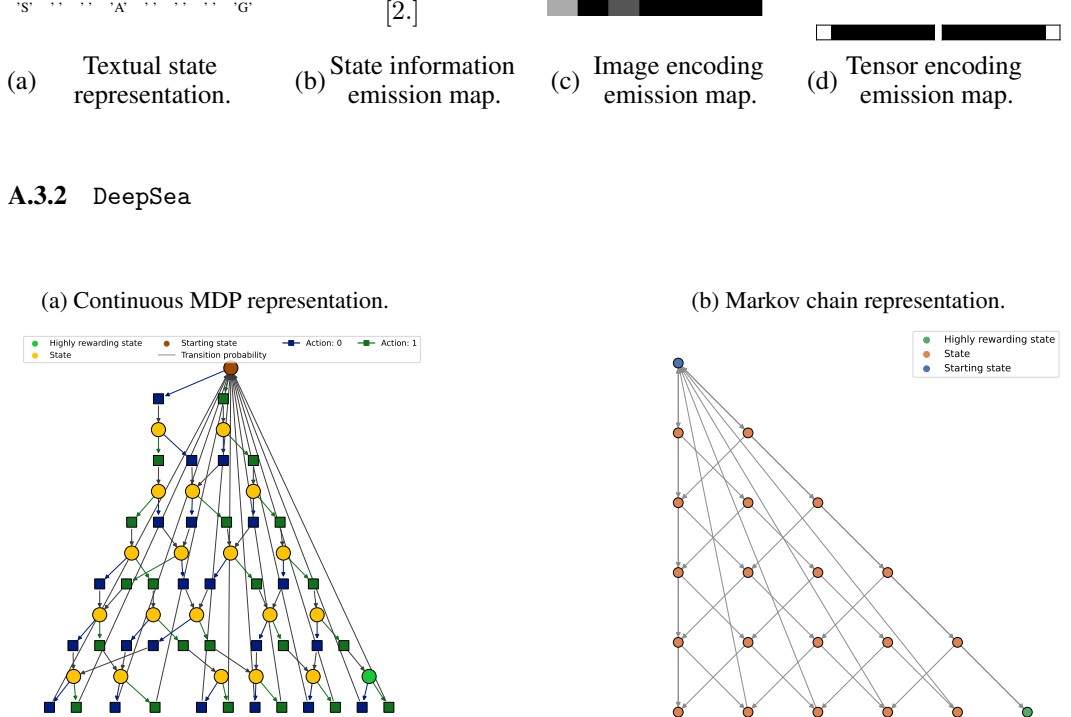

(a) Continuous MDP representation.

(b) Markov chain representation.

Figure 5: `DeepSea` MDP with size six.

The `DeepSea` MDP has been introduced by Osband et al. [27] as a *deep exploration* challenge. The MDP is a pyramid of states in which the agent starts at the top and dives a step down at each time step. Depending on the action chosen, the agent can either dive to the right or to the left. Once the agent reaches the base of the pyramid, it restarts from the top. The agent is rewarded when diving to the left, but a large reward can be obtained by reaching the bottom rightmost state. The main difficulty of `DeepSea` is that the agent will not be able to reach the highly rewarding state before it restarts if it chooses a single *wrong* action. We removed the `p_lazy` parameter from `DeepSea` due to the particular structure of this MDP. In the episodic setting, staying in the same state even once would make reaching the goal state impossible. The pyramid structure is evident in the visual representations in Fig. 5. In the textual representation for the `DeepSea` MDP, the letter `A` corresponds the to position of the agent.

Table 4: DeepSea emission map examples for a given state.

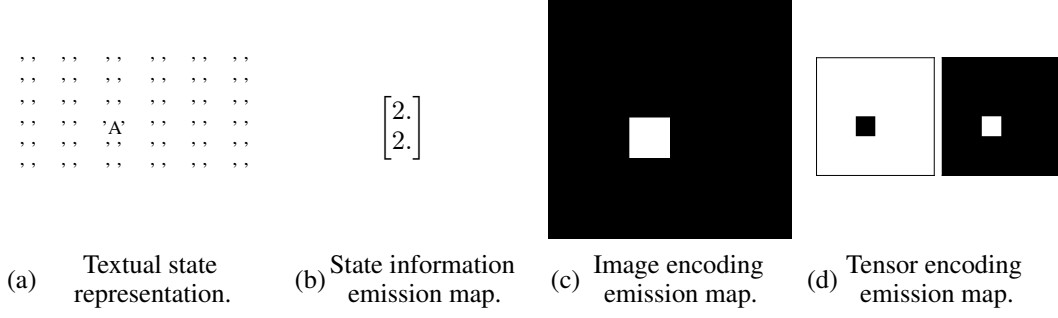

(a) Textual state representation.  (b) State information emission map.  (c) Image encoding emission map.  (d) Tensor encoding emission map.

### A.3.3 `SimpleGrid`

(a) Continuous MDP representation.                (b) Markov chain representation.

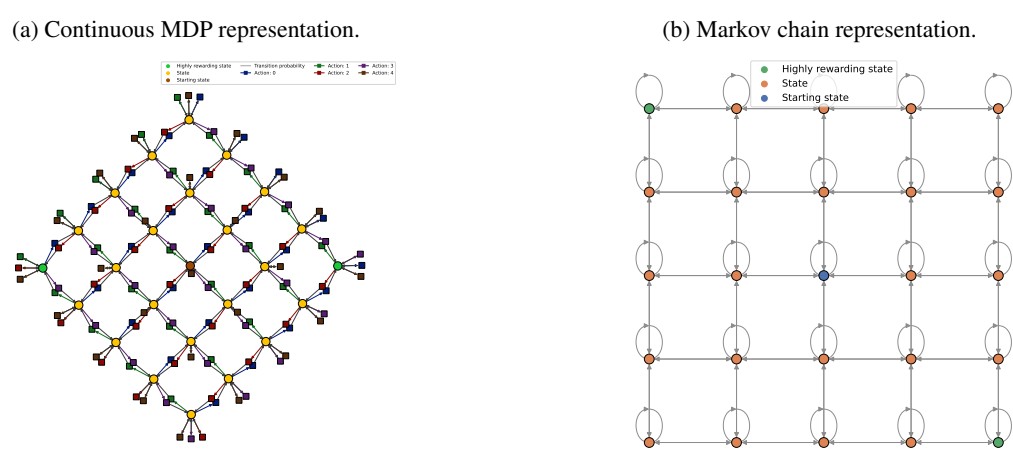

Figure 6: `SimpleGrid` MDP with size five.

The `SimpleGrid` MDP is a grid world in which the agent can either choose to move in one of the cardinal directions or stay in the same position. At each time step, the agent is given a small reward. Depending on the parameters of the MDP, some corner states yield a high reward whereas the other ones produce close to zero rewards. The starting states are selected from the states in the center of the grid in order to be as far as possible from the corners. The grid structure is clearly distinguishable in the visual representation in Fig. 6. In the textual representation for the `SimpleGrid` MDP, the letter `A` encodes the position of the agent and the symbols $+$ and $-$ represent the states that yield large reward and zero reward, respectively.

Table 5: SimpleGrid emission map examples for a given state.

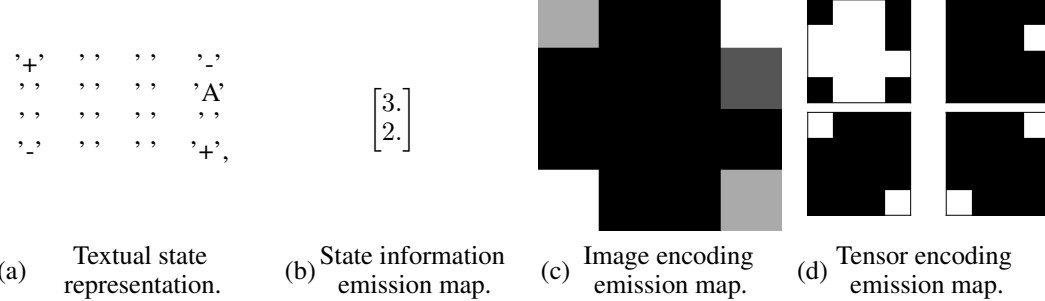

(a) Textual state representation.

(b) State information emission map.

(c) Image encoding emission map.

(d) Tensor encoding emission map.

### A.3.4 MiniGrid MDPs

Gym MiniGrid (MG) [26] is an important testbed for non-tabular reinforcement learning agents. It presents several families of MDPs that produce different challenges. The base structure is a grid world where an agent can move by going forward, rotating left, and rotating right. Depending on the MDP family, the agent can have further access to actions such as pick, drop, and interact. The goal is always to reach a highly rewarding state. `Colosseum` implements three families of MiniGrid MDPs: `MG-Empty`, `MG-Rooms`, and `MG-DoorKey`.

(a) Continuous MDP representation.

(b) Markov chain representation.

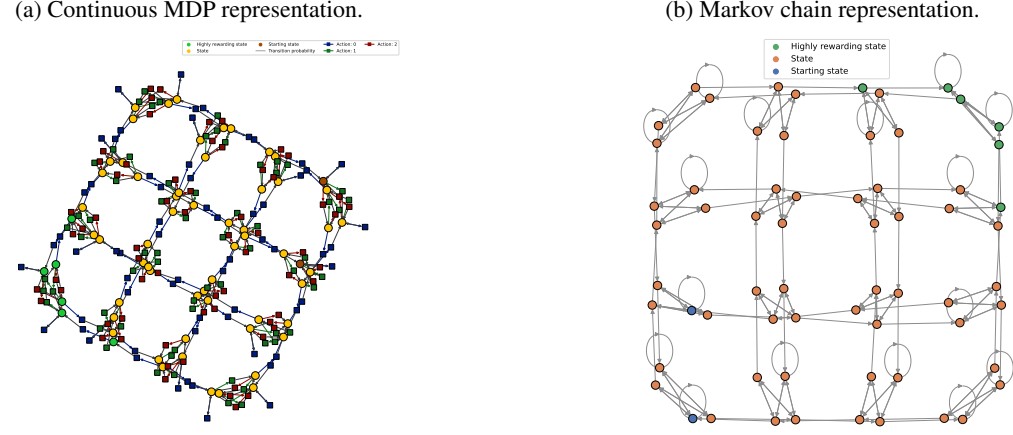

Figure 7: `MG-Empty` MDP with size four.

The `MG-Empty` MDP contains only the basic structure of an MG environment. The starting states are selected from a randomly selected border of the grid and the highly rewarding state is located on the border at the opposite side of the grid. Although `MG-Empty` appears similar to `SimpleGrid`, it implements a more complex mechanism to move between states that results in a completely different transition structure, which is evident in the visual representations in Fig. 7. Each group of four states corresponds to the agent rotating in the same position. In the textual representation for the `MG-Empty` MDP, the symbols >,v,<, and ∧ encode the position and the rotation of the agent, whereas the letter G represents the position of the goal.

The `MG-Rooms` is a collection of grids connected with narrow passages. The presence of such bottlenecks produces a significantly higher challenge for exploration when compared to open grids, especially when `p_rand` is non-zero. The rooms are evident in the visual representations in Fig. 8. In the textual representation for the `MG-Rooms` MDP, the symbols >,v,<, and ∧ encode the position and the rotation of the agent, the letter W represents a wall, and the letter G represents the goal.

In the `MG-DoorKey` environment, the grid world is divided into two rooms separated by a wall with a door that can be opened using a key. The key is positioned in the same room where the agent starts. Differently from `MG-Empty` and `MG-Rooms`, in this case, the agent has all the five actions available

Table 6: MiniGrid-Empty emission map examples for a given state.

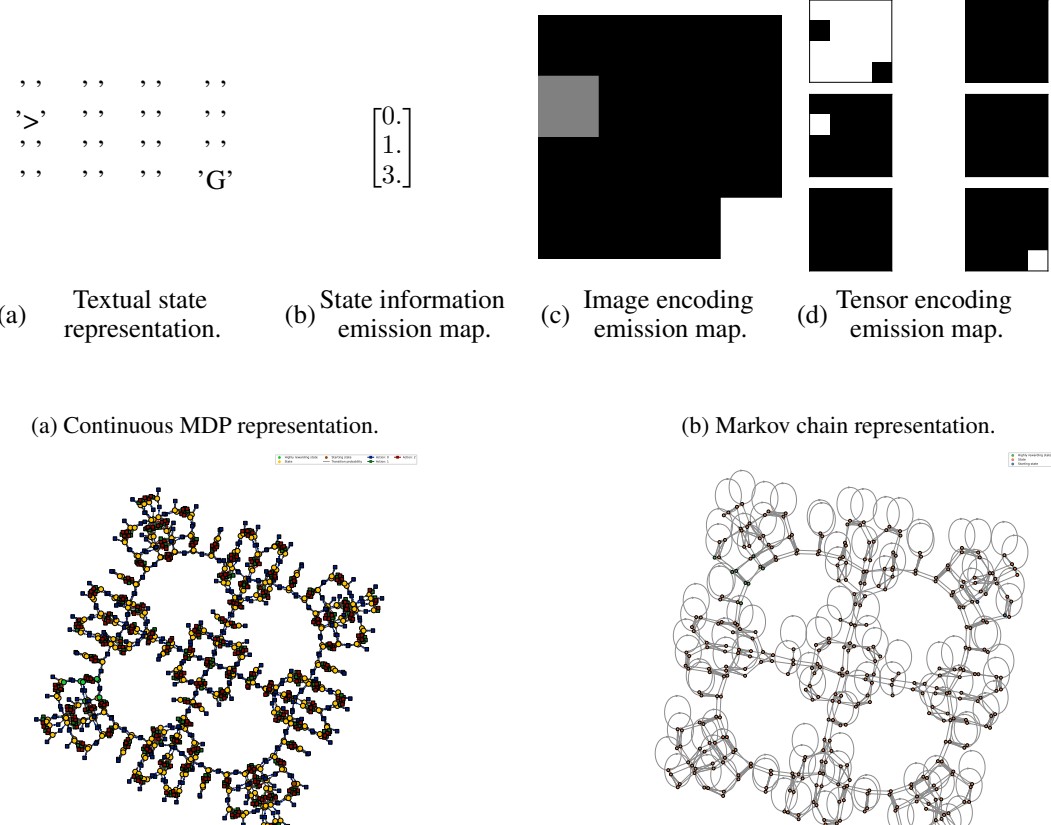

| (a) | Textual state representation. | (b) | State information emission map. | (c) | Image encoding emission map. | (d) | Tensor encoding emission map. |

(a) Continuous MDP representation.

(b) Markov chain representation.

Figure 8: `MG-Rooms` MDP with nine rooms of size three.

to allow it to interact with the key and the door. `MG-DoorKey` is a particularly challenging MDP for several reasons. First, the agent has to take a very long sequence of actions before reaching the highly rewarding state. Further, the action that picks the key produces effect only in the very few states in which the agent is correctly positioned in front of the key, and the action that opens the door only has effect in the single state in which the agent has the key and is correctly positioned in front of the door. In this case, it is not possible to clearly identify the structure of the grid in the visual representations in Fig. 9. In the continuous setting, every MDP instance from this class is weakly-communicating. Once the door has been opened, it is not possible to close it. In the textual representation for the `MG-DoorKey` MDP, the symbols >, v, <, and ∧ encode the position and the rotation of the agent, the

Table 7: MiniGrid-Rooms emission map examples for a given state.

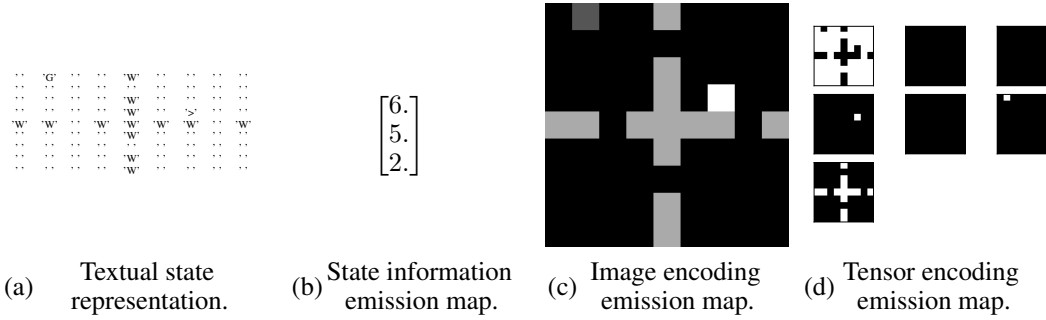

| (a) | Textual state representation. | (b) | State information emission map. | (c) | Image encoding emission map. | (d) | Tensor encoding emission map. |

(a) Continuous MDP representation.

(b) Markov chain representation.

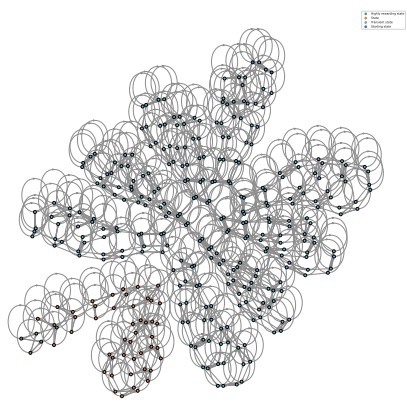

Figure 9: `MG-DoorKey` MDP with size four.

letter K represents the key, the letters C and O respectively stand for closed door and opened door, and the letter G represents the goal.

### A.3.5   `FrozenLake`

(a) Continuous MDP representation.

(b) Markov chain representation.

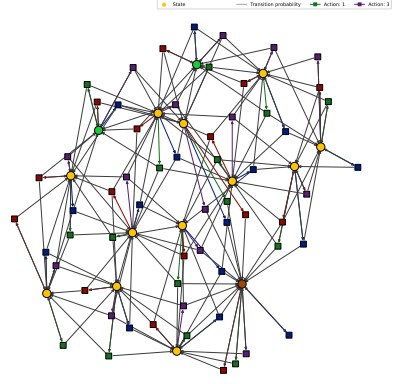
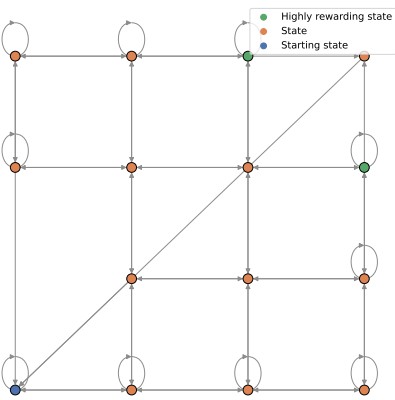

Figure 10: `FrozenLake` MDP with size five.

The `FrozenLake` MDP is a grid world where the agent has to walk over a frozen lake to reach a highly rewarding state. Some tiles of the grid are walkable, whereas others represent holes in which the agent may fall (leading back to the starting point). The agent can move in the cardinal directions. However, movement on the walkable tiles is not entirely deterministic, and so the agent risks falling into holes if it walks too close to them. The agent receives a small reward at each time step and zero reward when it falls into a hole. The starting position of the agent is the bottom left state, which is the one farthest away from the goal position. The challenge presented by `FrozenLake` lies in the high stochasticity of the movement. A successful agent has to learn to balance the risk of falling into holes with reaching the goal quickly. The structure of the grid is evident in the Markov chain representation but not in the MDP representation, as shown in Fig. 10. In the textual representation for the `FrozenLake` MDP, the letter A corresponds to the position of the agent, the letter F represents a frozen tile over which the agent can safely walk, the letter H stands for a hole, and the letter G represents the goal.

Table 8: FrozenLake emission map examples for a given state.

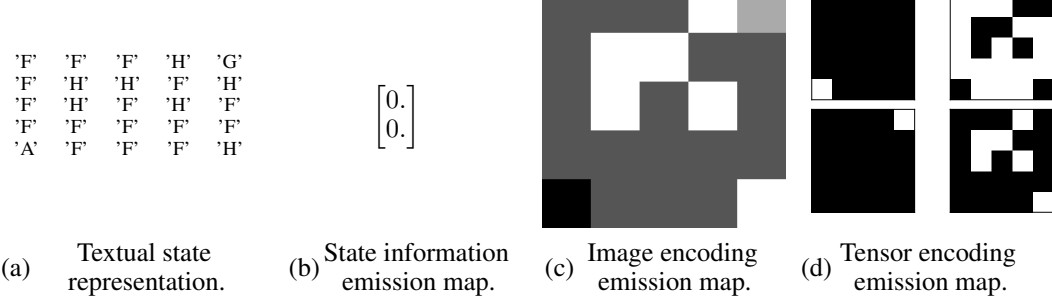

| | | | | |
|---|---|---|---|---|
| 'F' | 'F' | 'F' | 'H' | 'G' |
| 'F' | 'H' | 'H' | 'F' | 'H' |
| 'F' | 'H' | 'F' | 'H' | 'F' |
| 'F' | 'F' | 'F' | 'F' | 'F' |
| 'A' | 'F' | 'F' | 'F' | 'H' |

$$\begin{bmatrix} 0. \\ 0. \end{bmatrix}$$

(a) Textual state representation.  (b) State information emission map.  (c) Image encoding emission map.  (d) Tensor encoding emission map.

### A.3.6  Taxi

(a) Continuous MDP representation.          (b) Markov chain representation.

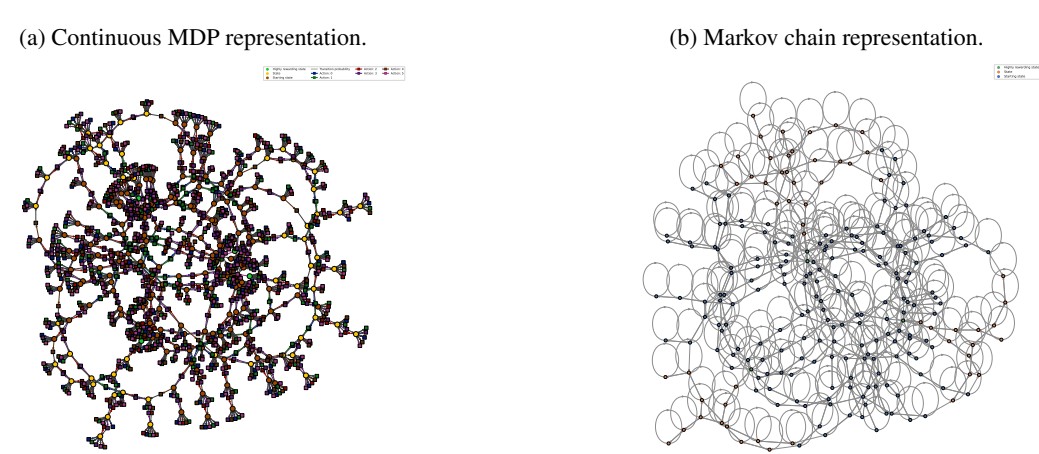

Figure 11: Taxi MDP with size four.

The Taxi MDP is a grid world where the agent has to pick up and drop off passengers [25]. Each time a passenger is taken to the correct location a new passenger and destination appear. The agent has six actions available, which correspond to the cardinal directions, picking a passenger, and dropping off a passenger. The agent receives a large reward when it drops off a passenger at the correct destination and zero reward when it tries to drop off a passenger at an incorrect destination. At every other time step, it receives a small reward. The transition and reward structures of the Taxi MDP are particularly challenging to visualize due to the complexity of the task, as can be seen in Fig. 11. In the textual representation for the Taxi MDP, the letter A encodes the position of the agent, the letter W represents a wall, the letter P represents the position of the passenger, and the letter D represents the final destination.

Table 9: Taxi emission map examples for a given state.

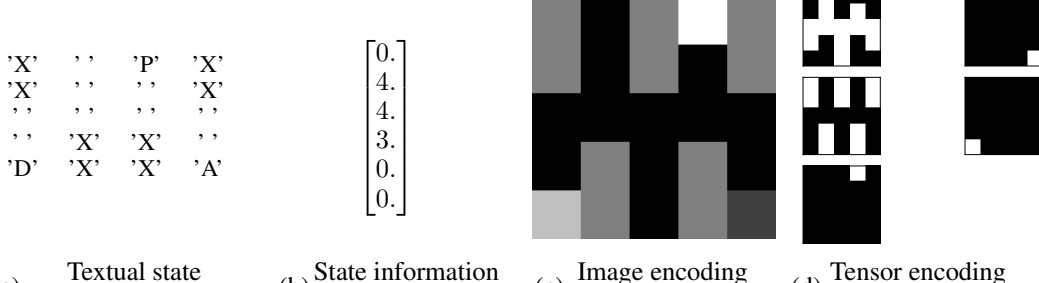

| (a) | Textual state representation. | (b) | State information emission map. | (c) | Image encoding emission map. | (d) | Tensor encoding emission map. |

## A.4 Diameter calculation

Recall that the diameter $D$ is defined as

$$D := \sup_{s_1 \neq s_2} \inf_{\pi} T^{\pi}_{s_1 \to s_2}.$$

From this definition, it is evident that the main challenge to computing the diameter is the infimum over policy space, which grows exponentially with the number of states and actions.

For Markov chains, the expected time to arrival $T_{s \to e}$ from state $s$ to state $e$ can be expressed recursively as

$$T_{s \to e} = \mathrm{P}(e \mid s) + \sum_{s'} \mathrm{P}(s' \mid s)(1 + T_{s' \to e}).$$

In the case of an MDP M, we may derive a Bellman optimality equation for a policy that minimizes the expected arrival time to a given state. Concretely, the expected time to arrival $T^*_{s \to e}$ when following the best policy to transition from state $s$ to state $e$ is given by

$$T^*_{s \to e} = \min_{a \in \mathcal{A}} \left( P(e \mid s, a) + \sum_{s'} P(s' \mid s, a)(1 + T^*_{s' \to e}) \right). \tag{4}$$

In other words, consider an MDP $\mathrm{M}'_e$ that is identical to M except for the reward kernel, which is given by $R(s, a) = -\mathbb{1}(s \neq e)$. By calculating the optimal value function for $\mathrm{M}'_e$ and taking the complement of its minimum across states, we may find $\max_s T^*_{s \to e}$. The diameter is obtained by computing this quantity for each state $e$ and selecting the maximum. This operation can be implemented efficiently through parallelization. Note that, in order to preserve the exact value of the diameter, we do not include a discount factor in Equation 4. As such, the resulting procedure is not yet guaranteed to converge. In practice, this does not represent an issue. We believe that a formal proof of convergence for the Bellman update corresponding to Equation 4 is possible.

# B    Measures of hardness computational complexity

Previous literature has never been concerned with the computational complexity of hardness measures as they were not designed to be practically computed. For this reason, there are no available efficient algorithms to compute most measures of hardness. In the following, we provide a discussion of the challenges in the computational complexity of the measures of hardness from Section 2.

## B.1    Mixing time

Recall that the mixing time of an MDP is defined as the maximum of the mixing time of the Markov chain yielded by a policy. Considering that the development of the efficient computation of the mixing time of a Markov chain is in itself an active area of research [33] and that the policy space grows exponentially in the cardinality of the state and action spaces, it becomes clear that the development of an efficient algorithm to compute the mixing time is extremely challenging, if not impossible.

## B.2    Diameter

Recall that the diameter is defined as the expected number of time steps required to transition between two states when following the best policy for doing so. Similarly to the mixing time, computing the diameter involves a minimization in policy space. However, we were able to remove the dependency on the exponentially growing policy space by noting that the diameter of an MDP M can be related to solving $|S|$ MDPs that are closely related to M with a discount factor of 1 (App. A.4). Lee and Sidford [34] propose a linear program to solve the non-linear Bellman equation that has $\tilde{O}(|S|^{2.5}|A|)$ computational complexity and no $\gamma$ dependency. Since computing the diameter requires solving $|S|$ MDPs, the resulting computational complexity is $\tilde{O}(|S|^{3.5}|A|)$.

## B.3    Distribution mismatch coefficient

Recall the the computation of the distribution mismatch coefficient involves a maximization in policy space and the stationary distribution. Although the stationary distribution of a Markov chain can be efficiently computed with several algorithms including linear programming, an algorithm that allows to remove the dependency on the exponentially growing policy space is not currently known for this measure of hardness.

## B.4    Action-gap regularity

Since the action-gap regularity is defined as a constant in the upper bound of an integral, a closed form equation for this measure of hardness is not available. Further, it has not been extended to MDPs with more than two actions.

## B.5    Environmental value norm

When the environmental value norm is to be interpreted as a measure of hardness for an MDP, Maillard et al. [19] suggest to use the environmental value norm of the optimal policy, which can be efficiently computed as the norm of the optimal state value function with respect to the transition probabilities of the Markov chain yielded by the optimal policy. Since the norm is computed as a simple matrix product, the dominating term in the computational complexity is the calculation of the optimal state value function. See Table 2 in Sidford et al. [35] for the computational complexity of several available algorithms. In Table 1, we report the computational complexity of value iteration since it is the most widely known of such algorithms.

## B.6    Sum of the reciprocals of the sub-optimality gaps

Similarly to the environmental value norm, the computational complexity of the sum of the reciprocals of the sub-optimality gaps is dominated by the calculation of the optimal state value function.

# C Normalization procedures

**Measures of hardness.** The normalization of the theoretical measures of hardness (diameter, environmental value norm, and the sum of the reciprocals of the sub-optimality gaps) is carried out by scaling the values to the range $[0, 1]$ given their maximum and minimum values. For example, in the empirical investigation of the measures of hardness (see Sec. 3.2), the maximum and minimum values are taken across all the different seeds and parameters. For the cumulative regret of the near-optimal agent (when used as an optimistic measure of hardness), we leverage the per-step normalization procedure that is described in the next paragraph. This procedure takes into account the bounded range of the optimal return of an MDP.

**Expected cumulative regret.** In the episodic case, the per-step normalized version of the episodic regret is obtained by dividing the regret by the difference between the value of the optimal policy and the value of the policy with the least value. For a policy $\pi$, this is given by

$$\frac{V^*_{0,\text{epi}}(s_0) - V^\pi_{0,\text{epi}}(s_0)}{V^*_{0,\text{epi}}(s_0) - V^-_{0,\text{epi}}(s_0)} \in [0, 1],$$

where $\pi^- = \arg\min_{\pi \in \Pi} V^\pi_{0,\text{epi}}(s_0)$. In the continuous case, the per-step normalized average instantaneous regret is obtained similarly as

$$\frac{\rho^* - \frac{1}{t} \sum_{k=0}^{t} r_t}{\rho^* - \rho^-} \in [0, 1],$$

where $r_t$ is the reward obtained by the policy at time step $t$ and $\rho^- = \min_{\pi \in \Pi} \rho^\pi$.

# D Empirical investigation of hardness measures

In our empirical investigation of the measures of hardness, we consider five MDP families (`MG-Empty`, `SimpleGrid`, `FrozenLake`, `RiverSwim`, and `DeepSea`) that include different levels of stochasticity and challenge. Each MDP family is tested in four scenarios that highlight different aspects of hardness. Note that each measure has been normalized (as described in App. C), which *solely* allows comparing trends (growth rates). Figures 12, 13, 14, 15 and 16 (pg. 28) report the results of our investigation along with the 95% bootstrapped confidence intervals over twelve seeds.

**Scenario 1.** We vary the probability `p_rand` that an MDP executes a random action instead of the action selected by an agent. As `p_rand` increases, estimating the optimal value function becomes easier since every policy yields increasingly similar value functions. This produces a decrease in the estimation complexity. However, intentionally visiting states becomes harder.

**Scenario 2.** We vary the probability `p_lazy` that an MDP stays in the same state instead of executing the action selected by an agent. Contrary to increasing `p_rand`, increasing `p_lazy` never benefits exploration through the execution of random actions. Increasing `p_lazy` decreases estimation complexity and increases visitation complexity.

**Scenario 3 and 4.** We vary the number of states across MDPs from the same family. In scenario 4, we also let `p_rand = 0.1` to study the impact of stochasticity. In these scenarios, increments in the number of states increase both estimation complexity and visitation complexity.

**Cumulative regret of a *near-optimal* agent.** In every scenario, the measures of hardness are compared with the cumulative regret of a *near-optimal* agent that serves as an optimistic approximation of a complete measure of hardness. The near-optimal agents have been chosen between the ones available in `Colosseum` with the lowest average cumulative regret in the benchmarking results (such as PSRL in the episodic setting and UCRL2 in the continuous setting). In order to optimistically approximate a complete measure of hardness, we tune the hyperparameters of the agents for each MDP in every scenario. Concretely, we perform a random search with the objective of minimizing the average cumulative regret resulting from an interaction of the agent with the MDP that lasts for 200 000 time steps with a maximum time limit of two minutes across three seeds. The budget for the random search is 120 samples.

**Computational power.** The empirical investigation has been carried out on a desktop PC equipped with an *AMD Ryzen 9 5950X 16-Core Processor* and required less than 24 hours for all the MDP families and scenarios. The most computationally intensive part of the procedure is the hyperparameter search.

**Limitations.** The main limitation of our empirical investigation is the selection of the MDP families, near-optimal agents, and scenarios. Although we believe to have proposed a solid methodology, we are open to discussing the inclusion of additional experiments to further enhance `Colosseum`.

## D.1 Analysis of results

**Diameter.** The diameter grows superlinearly with both `p_rand` and `p_lazy` since deliberate movement between states requires an exponentially increasing number of time steps. As clearly shown in the figures, this phenomenon is exacerbated in the episodic setting. If the agent is forced to take a random action or to stay in the same state, it can miss the opportunity to reach the target state in the current episode and has to try again in the next episode. Although the diameter highlights this sharply increasing visitation complexity, its trend overestimates the increase in cumulative regret of the tuned near-optimal agent, which is explained by the unaccounted decrease in estimation complexity. The diameter also increases almost linearly with the number of states. When `p_rand` is relatively small, an approximately linear relationship can still be observed. This linear trend underestimates the non-linear growth in hardness clearly shown in the cumulative regret of the tuned near-optimal agent but is in line with the mild increase in visitation complexity. `FrozenLake` in the episodic setting (Figures 14c and 14d) represents the only exception. Given the extremely high level of stochasticity of the MDP, increasing the number of states drastically increases the visitation complexity while making it easier for the agent to act near-optimally.

**Environmental value norm.** The environmental value norm decreases as `p_lazy` and `p_rand` increase because the optimal value of neighboring states becomes closer, which decreases the per-step variability of the optimal state value function. However, we note that for the `MG-Empty` and the `FrozenLake` MDP families in the continuous cases (see Figures 12f and 14f) as `p_lazy` increases, the environmental value norm first decreases and later increases. From a certain value of `p_lazy` onward, there is a significant probability of the agent remaining in the same state. This provokes large changes in the value of states that are distant from the highly rewarding states and no changes at all for highly rewarding states since the *lazy* transition is comparable to taking the optimal action. Due to the large changes in the suboptimal region of the state space and the absence of changes in the optimal region of the state space, the overall one-step variability of the state value function increases. Note that this does not happen in the episodic case due to the restarting mechanism and whether it happens or not in the continuous case depends on the transition and reward structure of an MDP. When the number of states increases but the transition and reward structures remain the same, the small increase in measured variability only causes the environmental value norm to grow sublinearly. These findings are strong evidence that this measure is only suited to capture estimation complexity.

**Sum of the reciprocals of the sub-optimality gaps.** The sum of the reciprocals of the sub-optimality gaps increases weakly superlinearly in scenarios 1 and 2. The probability of executing the action selected by the agent decreases when `p_lazy` and `p_rand` increase, and so the difference between the optimal value function and the optimal state-action value function decreases sharply. `FrozenLake` (Figures 14a, 14b, 14e and 14f) represents an exception as the sum of the reciprocals of the sub-optimality gaps is almost constant. `FrozenLake` naturally incorporates an exceptionally high level of stochasticity and so varying `p_lazy` and `p_rand` does not significantly affect the value functions. The sum of the reciprocals of the sub-optimality gaps increases almost linearly with the number of states. This is explained by the fact that the average value of the additional terms in the summation is often similar to the average value of the existing terms given the same structure of reward and transition kernels. This measure of hardness is not particularly apt at capturing estimation complexity, since it focuses solely on optimal policy identification. It also underestimates the increase in hardness induced by an increase in visitation complexity.

**Cumulative regret of the tuned near-optimal agent.** The trends of the cumulative regret of the tuned near-optimal agent present more variability when compared to the theoretical measures of hardness. This reflects the fact that this is an approximation of a complete measure of hardness based on agents that have specific strengths and weaknesses. Overall, we note a tendency of superlinear growth in scenarios 1 and 2. Such tendency is specifically marked for the grid worlds, such as `MG-Empty` (see Fig. 12) and `SimpleGrid` (see Fig. 13). In these MDP families, the highly rewarding states are located far from the starting states and therefore the visitation complexity plays a fundamental role. In the `FrokenLake` family (see Fig. 14), the trend is linear (episodic setting) or sub-linear (continuous setting), which is caused by the relatively low impact of the parameters `p_rand` and `p_lazy` in the already highly stochastic MDPs. The cumulative regret of the tuned near-optimal agent presents a moderately superlinear growth in the episodic case and remains almost constant for the `RiverSwim` family (see Fig. 15). This results from the fact that, in the continuous case, the absence of the restarting mechanism in combination with the chain structure of the MDP allows the agent to suffer only minimal impact from the increasing values of the parameters `p_rand` and `p_lazy`. Finally, for the `DeepSea` family, the regret is constant. Increases in `p_rand` dramatically reduce the possibility of visiting the highly rewarding state due to the pyramid structure of the MDP. In scenarios 3 and 4, the overall tendency is still superlinear but less marked compared to scenarios 1 and 2. The superlinear growth is most evident for the `MG-Empty` family (see Fig. 12) and for the `RiverSwim` family in the episodic case (see Figures 15c and 15d). For the `MG-Empty`, when the grid size is increased, and with it the number of states, the MDP becomes increasingly challenging to navigate since the agent has to coordinate its rotation with its forward movement in order to effectively transition between states. For the `RiverSwim` family, the challenge comes from the restarting mechanism on a chain structure. The agent is required to take a perfect sequence of actions in order to visit the last state of the chain, otherwise it will be reset to the start. In the continuous setting (see Figures 15g and 15h), instead, the trends are mostly linear, similarly to what happens in scenarios 1 and 2. The less challenging structure of the `SimpleGrid` family (see Fig. 13) induces weakly superlinear trends of the cumulative regret of the tuned near-optimal agent. We note that, contrary to the theoretical measures of hardness, the episodic setting does not appear to be harder (which would be suggested by steeper trends). This discrepancy is particularly noticeable in the

`FrozenLake` family (see Fig. 14) which yields a mostly linear trend in the continuous settings and clearly sublinear trends in the episodic settings. In the `DeepSea` family, the cumulative regret of the tuned near-optimal agent is almost constant in scenario 3 and almost linear in scenario 4. The main challenge for this family lies in the pyramidal structure of the MDP rather than the number of states. However, setting `p_rand` = 0.1 creates a more challenging task for the agent as more time steps are required to find the highly rewarding state. We also note that the difference in results between the episodic and continuous settings is minimal, which is unsurprising given the MDP structure.

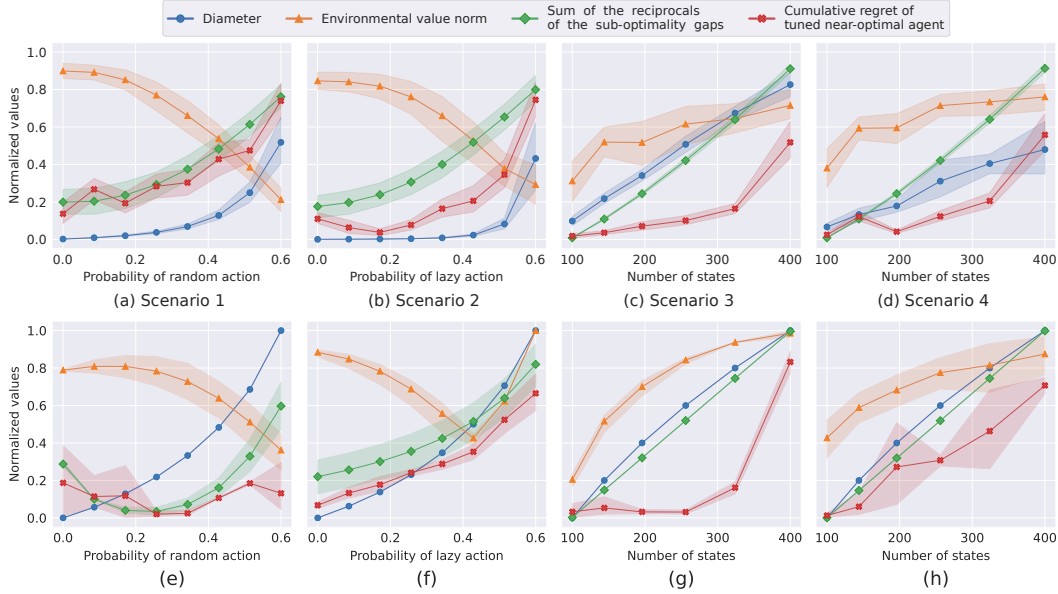

Figure 12: `MiniGridEmpty` results in the episodic (top) and continuous (bottom) settings, scenarios 1-4 correspond to the columns from left to right.

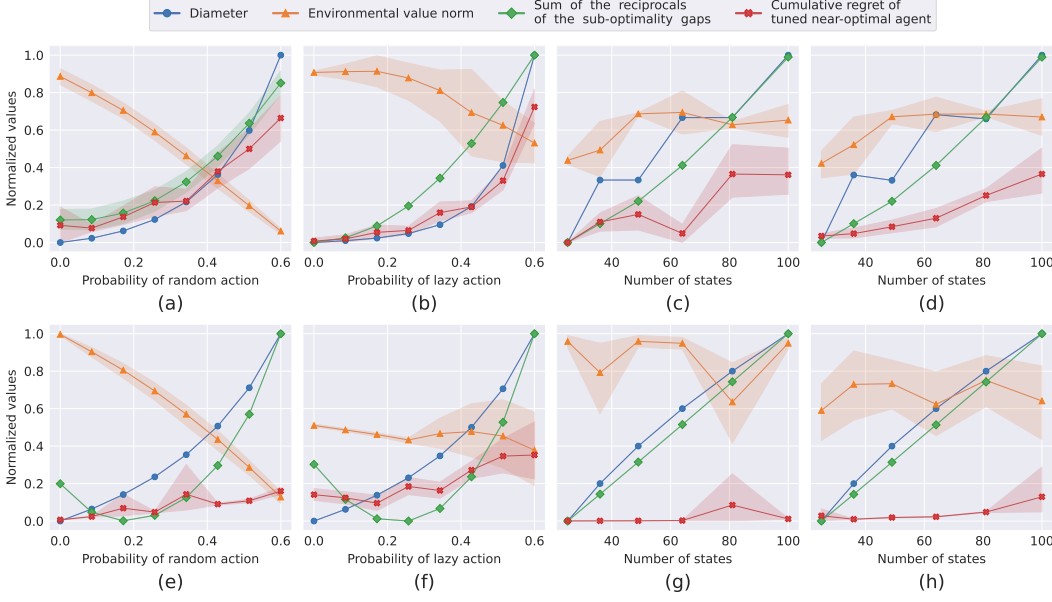

Figure 13: `SimpleGrid` results in the episodic (top) and continuous (bottom) settings, scenarios 1-4 correspond to the columns from left to right

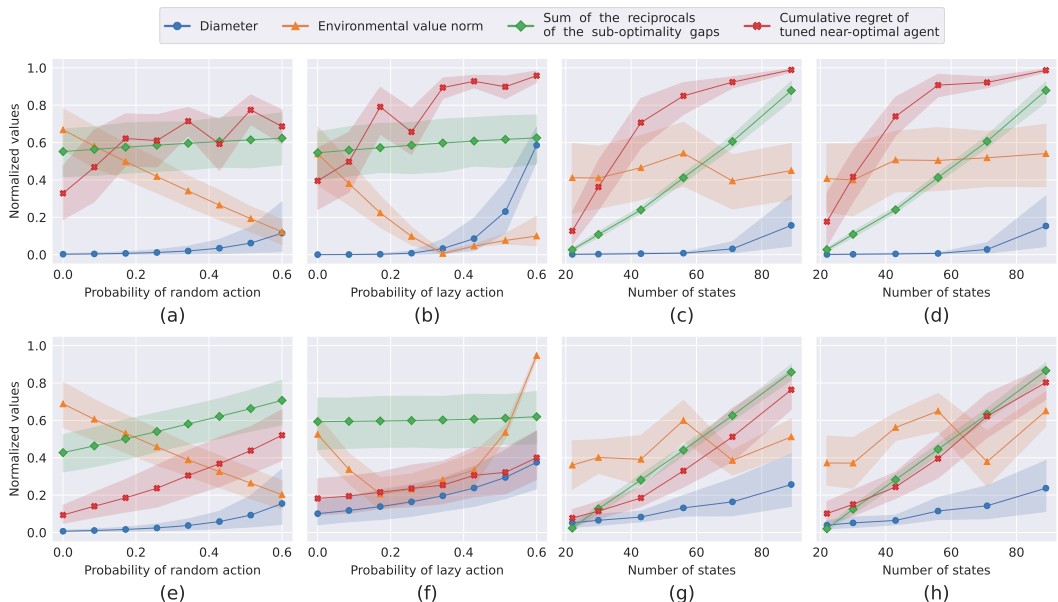

Figure 14: `FrozenLake` results in the episodic (top) and continuous (bottom) settings, scenarios 1-4 correspond to the columns from left to right

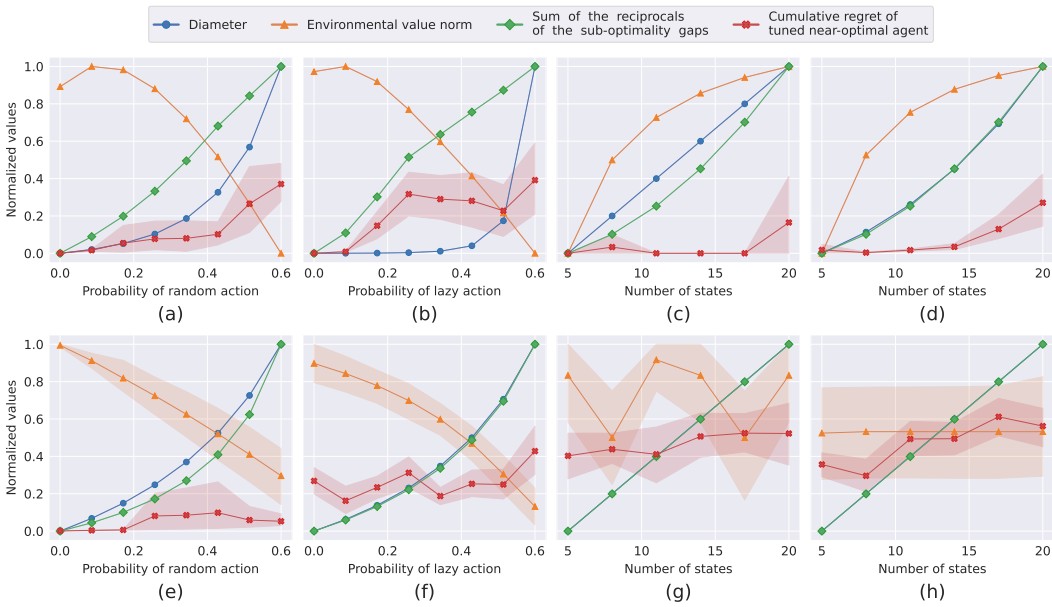

Figure 15: `RiverSwim` results in the episodic (top) and continuous (bottom) settings, scenarios 1-4 correspond to the columns from left to right

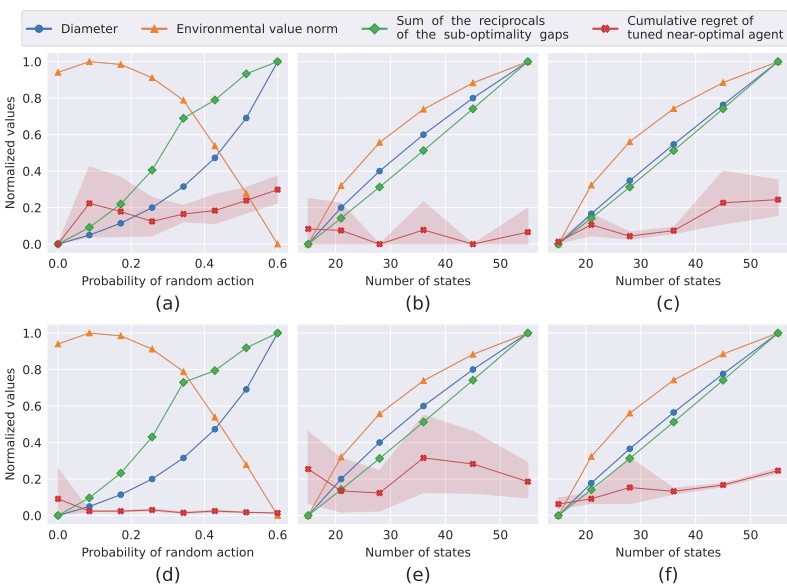

Figure 16: `DeepSea` results in the episodic (top) and continuous (bottom) settings, scenarios 1, 3, and 4 correspond to the columns from left to right

# E    Benchmarking

In the following sections, we provide details on the selection methodology for the environments in the benchmark, and we explain the full benchmarking procedure from the hyperparameters selection to the benchmark evaluation.

## E.1    Benchmark environments selection

The environments in the benchmark have been selected to be as diverse as possible with respect to the diameter and the environmental value norm. Based on the theoretical properties of these measures and the results of the empirical comparison in Section D, we believe that they represent valid proxies for the visitation complexity and the estimation complexity. The candidate environments have been sampled from a set of parameters such that their diameter is less than $100$ and the environmental value norm is less than $3.5$. This guarantees a sufficient challenge for the reinforcement learning agents while limiting the scale of the environments.

Figure 17 represents the benchmark MDPs placed according to their diameter and environmental value norm. The selection features MDPs with varying combinations of values of diameter in the interval $[20, 100]$ and environmental value norm in the interval $[0, 3.5]$

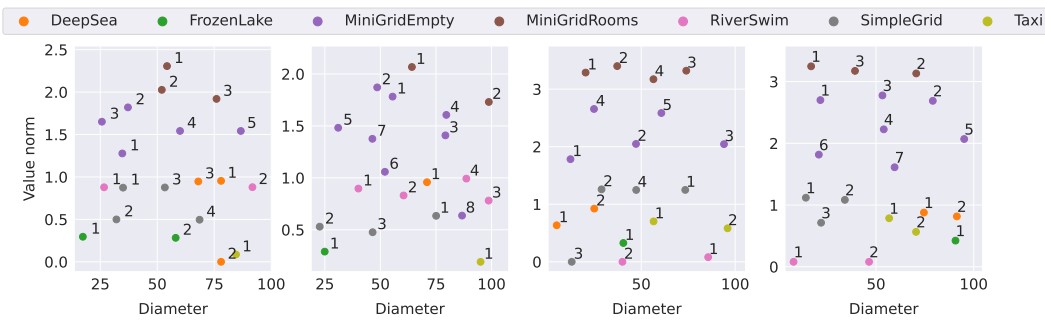

Figure 17: Positions in measure of hardness space of the set of MDPs in the benchmark.

## E.2    Hyperparameter selection

The hyperparameter selection procedure is to be considered an integral component of the `Colosseum` benchmarking procedure to ensure fair hyperparameter tuning.

Each `Colosseum` agent is required to define a sampling space for all its parameters. These sampling spaces are used by the package to conduct a random search optimization procedure with the objective of minimizing the cumulative regret across a set of randomly sampled environments. The random sampling procedure for environments is defined for each `Colosseum` MDP family and aims to provide a varied set of MDPs of up to moderate scale. Note that the *MG-DoorKey* family is excluded from the hyperparameter selection as it is weakly communicating in the continuous case. Tutorials on how to implement the aforementioned functions for novel agents and environments are available online.

The hyperparameters for the agents employed in the paper have been obtained with $50$ samples from the hyperparameter spaces, which have been evaluated on $12$ MDPs from each family, for a total of $84$ MDPs with a training time of $20$ minutes and a maximum number of total time steps of $200\,000$.

## E.3    Computational resources

The experiments for the benchmarking procedure have been carried out using CPUs from the Queen Mary University of London Apocrita HPC facility. Note that, due to the time constraint imposed by `Colosseum`, the computational resources required to run the benchmark are bounded, and the benchmarking procedure is easily parallelizable.

## E.4 Tabular setting

**Experimental procedure.** We set the total number of time steps to $500\,000$ with a maximum training time of 10 minutes for the tabular setting and 40 minutes for the non-tabular case. If an agent does not reach the maximum number of time steps before this time limit, learning is interrupted, and the agent keeps using its last best policy. This guarantees a fair comparison between agents with different computational costs. The performance indicators are computed every 100 time steps. Each interaction between an agent and an MDP is repeated for 20 seeds. The per-step normalized cumulative regret (defined in App. C) is employed as a performance measure since it provides a unified scale across different MDPs.

**Benchmark hardness.** In order to illustrate how hardness measures relate to cumulative regret in the benchmark, Figures 18a, 18b, 18c, and 18d place the average cumulative regret obtained by each agent in each benchmark MDP in a coordinate that corresponds to the diameter and the environmental value norm of that MDP. In the episodic setting (Figures 18a and 18b), we note that the environmental value norm has an evident impact on the Q-learning agent, whereas the effect of the diameter is most noticeable in the communicating case. Still, in the episodic setting, the diameter has a small influence compared to the environmental value norm for PSRL. In the continuous setting (Figures 18c and 18d), there is generally a positive relationship between both of these hardness measures and the average cumulative regret for UCRL2. For Q-learning and PSRL, the diameter seems to have a generally smaller influence on the average cumulative regret.

**Cumulative regret plots.** Figures 19, 20, 21, and 22 report the expected cumulative regrets for the agents during the agent/MDP interactions along with the cumulative regret of an agent that selects action at random, which provides an informative baseline. Contrary to the episodic setting, in the continuous setting, the training of UCRL2 and PSRL is stopped for several MDPs of the benchmark. For PSRL, this typically happens before reaching $10\,000$ time steps, which is particularly damaging. At this point, the agent has not properly explored the MDP and so it is forced to continue the interaction following a policy that yields a regret similar to the one of the random agent. UCRL2, instead, tends to terminate the allocated training time at later time steps, which penalizes the performance less.

**Cumulative regret tables.** In Tables 11, 12, 13, and 14, we report the per-step normalized regrets with standard deviations along with the number of seeds for which the agent has been able to complete the total number of training time steps before exceeding the time limit. We highlight in bold the best performing agent for each MDP. The same information has been summarized in Table 2 (Section 3.3).

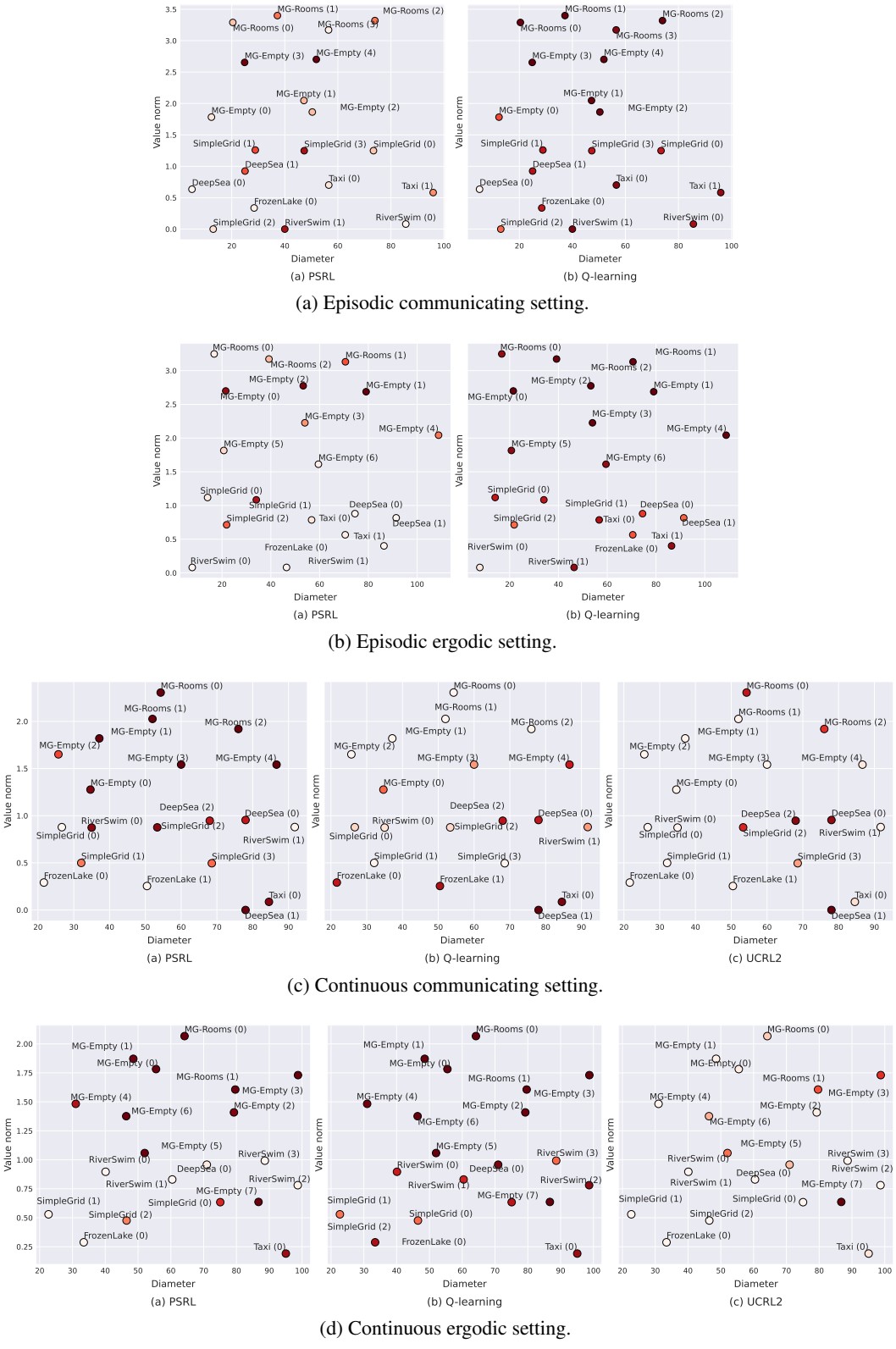

Figure 18: Average cumulative regret obtained by the agents in the continuous ergodic setting placed according to the diameter and the value norm values of the benchmark MDPs.

Table 10: Summary of benchmark results for the non-tabular `bsuite` baselines.

(a) Episodic communicating setting.

|  | PSRL | Q-learning |
|---|---|---|
| DeepSea | **0.00** ± 0.00 | 0.01 ± 0.01 |
|  | **0.54** ± 0.01 | 0.83 ± 0.02 |
| FrozenLake | **0.03** ± 0.11 | 0.78 ± 0.04 |
| MG-Empty | **0.09** ± 0.05 | 0.59 ± 0.07 |
|  | **0.24** ± 0.15 | 0.99 ± 0.00 |
|  | **0.23** ± 0.12 | 0.99 ± 0.01 |
|  | **0.91** ± 0.09 | 1.00 ± 0.00 |
|  | **0.93** ± 0.09 | 1.00 ± 0.00 |
| MG-Rooms | **0.21** ± 0.29 | 0.99 ± 0.01 |
|  | **0.44** ± 0.39 | 1.00 ± 0.00 |
|  | **0.43** ± 0.39 | 1.00 ± 0.00 |
|  | **0.04** ± 0.04 | 0.94 ± 0.05 |
| RiverSwim | **0.00** ± 0.00 | 0.87 ± 0.00 |
|  | **0.80** ± 0.00 | 0.96 ± 0.01 |
| SimpleGrid | **0.20** ± 0.15 | 0.78 ± 0.10 |
|  | **0.55** ± 0.15 | 0.80 ± 0.00 |
|  | **0.11** ± 0.01 | 0.50 ± 0.00 |
|  | **0.79** ± 0.04 | **0.79** ± 0.04 |
| Taxi | **0.09** ± 0.01 | 0.94 ± 0.00 |
|  | **0.36** ± 0.06 | 0.91 ± 0.01 |
| *Average* | **0.35** ± 0.30 | 0.83 ± 0.23 |

(b) Continuous communicating setting.

|  | PSRL | Q-learning | UCRL2 |
|---|---|---|---|
| DeepSea | **0.78** ± 0.05 | **0.78** ± 0.00 | 0.90 ± 0.01 |
|  | **0.99** ± 0.00 | **0.99** ± 0.00 | **0.99** ± 0.00 |
|  | **0.79** ± 0.04 | **0.79** ± 0.00 | 0.92 ± 0.01 |
| FrozenLake | **0.01** ± 0.04 | 0.77 ± 0.04 | **0.01** ± 0.01 |
|  | **0.01** ± 0.02 | 0.84 ± 0.04 | 0.04 ± 0.06 |
| MG-Empty | 0.95 ± 0.22 | 0.51 ± 0.23 | **0.02** ± 0.00 |
|  | 1.00 ± 0.00 | **0.01** ± 0.00 | 0.02 ± 0.00 |
|  | 0.60 ± 0.50 | **0.00** ± 0.00 | 0.01 ± 0.00 |
|  | 1.00 ± 0.00 | 0.35 ± 0.17 | **0.01** ± 0.00 |
|  | 1.00 ± 0.00 | 0.75 ± 0.21 | **0.08** ± 0.20 |
| MG-Rooms | 1.00 ± 0.00 | **0.01** ± 0.01 | 0.78 ± 0.40 |
|  | 1.00 ± 0.00 | **0.01** ± 0.01 | 0.02 ± 0.01 |
|  | 1.00 ± 0.00 | **0.02** ± 0.02 | 0.66 ± 0.47 |
| RiverSwim | **0.00** ± 0.01 | 0.16 ± 0.03 | **0.00** ± 0.00 |
|  | **0.01** ± 0.00 | 0.34 ± 0.14 | 0.02 ± 0.01 |
| SimpleGrid | 0.93 ± 0.00 | 0.11 ± 0.01 | **0.01** ± 0.00 |
|  | 0.45 ± 0.15 | **0.01** ± 0.00 | **0.01** ± 0.00 |
|  | 0.93 ± 0.00 | **0.15** ± 0.01 | 0.70 ± 0.40 |
|  | 0.50 ± 0.00 | **0.01** ± 0.00 | 0.33 ± 0.24 |
| Taxi | 0.94 ± 0.04 | 0.95 ± 0.00 | **0.12** ± 0.01 |
| *Average* | 0.69 ± 0.38 | 0.38 ± 0.37 | **0.28** ± 0.37 |

(c) Episodic ergodic setting.

|  | PSRL | Q-learning |
|---|---|---|
| DeepSea | **0.01** ± 0.00 | 0.64 ± 0.00 |
|  | **0.00** ± 0.00 | 0.52 ± 0.01 |
| FrozenLake | **0.01** ± 0.00 | 0.90 ± 0.01 |
| MG-Empty | **0.86** ± 0.16 | 1.00 ± 0.00 |
|  | **0.94** ± 0.07 | 1.00 ± 0.00 |
|  | **0.91** ± 0.09 | 1.00 ± 0.00 |
|  | **0.35** ± 0.10 | 1.00 ± 0.00 |
|  | **0.44** ± 0.12 | 1.00 ± 0.00 |
|  | **0.14** ± 0.08 | 0.92 ± 0.04 |
|  | **0.04** ± 0.03 | 0.91 ± 0.03 |
| MG-Rooms | **0.05** ± 0.04 | 0.90 ± 0.04 |
|  | **0.54** ± 0.36 | 1.00 ± 0.00 |
|  | **0.24** ± 0.29 | 0.99 ± 0.01 |
| RiverSwim | **0.00** ± 0.00 | 0.07 ± 0.02 |
|  | **0.00** ± 0.00 | 0.91 ± 0.01 |
| SimpleGrid | **0.05** ± 0.01 | 0.78 ± 0.03 |
|  | **0.79** ± 0.03 | **0.79** ± 0.03 |
|  | **0.50** ± 0.03 | **0.50** ± 0.03 |
| Taxi | **0.08** ± 0.01 | 0.84 ± 0.01 |
|  | **0.05** ± 0.00 | 0.56 ± 0.02 |
| *Average* | **0.30** ± 0.33 | 0.81 ± 0.24 |

(d) Continuous ergodic setting.

|  | PSRL | Q-learning | UCRL2 |
|---|---|---|---|
| DeepSea | **0.06** ± 0.01 | 0.94 ± 0.00 | 0.23 ± 0.05 |
| FrozenLake | **0.01** ± 0.03 | 0.83 ± 0.03 | **0.01** ± 0.02 |
| MG-Empty | 0.99 ± 0.01 | 0.98 ± 0.02 | **0.05** ± 0.06 |
|  | 0.98 ± 0.04 | 0.98 ± 0.02 | **0.03** ± 0.05 |
|  | 0.95 ± 0.03 | 0.97 ± 0.00 | **0.04** ± 0.01 |
|  | 0.99 ± 0.01 | 0.98 ± 0.01 | **0.54** ± 0.26 |
|  | 0.83 ± 0.31 | 0.96 ± 0.01 | **0.01** ± 0.00 |
|  | 0.99 ± 0.02 | 0.98 ± 0.02 | **0.45** ± 0.35 |
|  | 0.99 ± 0.01 | 0.98 ± 0.03 | **0.27** ± 0.33 |
|  | 0.99 ± 0.01 | 0.98 ± 0.01 | **0.93** ± 0.09 |
| MG-Rooms | 0.99 ± 0.02 | 0.98 ± 0.03 | **0.18** ± 0.29 |
|  | 1.00 ± 0.00 | 0.98 ± 0.02 | **0.62** ± 0.36 |
| RiverSwim | **0.00** ± 0.00 | 0.73 ± 0.19 | **0.00** ± 0.00 |
|  | **0.00** ± 0.00 | 0.71 ± 0.22 | 0.01 ± 0.00 |
|  | 0.02 ± 0.04 | 0.90 ± 0.06 | **0.01** ± 0.01 |
|  | **0.01** ± 0.00 | 0.50 ± 0.25 | **0.01** ± 0.01 |
| SimpleGrid | 0.70 ± 0.19 | 0.78 ± 0.00 | **0.01** ± 0.01 |
|  | 0.01 ± 0.02 | 0.46 ± 0.08 | **0.00** ± 0.00 |
|  | 0.43 ± 0.16 | 0.49 ± 0.00 | **0.00** ± 0.00 |
| Taxi | 0.89 ± 0.08 | 0.87 ± 0.01 | **0.09** ± 0.01 |
| *Average* | 0.59 ± 0.44 | 0.85 ± 0.18 | **0.17** ± 0.26 |

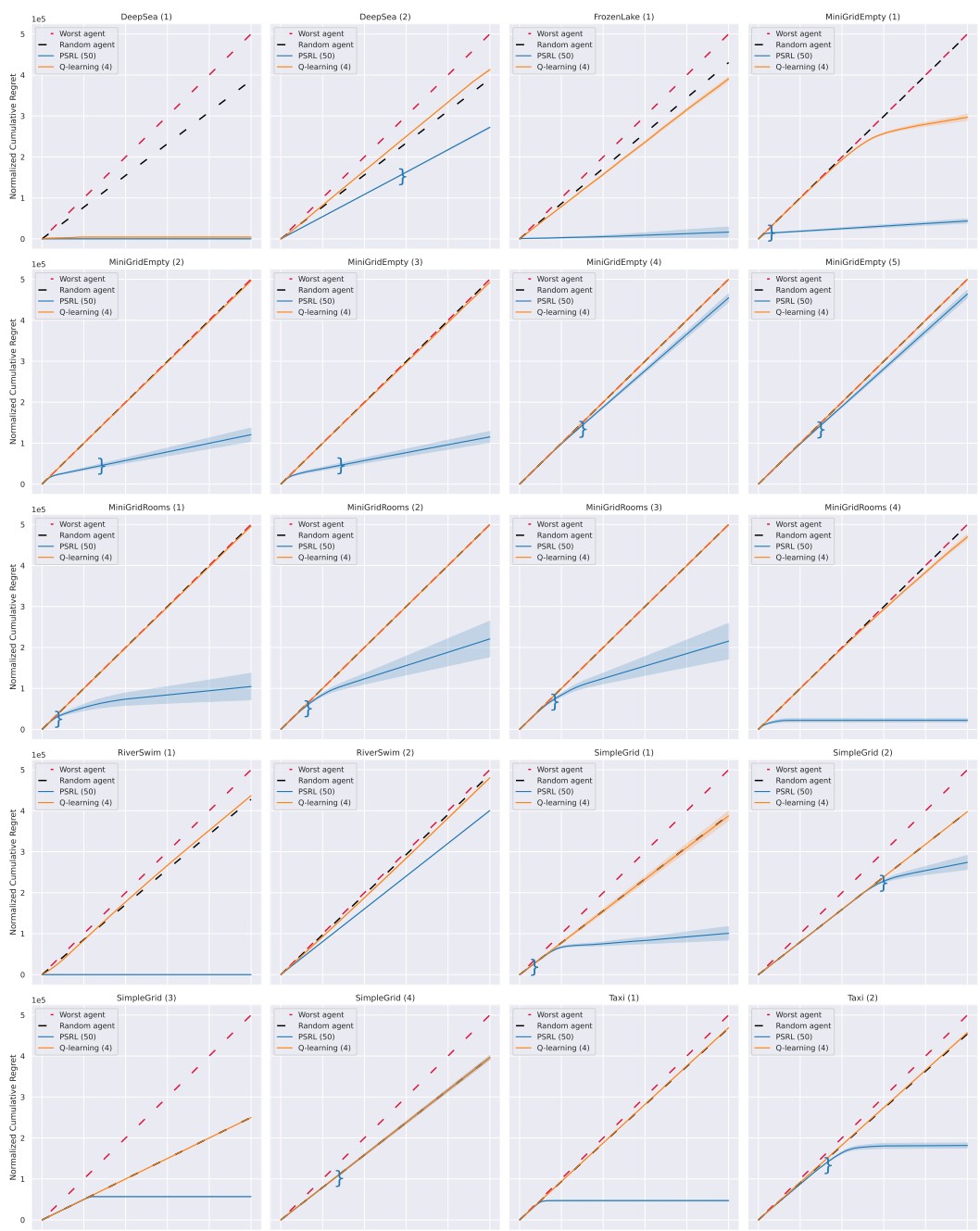

Figure 19: Episodic communicating benchmark results.

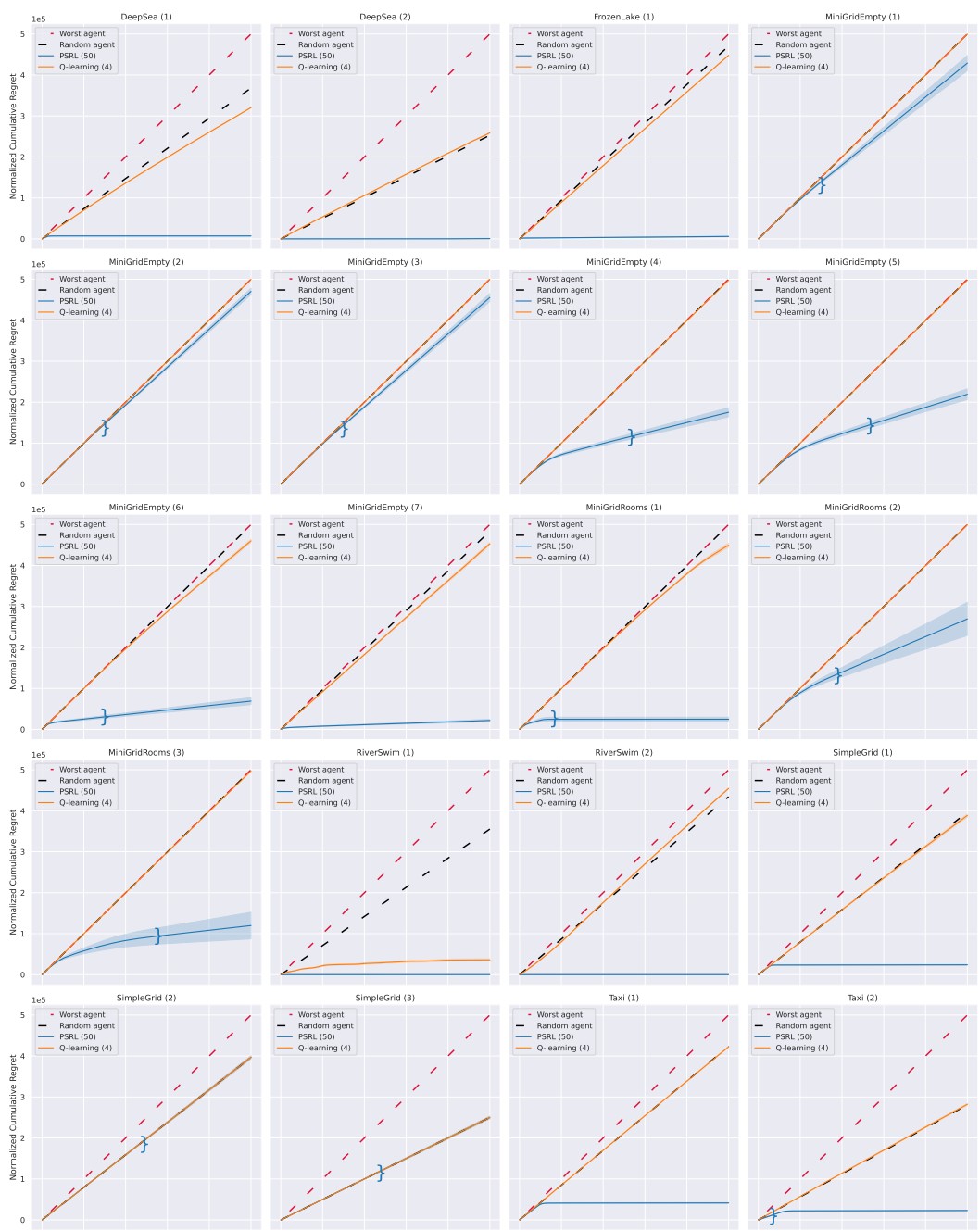

Figure 20: Episodic ergodic benchmark results.

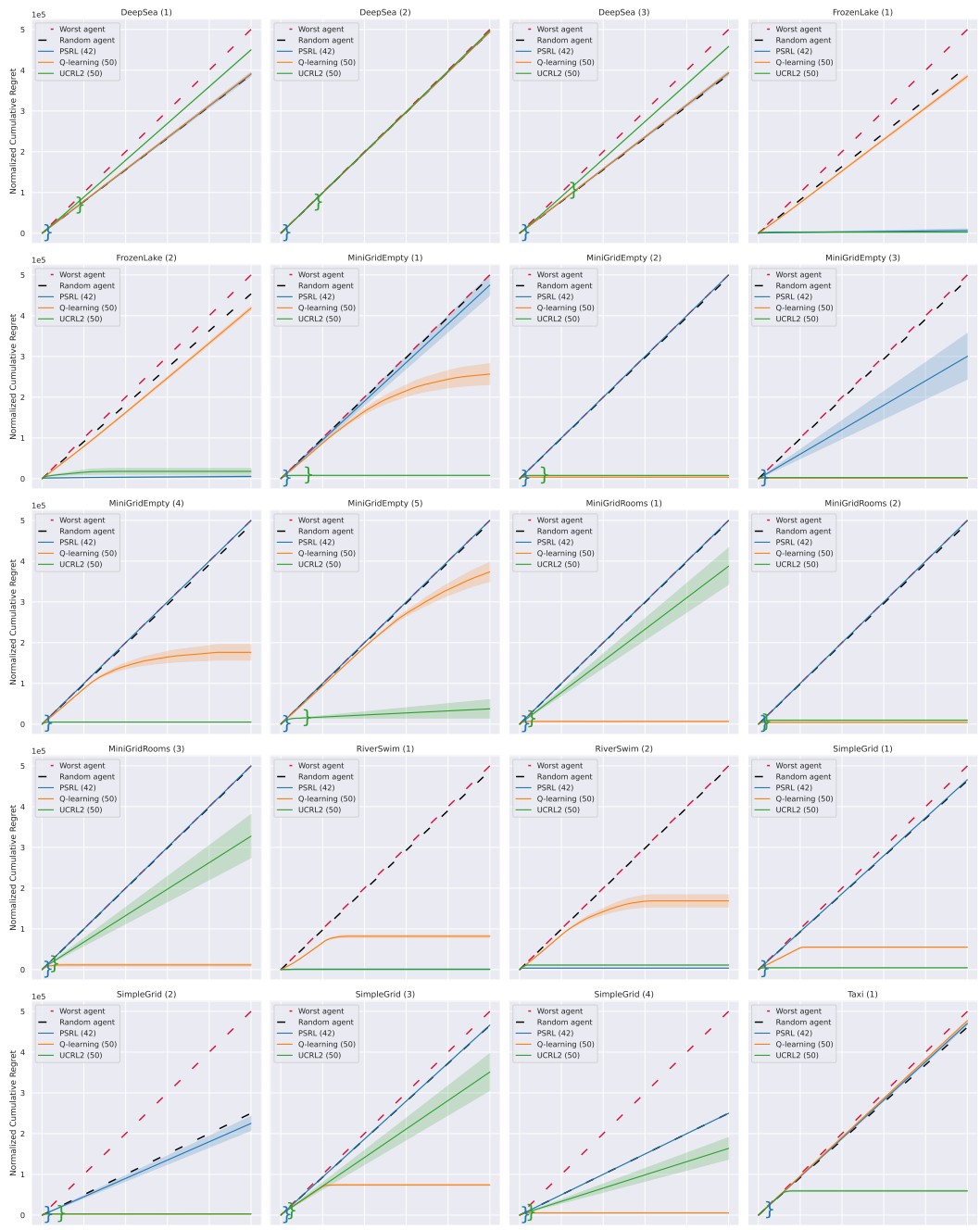

Figure 21: Continuous communicating benchmark results.

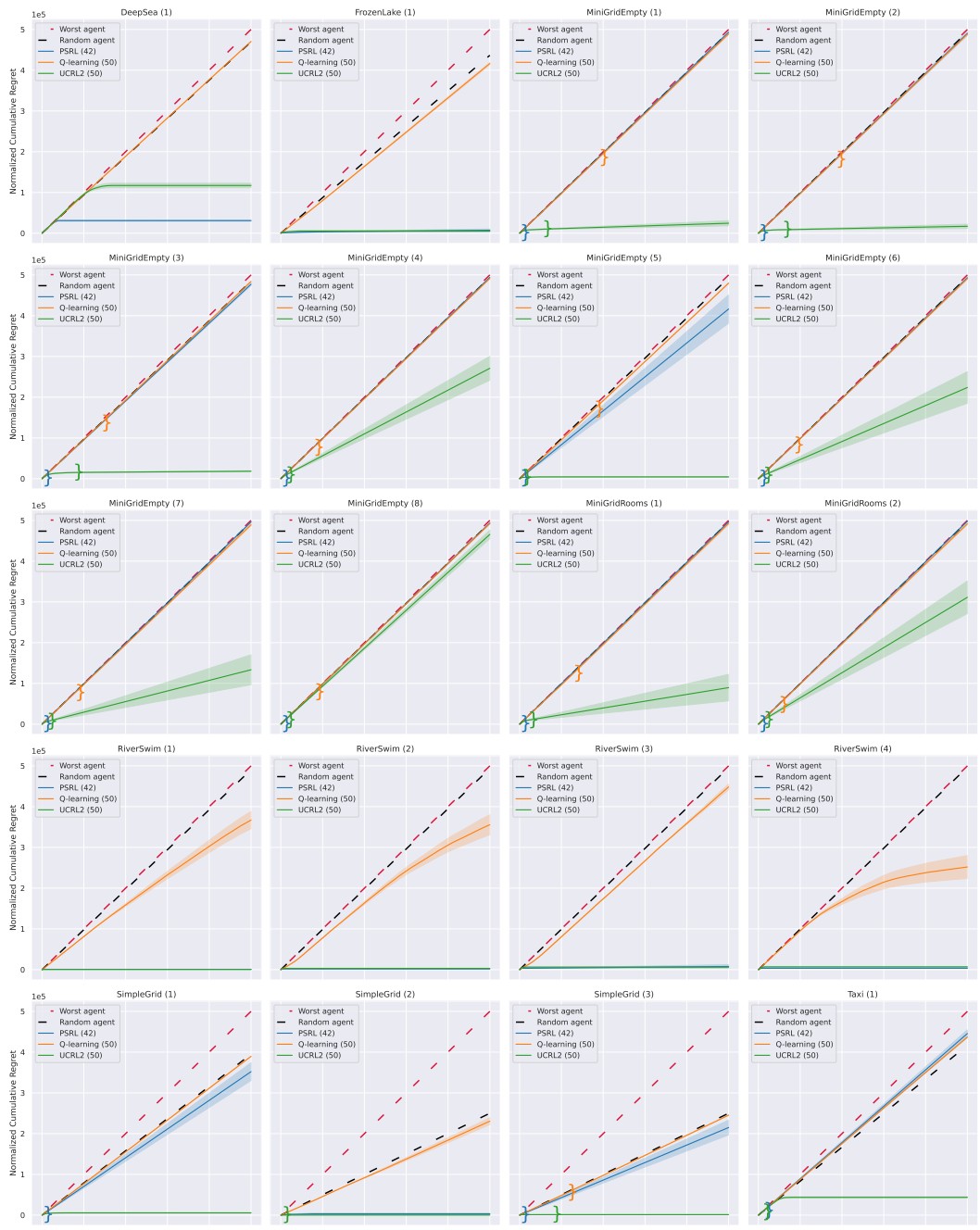

Figure 22: Continuous ergodic benchmark results.

Table 11: Episodic communicating benchmark final time step performance indicators.

| MDP | Agent | Norm. cumulative reward | Norm. cumulative expected reward | Norm. cumulative regret | Steps per second | # completed seeds |
|---|---|---|---|---|---|---|
| DeepSea (1) | PSRL | $1.00 \pm 0.00$ | $1.00 \pm 0.00$ | $0.00 \pm 0.00$ | $0.02 \pm 0.00$ | 20/20 |
| | Q-learning | $0.99 \pm 0.01$ | $0.99 \pm 0.01$ | $0.01 \pm 0.01$ | $0.03 \pm 0.01$ | 20/20 |
| DeepSea (2) | PSRL | $0.23 \pm 0.06$ | $0.46 \pm 0.01$ | $0.54 \pm 0.01$ | $0.00 \pm 0.00$ | 0/20 |
| | Q-learning | $0.17 \pm 0.02$ | $0.17 \pm 0.02$ | $0.83 \pm 0.02$ | $0.01 \pm 0.00$ | 20/20 |
| FrozenLake (1) | PSRL | $0.92 \pm 0.15$ | $0.97 \pm 0.11$ | $0.03 \pm 0.11$ | $0.01 \pm 0.01$ | 20/20 |
| | Q-learning | $0.22 \pm 0.04$ | $0.22 \pm 0.04$ | $0.78 \pm 0.04$ | $0.01 \pm 0.00$ | 20/20 |
| MG-Empty (1) | PSRL | $0.88 \pm 0.05$ | $0.92 \pm 0.04$ | $0.09 \pm 0.04$ | $0.00 \pm 0.00$ | 19/20 |
| | Q-learning | $0.43 \pm 0.06$ | $0.43 \pm 0.06$ | $0.59 \pm 0.07$ | $0.01 \pm 0.00$ | 20/20 |
| MG-Empty (2) | PSRL | $0.72 \pm 0.14$ | $0.83 \pm 0.08$ | $0.24 \pm 0.14$ | $0.00 \pm 0.00$ | 14/20 |
| | Q-learning | $0.01 \pm 0.01$ | $0.01 \pm 0.01$ | $0.99 \pm 0.00$ | $0.01 \pm 0.00$ | 20/20 |
| MG-Empty (3) | PSRL | $0.75 \pm 0.09$ | $0.80 \pm 0.10$ | $0.23 \pm 0.12$ | $0.00 \pm 0.00$ | 14/20 |
| | Q-learning | $0.02 \pm 0.01$ | $0.02 \pm 0.01$ | $0.99 \pm 0.00$ | $0.01 \pm 0.00$ | 20/20 |
| MG-Empty (4) | PSRL | $0.00 \pm 0.00$ | $0.12 \pm 0.12$ | $0.91 \pm 0.09$ | $0.00 \pm 0.00$ | 0/20 |
| | Q-learning | $0.00 \pm 0.00$ | $0.00 \pm 0.00$ | $1.00 \pm 0.00$ | $0.01 \pm 0.00$ | 20/20 |
| MG-Empty (5) | PSRL | $0.00 \pm 0.00$ | $0.10 \pm 0.13$ | $0.93 \pm 0.08$ | $0.00 \pm 0.00$ | 0/20 |
| | Q-learning | $0.00 \pm 0.00$ | $0.00 \pm 0.00$ | $1.00 \pm 0.00$ | $0.01 \pm 0.00$ | 20/20 |
| MG-Rooms (1) | PSRL | $0.22 \pm 0.33$ | $0.79 \pm 0.28$ | $0.21 \pm 0.28$ | $0.00 \pm 0.00$ | 18/20 |
| | Q-learning | $0.01 \pm 0.01$ | $0.01 \pm 0.01$ | $0.99 \pm 0.01$ | $0.01 \pm 0.00$ | 20/20 |
| MG-Rooms (2) | PSRL | $0.00 \pm 0.00$ | $0.56 \pm 0.38$ | $0.44 \pm 0.38$ | $0.00 \pm 0.00$ | 14/20 |
| | Q-learning | $0.00 \pm 0.00$ | $0.00 \pm 0.00$ | $1.00 \pm 0.00$ | $0.01 \pm 0.00$ | 20/20 |
| MG-Rooms (3) | PSRL | $0.00 \pm 0.00$ | $0.57 \pm 0.38$ | $0.43 \pm 0.38$ | $0.00 \pm 0.00$ | 12/20 |
| | Q-learning | $0.00 \pm 0.00$ | $0.00 \pm 0.00$ | $1.00 \pm 0.00$ | $0.01 \pm 0.00$ | 20/20 |
| MG-Rooms (4) | PSRL | $0.62 \pm 0.34$ | $0.96 \pm 0.04$ | $0.04 \pm 0.04$ | $0.01 \pm 0.00$ | 20/20 |
| | Q-learning | $0.07 \pm 0.05$ | $0.07 \pm 0.05$ | $0.94 \pm 0.05$ | $0.01 \pm 0.00$ | 20/20 |
| RiverSwim (1) | PSRL | $0.94 \pm 0.17$ | $1.00 \pm 0.00$ | $0.00 \pm 0.00$ | $0.02 \pm 0.00$ | 20/20 |
| | Q-learning | $0.13 \pm 0.00$ | $0.13 \pm 0.00$ | $0.87 \pm 0.00$ | $0.01 \pm 0.00$ | 20/20 |
| RiverSwim (2) | PSRL | $0.17 \pm 0.00$ | $0.20 \pm 0.00$ | $0.80 \pm 0.00$ | $0.01 \pm 0.00$ | 20/20 |
| | Q-learning | $0.04 \pm 0.01$ | $0.04 \pm 0.01$ | $0.96 \pm 0.01$ | $0.01 \pm 0.00$ | 20/20 |
| SimpleGrid (1) | PSRL | $0.63 \pm 0.18$ | $0.80 \pm 0.14$ | $0.20 \pm 0.14$ | $0.00 \pm 0.00$ | 19/20 |
| | Q-learning | $0.22 \pm 0.10$ | $0.22 \pm 0.10$ | $0.78 \pm 0.10$ | $0.01 \pm 0.00$ | 20/20 |
| SimpleGrid (2) | PSRL | $0.37 \pm 0.15$ | $0.45 \pm 0.15$ | $0.55 \pm 0.15$ | $0.00 \pm 0.00$ | 4/20 |
| | Q-learning | $0.21 \pm 0.00$ | $0.20 \pm 0.00$ | $0.80 \pm 0.00$ | $0.01 \pm 0.00$ | 20/20 |
| SimpleGrid (3) | PSRL | $0.51 \pm 0.03$ | $0.89 \pm 0.01$ | $0.11 \pm 0.01$ | $0.00 \pm 0.00$ | 20/20 |
| | Q-learning | $0.50 \pm 0.00$ | $0.50 \pm 0.00$ | $0.50 \pm 0.00$ | $0.01 \pm 0.00$ | 20/20 |
| SimpleGrid (4) | PSRL | $0.21 \pm 0.04$ | $0.21 \pm 0.04$ | $0.79 \pm 0.04$ | $0.00 \pm 0.00$ | 0/20 |
| | Q-learning | $0.21 \pm 0.04$ | $0.21 \pm 0.04$ | $0.79 \pm 0.04$ | $0.00 \pm 0.00$ | 20/20 |
| Taxi (1) | PSRL | $0.88 \pm 0.01$ | $0.91 \pm 0.01$ | $0.09 \pm 0.01$ | $0.01 \pm 0.00$ | 20/20 |
| | Q-learning | $0.06 \pm 0.00$ | $0.06 \pm 0.00$ | $0.94 \pm 0.00$ | $0.01 \pm 0.00$ | 20/20 |
| Taxi (2) | PSRL | $0.51 \pm 0.11$ | $0.64 \pm 0.06$ | $0.36 \pm 0.05$ | $0.00 \pm 0.00$ | 13/20 |
| | Q-learning | $0.09 \pm 0.01$ | $0.09 \pm 0.01$ | $0.91 \pm 0.01$ | $0.01 \pm 0.00$ | 20/20 |

Table 12: Episodic ergodic benchmark final time step performance indicators.

| MDP | Agent | Norm. cumulative reward | Norm. cumulative expected reward | Norm. cumulative regret | Steps per second | # completed seeds |
|---|---|---|---|---|---|---|
| DeepSea (1) | PSRL | $0.69 \pm 0.17$ | $0.99 \pm 0.00$ | $0.01 \pm 0.00$ | $0.01 \pm 0.00$ | 20/20 |
| | Q-learning | $0.36 \pm 0.00$ | $0.36 \pm 0.00$ | $0.64 \pm 0.00$ | $0.01 \pm 0.00$ | 20/20 |
| DeepSea (2) | PSRL | $0.79 \pm 0.07$ | $1.00 \pm 0.00$ | $0.00 \pm 0.00$ | $0.01 \pm 0.00$ | 20/20 |
| | Q-learning | $0.48 \pm 0.01$ | $0.48 \pm 0.01$ | $0.52 \pm 0.01$ | $0.01 \pm 0.00$ | 20/20 |
| FrozenLake (1) | PSRL | $0.95 \pm 0.08$ | $0.99 \pm 0.00$ | $0.01 \pm 0.00$ | $0.01 \pm 0.00$ | 20/20 |
| | Q-learning | $0.10 \pm 0.01$ | $0.10 \pm 0.01$ | $0.90 \pm 0.01$ | $0.01 \pm 0.00$ | 20/20 |
| MG-Empty (1) | PSRL | $0.00 \pm 0.00$ | $0.18 \pm 0.20$ | $0.86 \pm 0.16$ | $0.00 \pm 0.00$ | 0/20 |
| | Q-learning | $0.00 \pm 0.00$ | $0.00 \pm 0.00$ | $1.00 \pm 0.00$ | $0.01 \pm 0.00$ | 20/20 |
| MG-Empty (2) | PSRL | $0.00 \pm 0.00$ | $0.08 \pm 0.10$ | $0.94 \pm 0.07$ | $0.00 \pm 0.00$ | 0/20 |
| | Q-learning | $0.00 \pm 0.00$ | $0.00 \pm 0.00$ | $1.00 \pm 0.00$ | $0.01 \pm 0.00$ | 20/20 |
| MG-Empty (3) | PSRL | $0.00 \pm 0.00$ | $0.12 \pm 0.12$ | $0.91 \pm 0.09$ | $0.00 \pm 0.00$ | 0/20 |
| | Q-learning | $0.00 \pm 0.00$ | $0.00 \pm 0.00$ | $1.00 \pm 0.00$ | $0.01 \pm 0.00$ | 20/20 |
| MG-Empty (4) | PSRL | $0.60 \pm 0.08$ | $0.69 \pm 0.09$ | $0.35 \pm 0.10$ | $0.00 \pm 0.00$ | 0/20 |
| | Q-learning | $0.01 \pm 0.00$ | $0.01 \pm 0.00$ | $1.00 \pm 0.00$ | $0.01 \pm 0.00$ | 20/20 |
| MG-Empty (5) | PSRL | $0.52 \pm 0.10$ | $0.62 \pm 0.11$ | $0.44 \pm 0.11$ | $0.00 \pm 0.00$ | 0/20 |
| | Q-learning | $0.01 \pm 0.01$ | $0.01 \pm 0.00$ | $1.00 \pm 0.00$ | $0.01 \pm 0.00$ | 20/20 |
| MG-Empty (6) | PSRL | $0.84 \pm 0.06$ | $0.87 \pm 0.06$ | $0.14 \pm 0.07$ | $0.00 \pm 0.00$ | 14/20 |
| | Q-learning | $0.10 \pm 0.05$ | $0.10 \pm 0.05$ | $0.92 \pm 0.03$ | $0.01 \pm 0.00$ | 20/20 |
| MG-Empty (7) | PSRL | $0.95 \pm 0.02$ | $0.97 \pm 0.02$ | $0.04 \pm 0.03$ | $0.00 \pm 0.00$ | 20/20 |
| | Q-learning | $0.13 \pm 0.05$ | $0.13 \pm 0.05$ | $0.91 \pm 0.03$ | $0.01 \pm 0.00$ | 20/20 |
| MG-Rooms (1) | PSRL | $0.80 \pm 0.03$ | $0.95 \pm 0.04$ | $0.05 \pm 0.04$ | $0.00 \pm 0.00$ | 17/20 |
| | Q-learning | $0.10 \pm 0.04$ | $0.10 \pm 0.04$ | $0.90 \pm 0.04$ | $0.01 \pm 0.00$ | 20/20 |
| MG-Rooms (2) | PSRL | $0.00 \pm 0.00$ | $0.47 \pm 0.35$ | $0.54 \pm 0.35$ | $0.00 \pm 0.00$ | 0/20 |
| | Q-learning | $0.00 \pm 0.00$ | $0.00 \pm 0.00$ | $1.00 \pm 0.00$ | $0.01 \pm 0.00$ | 20/20 |
| MG-Rooms (3) | PSRL | $0.39 \pm 0.25$ | $0.76 \pm 0.28$ | $0.24 \pm 0.28$ | $0.00 \pm 0.00$ | 0/20 |
| | Q-learning | $0.01 \pm 0.01$ | $0.01 \pm 0.01$ | $0.99 \pm 0.01$ | $0.01 \pm 0.00$ | 20/20 |
| RiverSwim (1) | PSRL | $0.98 \pm 0.07$ | $1.00 \pm 0.00$ | $0.00 \pm 0.00$ | $0.02 \pm 0.00$ | 20/20 |
| | Q-learning | $0.93 \pm 0.02$ | $0.93 \pm 0.02$ | $0.07 \pm 0.02$ | $0.01 \pm 0.00$ | 20/20 |
| RiverSwim (2) | PSRL | $0.93 \pm 0.17$ | $1.00 \pm 0.00$ | $0.00 \pm 0.00$ | $0.02 \pm 0.00$ | 20/20 |
| | Q-learning | $0.09 \pm 0.01$ | $0.09 \pm 0.01$ | $0.91 \pm 0.01$ | $0.01 \pm 0.00$ | 20/20 |
| SimpleGrid (1) | PSRL | $0.93 \pm 0.01$ | $0.95 \pm 0.01$ | $0.05 \pm 0.01$ | $0.00 \pm 0.00$ | 20/20 |
| | Q-learning | $0.22 \pm 0.03$ | $0.22 \pm 0.03$ | $0.78 \pm 0.03$ | $0.01 \pm 0.00$ | 20/20 |
| SimpleGrid (2) | PSRL | $0.21 \pm 0.03$ | $0.21 \pm 0.03$ | $0.79 \pm 0.03$ | $0.00 \pm 0.00$ | 0/20 |
| | Q-learning | $0.21 \pm 0.03$ | $0.21 \pm 0.03$ | $0.79 \pm 0.03$ | $0.01 \pm 0.00$ | 20/20 |
| SimpleGrid (3) | PSRL | $0.51 \pm 0.04$ | $0.50 \pm 0.03$ | $0.50 \pm 0.03$ | $0.00 \pm 0.00$ | 0/20 |
| | Q-learning | $0.50 \pm 0.03$ | $0.50 \pm 0.03$ | $0.50 \pm 0.03$ | $0.01 \pm 0.00$ | 20/20 |
| Taxi (1) | PSRL | $0.89 \pm 0.01$ | $0.92 \pm 0.01$ | $0.08 \pm 0.01$ | $0.01 \pm 0.00$ | 20/20 |
| | Q-learning | $0.15 \pm 0.01$ | $0.15 \pm 0.01$ | $0.84 \pm 0.01$ | $0.01 \pm 0.00$ | 20/20 |
| Taxi (2) | PSRL | $0.93 \pm 0.01$ | $0.95 \pm 0.00$ | $0.05 \pm 0.00$ | $0.00 \pm 0.00$ | 19/20 |
| | Q-learning | $0.44 \pm 0.02$ | $0.44 \pm 0.02$ | $0.56 \pm 0.02$ | $0.01 \pm 0.00$ | 20/20 |

Table 13: Continuous communicating benchmark final time step performance indicators.

| MDP | Agent | Norm. cumulative reward | Norm. cumulative expected reward | Norm. cumulative regret | Steps per second | # completed seeds |
|---|---|---|---|---|---|---|
| DeepSea (1) | PSRL | $0.22 \pm 0.01$ | $0.22 \pm 0.05$ | $0.78 \pm 0.05$ | $0.00 \pm 0.00$ | 0/20 |
| | Q-learning | $0.22 \pm 0.00$ | $0.22 \pm 0.00$ | $0.78 \pm 0.00$ | $0.00 \pm 0.00$ | 20/20 |
| | UCRL2 | $0.19 \pm 0.03$ | $0.10 \pm 0.01$ | $0.90 \pm 0.01$ | $0.00 \pm 0.00$ | 0/20 |
| DeepSea (2) | PSRL | $0.01 \pm 0.00$ | $0.01 \pm 0.00$ | $0.99 \pm 0.00$ | $0.00 \pm 0.00$ | 0/20 |
| | Q-learning | $0.01 \pm 0.00$ | $0.01 \pm 0.00$ | $0.99 \pm 0.00$ | $0.00 \pm 0.00$ | 20/20 |
| | UCRL2 | $0.01 \pm 0.00$ | $0.01 \pm 0.00$ | $0.99 \pm 0.00$ | $0.00 \pm 0.00$ | 0/20 |
| DeepSea (3) | PSRL | $0.22 \pm 0.01$ | $0.21 \pm 0.04$ | $0.79 \pm 0.04$ | $0.00 \pm 0.00$ | 0/20 |
| | Q-learning | $0.21 \pm 0.00$ | $0.21 \pm 0.00$ | $0.79 \pm 0.00$ | $0.00 \pm 0.00$ | 20/20 |
| | UCRL2 | $0.17 \pm 0.03$ | $0.08 \pm 0.01$ | $0.92 \pm 0.01$ | $0.00 \pm 0.00$ | 0/20 |
| FrozenLake (1) | PSRL | $0.19 \pm 0.20$ | $0.99 \pm 0.03$ | $0.01 \pm 0.03$ | $0.02 \pm 0.00$ | 20/20 |
| | Q-learning | $0.23 \pm 0.04$ | $0.23 \pm 0.04$ | $0.77 \pm 0.04$ | $0.01 \pm 0.00$ | 20/20 |
| | UCRL2 | $0.68 \pm 0.17$ | $0.99 \pm 0.01$ | $0.01 \pm 0.01$ | $0.02 \pm 0.01$ | 20/20 |
| FrozenLake (2) | PSRL | $0.28 \pm 0.29$ | $0.99 \pm 0.02$ | $0.01 \pm 0.02$ | $0.01 \pm 0.00$ | 20/20 |
| | Q-learning | $0.14 \pm 0.04$ | $0.16 \pm 0.04$ | $0.84 \pm 0.04$ | $0.01 \pm 0.00$ | 20/20 |
| | UCRL2 | $0.34 \pm 0.20$ | $0.96 \pm 0.07$ | $0.04 \pm 0.06$ | $0.02 \pm 0.01$ | 20/20 |
| MG-Empty (1) | PSRL | $0.05 \pm 0.22$ | $0.05 \pm 0.22$ | $0.95 \pm 0.22$ | $0.00 \pm 0.00$ | 0/20 |
| | Q-learning | $0.49 \pm 0.22$ | $0.49 \pm 0.22$ | $0.51 \pm 0.22$ | $0.01 \pm 0.00$ | 20/20 |
| | UCRL2 | $0.29 \pm 0.38$ | $0.98 \pm 0.00$ | $0.02 \pm 0.00$ | $0.00 \pm 0.00$ | 7/20 |
| MG-Empty (2) | PSRL | $0.00 \pm 0.00$ | $0.00 \pm 0.00$ | $1.00 \pm 0.00$ | $0.00 \pm 0.00$ | 0/20 |
| | Q-learning | $0.99 \pm 0.00$ | $0.99 \pm 0.00$ | $0.01 \pm 0.00$ | $0.01 \pm 0.00$ | 20/20 |
| | UCRL2 | $0.91 \pm 0.02$ | $0.98 \pm 0.00$ | $0.02 \pm 0.00$ | $0.00 \pm 0.00$ | 9/20 |
| MG-Empty (3) | PSRL | $0.00 \pm 0.00$ | $0.40 \pm 0.49$ | $0.60 \pm 0.49$ | $0.00 \pm 0.00$ | 0/20 |
| | Q-learning | $1.00 \pm 0.00$ | $1.00 \pm 0.00$ | $0.00 \pm 0.00$ | $0.02 \pm 0.01$ | 20/20 |
| | UCRL2 | $0.96 \pm 0.01$ | $0.99 \pm 0.00$ | $0.01 \pm 0.00$ | $0.01 \pm 0.00$ | 20/20 |
| MG-Empty (4) | PSRL | $0.00 \pm 0.00$ | $0.00 \pm 0.00$ | $1.00 \pm 0.00$ | $0.00 \pm 0.00$ | 0/20 |
| | Q-learning | $0.64 \pm 0.17$ | $0.65 \pm 0.17$ | $0.35 \pm 0.17$ | $0.02 \pm 0.01$ | 20/20 |
| | UCRL2 | $0.57 \pm 0.39$ | $0.99 \pm 0.00$ | $0.01 \pm 0.00$ | $0.01 \pm 0.00$ | 20/20 |
| MG-Empty (5) | PSRL | $0.00 \pm 0.00$ | $0.00 \pm 0.00$ | $1.00 \pm 0.00$ | $0.00 \pm 0.00$ | 0/20 |
| | Q-learning | $0.25 \pm 0.21$ | $0.25 \pm 0.20$ | $0.75 \pm 0.20$ | $0.01 \pm 0.00$ | 20/20 |
| | UCRL2 | $0.00 \pm 0.00$ | $0.92 \pm 0.20$ | $0.08 \pm 0.20$ | $0.00 \pm 0.00$ | 0/20 |
| MG-Rooms (1) | PSRL | $0.00 \pm 0.00$ | $0.00 \pm 0.00$ | $1.00 \pm 0.00$ | $0.00 \pm 0.00$ | 0/20 |
| | Q-learning | $0.99 \pm 0.01$ | $0.99 \pm 0.01$ | $0.01 \pm 0.01$ | $0.01 \pm 0.00$ | 20/20 |
| | UCRL2 | $0.00 \pm 0.00$ | $0.22 \pm 0.39$ | $0.78 \pm 0.39$ | $0.00 \pm 0.00$ | 0/20 |
| MG-Rooms (2) | PSRL | $0.00 \pm 0.00$ | $0.00 \pm 0.00$ | $1.00 \pm 0.00$ | $0.00 \pm 0.00$ | 0/20 |
| | Q-learning | $0.99 \pm 0.01$ | $0.99 \pm 0.01$ | $0.01 \pm 0.01$ | $0.02 \pm 0.01$ | 20/20 |
| | UCRL2 | $0.85 \pm 0.19$ | $0.98 \pm 0.01$ | $0.02 \pm 0.01$ | $0.00 \pm 0.00$ | 18/20 |
| MG-Rooms (3) | PSRL | $0.00 \pm 0.00$ | $0.00 \pm 0.00$ | $1.00 \pm 0.00$ | $0.00 \pm 0.00$ | 0/20 |
| | Q-learning | $0.98 \pm 0.02$ | $0.98 \pm 0.02$ | $0.02 \pm 0.02$ | $0.01 \pm 0.00$ | 20/20 |
| | UCRL2 | $0.00 \pm 0.00$ | $0.34 \pm 0.46$ | $0.66 \pm 0.46$ | $0.00 \pm 0.00$ | 0/20 |
| RiverSwim (1) | PSRL | $0.85 \pm 0.36$ | $1.00 \pm 0.01$ | $0.00 \pm 0.01$ | $0.01 \pm 0.00$ | 20/20 |
| | Q-learning | $0.83 \pm 0.03$ | $0.84 \pm 0.03$ | $0.16 \pm 0.03$ | $0.03 \pm 0.01$ | 20/20 |
| | UCRL2 | $0.99 \pm 0.00$ | $1.00 \pm 0.00$ | $0.00 \pm 0.00$ | $0.02 \pm 0.00$ | 20/20 |
| RiverSwim (2) | PSRL | $0.93 \pm 0.22$ | $0.99 \pm 0.00$ | $0.01 \pm 0.00$ | $0.01 \pm 0.00$ | 20/20 |
| | Q-learning | $0.66 \pm 0.13$ | $0.66 \pm 0.13$ | $0.34 \pm 0.13$ | $0.02 \pm 0.01$ | 20/20 |
| | UCRL2 | $0.95 \pm 0.02$ | $0.98 \pm 0.01$ | $0.02 \pm 0.01$ | $0.01 \pm 0.00$ | 20/20 |
| SimpleGrid (1) | PSRL | $0.07 \pm 0.00$ | $0.07 \pm 0.00$ | $0.93 \pm 0.00$ | $0.00 \pm 0.00$ | 0/20 |
| | Q-learning | $0.89 \pm 0.01$ | $0.89 \pm 0.01$ | $0.11 \pm 0.01$ | $0.02 \pm 0.00$ | 20/20 |
| | UCRL2 | $0.56 \pm 0.37$ | $0.99 \pm 0.00$ | $0.01 \pm 0.00$ | $0.00 \pm 0.00$ | 20/20 |
| SimpleGrid (2) | PSRL | $0.50 \pm 0.00$ | $0.55 \pm 0.15$ | $0.45 \pm 0.15$ | $0.00 \pm 0.00$ | 0/20 |
| | Q-learning | $0.99 \pm 0.00$ | $0.99 \pm 0.00$ | $0.01 \pm 0.00$ | $0.02 \pm 0.00$ | 20/20 |
| | UCRL2 | $0.61 \pm 0.18$ | $0.99 \pm 0.00$ | $0.01 \pm 0.00$ | $0.00 \pm 0.00$ | 13/20 |
| SimpleGrid (3) | PSRL | $0.07 \pm 0.00$ | $0.07 \pm 0.00$ | $0.93 \pm 0.00$ | $0.00 \pm 0.00$ | 0/20 |
| | Q-learning | $0.85 \pm 0.01$ | $0.85 \pm 0.01$ | $0.15 \pm 0.01$ | $0.01 \pm 0.00$ | 20/20 |
| | UCRL2 | $0.07 \pm 0.00$ | $0.30 \pm 0.39$ | $0.70 \pm 0.39$ | $0.00 \pm 0.00$ | 0/20 |
| SimpleGrid (4) | PSRL | $0.50 \pm 0.00$ | $0.50 \pm 0.00$ | $0.50 \pm 0.00$ | $0.00 \pm 0.00$ | 0/20 |
| | Q-learning | $0.99 \pm 0.00$ | $0.99 \pm 0.00$ | $0.01 \pm 0.00$ | $0.01 \pm 0.00$ | 20/20 |
| | UCRL2 | $0.50 \pm 0.00$ | $0.67 \pm 0.23$ | $0.33 \pm 0.23$ | $0.00 \pm 0.00$ | 0/20 |
| Taxi (1) | PSRL | $0.04 \pm 0.03$ | $0.06 \pm 0.03$ | $0.94 \pm 0.03$ | $0.00 \pm 0.00$ | 0/20 |
| | Q-learning | $0.10 \pm 0.01$ | $0.05 \pm 0.00$ | $0.95 \pm 0.00$ | $0.01 \pm 0.00$ | 20/20 |
| | UCRL2 | $0.06 \pm 0.03$ | $0.88 \pm 0.01$ | $0.12 \pm 0.01$ | $0.01 \pm 0.00$ | 20/20 |

Table 14: Continuous ergodic benchmark final time step performance indicators.

| MDP | Agent | Norm. cumulative reward | Norm. cumulative expected reward | Norm. cumulative regret | Steps per second | # completed seeds |
|---|---|---|---|---|---|---|
| DeepSea (1) | PSRL | $0.80 \pm 0.24$ | $0.93 \pm 0.01$ | $0.06 \pm 0.01$ | $0.00 \pm 0.00$ | 20/20 |
| | Q-learning | $0.06 \pm 0.00$ | $0.06 \pm 0.00$ | $0.94 \pm 0.00$ | $0.01 \pm 0.00$ | 20/20 |
| | UCRL2 | $0.04 \pm 0.01$ | $0.74 \pm 0.04$ | $0.23 \pm 0.05$ | $0.01 \pm 0.00$ | 20/20 |
| FrozenLake (1) | PSRL | $0.15 \pm 0.12$ | $0.98 \pm 0.03$ | $0.01 \pm 0.03$ | $0.01 \pm 0.00$ | 20/20 |
| | Q-learning | $0.16 \pm 0.03$ | $0.17 \pm 0.03$ | $0.83 \pm 0.03$ | $0.01 \pm 0.00$ | 20/20 |
| | UCRL2 | $0.41 \pm 0.15$ | $0.99 \pm 0.02$ | $0.01 \pm 0.02$ | $0.02 \pm 0.00$ | 20/20 |
| MG-Empty (1) | PSRL | $0.01 \pm 0.01$ | $0.01 \pm 0.01$ | $0.99 \pm 0.01$ | $0.00 \pm 0.00$ | 0/20 |
| | Q-learning | $0.02 \pm 0.01$ | $0.02 \pm 0.02$ | $0.98 \pm 0.02$ | $0.00 \pm 0.00$ | 0/20 |
| | UCRL2 | $0.04 \pm 0.02$ | $0.95 \pm 0.06$ | $0.05 \pm 0.06$ | $0.00 \pm 0.00$ | 0/20 |
| MG-Empty (2) | PSRL | $0.01 \pm 0.01$ | $0.02 \pm 0.04$ | $0.98 \pm 0.04$ | $0.00 \pm 0.00$ | 0/20 |
| | Q-learning | $0.02 \pm 0.02$ | $0.02 \pm 0.02$ | $0.98 \pm 0.02$ | $0.00 \pm 0.00$ | 0/20 |
| | UCRL2 | $0.03 \pm 0.03$ | $0.97 \pm 0.05$ | $0.03 \pm 0.05$ | $0.00 \pm 0.00$ | 0/20 |
| MG-Empty (3) | PSRL | $0.03 \pm 0.02$ | $0.05 \pm 0.03$ | $0.95 \pm 0.03$ | $0.00 \pm 0.00$ | 0/20 |
| | Q-learning | $0.03 \pm 0.00$ | $0.03 \pm 0.00$ | $0.97 \pm 0.00$ | $0.00 \pm 0.00$ | 14/20 |
| | UCRL2 | $0.07 \pm 0.02$ | $0.96 \pm 0.01$ | $0.04 \pm 0.01$ | $0.00 \pm 0.00$ | 8/20 |
| MG-Empty (4) | PSRL | $0.01 \pm 0.01$ | $0.01 \pm 0.01$ | $0.99 \pm 0.01$ | $0.00 \pm 0.00$ | 0/20 |
| | Q-learning | $0.02 \pm 0.01$ | $0.02 \pm 0.01$ | $0.98 \pm 0.01$ | $0.00 \pm 0.00$ | 0/20 |
| | UCRL2 | $0.01 \pm 0.00$ | $0.46 \pm 0.26$ | $0.54 \pm 0.26$ | $0.00 \pm 0.00$ | 0/20 |
| MG-Empty (5) | PSRL | $0.03 \pm 0.06$ | $0.17 \pm 0.30$ | $0.83 \pm 0.30$ | $0.00 \pm 0.00$ | 0/20 |
| | Q-learning | $0.05 \pm 0.01$ | $0.04 \pm 0.01$ | $0.96 \pm 0.01$ | $0.00 \pm 0.00$ | 12/20 |
| | UCRL2 | $0.26 \pm 0.23$ | $0.99 \pm 0.00$ | $0.01 \pm 0.00$ | $0.00 \pm 0.00$ | 19/20 |
| MG-Empty (6) | PSRL | $0.01 \pm 0.01$ | $0.01 \pm 0.02$ | $0.99 \pm 0.02$ | $0.00 \pm 0.00$ | 0/20 |
| | Q-learning | $0.02 \pm 0.02$ | $0.02 \pm 0.02$ | $0.98 \pm 0.02$ | $0.00 \pm 0.00$ | 0/20 |
| | UCRL2 | $0.01 \pm 0.00$ | $0.55 \pm 0.34$ | $0.45 \pm 0.34$ | $0.00 \pm 0.00$ | 0/20 |
| MG-Empty (7) | PSRL | $0.01 \pm 0.01$ | $0.01 \pm 0.01$ | $0.99 \pm 0.01$ | $0.00 \pm 0.00$ | 0/20 |
| | Q-learning | $0.02 \pm 0.02$ | $0.02 \pm 0.03$ | $0.98 \pm 0.03$ | $0.00 \pm 0.00$ | 0/20 |
| | UCRL2 | $0.01 \pm 0.00$ | $0.73 \pm 0.32$ | $0.27 \pm 0.32$ | $0.00 \pm 0.00$ | 0/20 |
| MG-Empty (8) | PSRL | $0.01 \pm 0.01$ | $0.01 \pm 0.01$ | $0.99 \pm 0.01$ | $0.00 \pm 0.00$ | 0/20 |
| | Q-learning | $0.02 \pm 0.01$ | $0.02 \pm 0.01$ | $0.98 \pm 0.01$ | $0.00 \pm 0.00$ | 0/20 |
| | UCRL2 | $0.01 \pm 0.00$ | $0.07 \pm 0.09$ | $0.93 \pm 0.09$ | $0.00 \pm 0.00$ | 0/20 |
| MG-Rooms (1) | PSRL | $0.01 \pm 0.04$ | $0.01 \pm 0.02$ | $0.99 \pm 0.02$ | $0.00 \pm 0.00$ | 0/20 |
| | Q-learning | $0.04 \pm 0.02$ | $0.02 \pm 0.03$ | $0.98 \pm 0.03$ | $0.00 \pm 0.00$ | 0/20 |
| | UCRL2 | $0.01 \pm 0.00$ | $0.82 \pm 0.28$ | $0.18 \pm 0.28$ | $0.00 \pm 0.00$ | 0/20 |
| MG-Rooms (2) | PSRL | $0.00 \pm 0.00$ | $0.00 \pm 0.00$ | $1.00 \pm 0.00$ | $0.00 \pm 0.00$ | 0/20 |
| | Q-learning | $0.01 \pm 0.02$ | $0.02 \pm 0.02$ | $0.98 \pm 0.02$ | $0.00 \pm 0.00$ | 0/20 |
| | UCRL2 | $0.00 \pm 0.00$ | $0.38 \pm 0.35$ | $0.62 \pm 0.35$ | $0.00 \pm 0.00$ | 0/20 |
| RiverSwim (1) | PSRL | $0.77 \pm 0.38$ | $1.00 \pm 0.00$ | $0.00 \pm 0.00$ | $0.01 \pm 0.00$ | 20/20 |
| | Q-learning | $0.21 \pm 0.15$ | $0.27 \pm 0.19$ | $0.73 \pm 0.19$ | $0.01 \pm 0.01$ | 20/20 |
| | UCRL2 | $0.85 \pm 0.24$ | $1.00 \pm 0.00$ | $0.00 \pm 0.00$ | $0.02 \pm 0.00$ | 20/20 |
| RiverSwim (2) | PSRL | $0.90 \pm 0.30$ | $1.00 \pm 0.00$ | $0.00 \pm 0.00$ | $0.01 \pm 0.00$ | 20/20 |
| | Q-learning | $0.29 \pm 0.21$ | $0.29 \pm 0.21$ | $0.71 \pm 0.21$ | $0.01 \pm 0.00$ | 20/20 |
| | UCRL2 | $0.90 \pm 0.20$ | $0.99 \pm 0.00$ | $0.01 \pm 0.00$ | $0.02 \pm 0.00$ | 20/20 |
| RiverSwim (3) | PSRL | $0.88 \pm 0.24$ | $0.98 \pm 0.04$ | $0.02 \pm 0.04$ | $0.01 \pm 0.00$ | 20/20 |
| | Q-learning | $0.09 \pm 0.06$ | $0.10 \pm 0.06$ | $0.90 \pm 0.06$ | $0.01 \pm 0.00$ | 20/20 |
| | UCRL2 | $0.80 \pm 0.25$ | $0.99 \pm 0.01$ | $0.01 \pm 0.01$ | $0.01 \pm 0.00$ | 20/20 |
| RiverSwim (4) | PSRL | $0.89 \pm 0.26$ | $0.99 \pm 0.00$ | $0.01 \pm 0.00$ | $0.01 \pm 0.00$ | 20/20 |
| | Q-learning | $0.82 \pm 0.20$ | $0.50 \pm 0.24$ | $0.50 \pm 0.25$ | $0.02 \pm 0.01$ | 20/20 |
| | UCRL2 | $0.82 \pm 0.29$ | $0.99 \pm 0.01$ | $0.01 \pm 0.01$ | $0.01 \pm 0.00$ | 20/20 |
| SimpleGrid (1) | PSRL | $0.22 \pm 0.07$ | $0.30 \pm 0.19$ | $0.70 \pm 0.19$ | $0.00 \pm 0.00$ | 0/20 |
| | Q-learning | $0.22 \pm 0.00$ | $0.22 \pm 0.00$ | $0.78 \pm 0.00$ | $0.01 \pm 0.00$ | 20/20 |
| | UCRL2 | $0.24 \pm 0.04$ | $0.99 \pm 0.01$ | $0.01 \pm 0.01$ | $0.00 \pm 0.00$ | 20/20 |
| SimpleGrid (2) | PSRL | $0.78 \pm 0.22$ | $0.99 \pm 0.02$ | $0.01 \pm 0.02$ | $0.00 \pm 0.00$ | 20/20 |
| | Q-learning | $0.67 \pm 0.04$ | $0.54 \pm 0.07$ | $0.46 \pm 0.07$ | $0.01 \pm 0.00$ | 20/20 |
| | UCRL2 | $0.79 \pm 0.16$ | $1.00 \pm 0.00$ | $0.00 \pm 0.00$ | $0.01 \pm 0.00$ | 19/20 |
| SimpleGrid (3) | PSRL | $0.48 \pm 0.10$ | $0.57 \pm 0.16$ | $0.43 \pm 0.16$ | $0.00 \pm 0.00$ | 0/20 |
| | Q-learning | $0.51 \pm 0.00$ | $0.51 \pm 0.00$ | $0.49 \pm 0.00$ | $0.00 \pm 0.00$ | 15/20 |
| | UCRL2 | $0.50 \pm 0.00$ | $1.00 \pm 0.00$ | $0.00 \pm 0.00$ | $0.00 \pm 0.00$ | 0/20 |
| Taxi (1) | PSRL | $0.13 \pm 0.08$ | $0.11 \pm 0.08$ | $0.89 \pm 0.08$ | $0.00 \pm 0.00$ | 0/20 |
| | Q-learning | $0.19 \pm 0.01$ | $0.13 \pm 0.01$ | $0.87 \pm 0.01$ | $0.01 \pm 0.00$ | 20/20 |
| | UCRL2 | $0.14 \pm 0.06$ | $0.91 \pm 0.01$ | $0.09 \pm 0.01$ | $0.00 \pm 0.00$ | 18/20 |

### E.5 Non-tabular setting

We now present the full results of the benchmarking procedure for the non-tabular baseline agents from `bsuite`, ActorCritic, ActorCriticRNN, BootDQN, and DQN.

In order to keep the hardness induced by the emission map as low as possible while still testing the effective non-tabular capabilities of the agent, we employ the deterministic state information map, which provides clear state-identifying information. The experimental procedure is the same as for the tabular case (described in the previous Section) with the only difference being that the total training time given to the agent is $40$ minutes both for the continuous and episodic settings.

The performances of the agents are in line with the results reported in Osband et al. [4], with the exception of BootDQN. Differently from `bsuite`, the `Colosseum` benchmarking procedure penalizes particularly computationally expensive algorithms by limiting the training time. Being BootDQN the most computationally intensive algorithm due to the ensemble training, this agent often breaks the time limit, which consequently worsens its overall performance.

Tables 15a, 15b, 15c, and 15d report a summary of the performance of the agents in terms of normalized cumulative regrets. DQN is the best performing agent on average for all the different settings. Unsurprisingly, it performs relatively better in the ergodic setting compared to the communicating one, in which the exploration challenge is harder. This is especially clear from the results for the `DeepSea` family in the continuous setting. DQN is not able to perform well in any of the instances in the communicating case (Table 15c). Although BootDQN generally performs better than the ActorCritic agents, it is not able to fully express its potential due to the associated computational burden. Interestingly, the ActorCritic agent without the recurrent component performs better than the version equipped with it. Note that this is not due to a different computational cost as ActorCriticRNN is always able to complete the interactions with the MDPs within the given time limit.

Figure 23, 24, 25, and 26 show how the cumulative regret, along with standard error intervals, of the agents evolve during their interaction with the benchmark environments. Note that the } symbol is always reported for BootDQN, meaning that, for at least one of the seeds, the agent ran out of time. We note that, overall, the variability of the performances across the different seeds is low, with the exception of a few cases. In particular, the `RiverSwim` family is associated with moderate variability. This is not surprising since, for this MDP family, the agent can easily get trapped in a sub-optimal policy if it does not fully explore the state space.

Figures 27a, 27b, 27c, and 27d place the regret of the agents on a position corresponding to the diameter and value norm of each MDP respectively for the episodic communicating, episodic ergodic, continuous communicating, and continuous ergodic settings. Overall, we observe that the harder environments effectively induce higher regret. This relationship is visibly stronger when the agents perform better across the environments, (see the ActorCritic agents in Figure 27c and DQN in all settings). Interestingly, and similarly to the tabular case, while the best performing agent (DQN) is evidently impacted more by the diameter, the opposite holds for the other agents. Regardless of the visitation complexity, this suggests that an agent that fails to handle the estimation complexity of an environment is bound to perform badly both in tabular and non-tabular settings.

Table 15: Summary of benchmark results for the non-tabular `bsuite` baselines.

(a) Episodic communicating setting.

| | ActorCritic | ActorCriticRNN | BootDQN | DQN |
|---|---|---|---|---|
| DeepSea | $0.33 \pm 0.24$ | $0.45 \pm 0.14$ | $0.17 \pm 0.24$ | $\mathbf{0.05} \pm 0.09$ |
| | $0.32 \pm 0.28$ | $0.45 \pm 0.21$ | $0.28 \pm 0.31$ | $\mathbf{0.21} \pm 0.26$ |
| FrozenLake | $0.12 \pm 0.08$ | $0.42 \pm 0.06$ | $0.25 \pm 0.25$ | $\mathbf{0.02} \pm 0.01$ |
| MG-Empty | $0.75 \pm 0.38$ | $0.92 \pm 0.15$ | $0.26 \pm 0.38$ | $\mathbf{0.04} \pm 0.07$ |
| | $0.81 \pm 0.29$ | $0.94 \pm 0.11$ | $0.43 \pm 0.37$ | $\mathbf{0.09} \pm 0.06$ |
| | $0.60 \pm 0.33$ | $0.85 \pm 0.19$ | $0.39 \pm 0.40$ | $\mathbf{0.07} \pm 0.03$ |
| | $0.96 \pm 0.08$ | $0.96 \pm 0.07$ | $0.31 \pm 0.39$ | $\mathbf{0.10} \pm 0.28$ |
| | $0.99 \pm 0.03$ | $0.96 \pm 0.06$ | $0.33 \pm 0.33$ | $\mathbf{0.10} \pm 0.09$ |
| MG-Rooms | $1.00 \pm 0.00$ | $1.00 \pm 0.00$ | $0.31 \pm 0.37$ | $\mathbf{0.19} \pm 0.32$ |
| | $1.00 \pm 0.01$ | $1.00 \pm 0.00$ | $\mathbf{0.34} \pm 0.39$ | $\mathbf{0.34} \pm 0.48$ |
| | $1.00 \pm 0.00$ | $1.00 \pm 0.01$ | $0.54 \pm 0.43$ | $\mathbf{0.36} \pm 0.48$ |
| | $0.87 \pm 0.30$ | $1.00 \pm 0.00$ | $0.44 \pm 0.41$ | $\mathbf{0.27} \pm 0.44$ |
| RiverSwim | $\mathbf{0.00} \pm 0.00$ | $\mathbf{0.00} \pm 0.00$ | $0.01 \pm 0.01$ | $\mathbf{0.00} \pm 0.00$ |
| | $0.53 \pm 0.39$ | $0.67 \pm 0.31$ | $\mathbf{0.13} \pm 0.21$ | $0.21 \pm 0.36$ |
| SimpleGrid | $0.80 \pm 0.10$ | $0.78 \pm 0.13$ | $0.44 \pm 0.35$ | $\mathbf{0.06} \pm 0.02$ |
| | $0.80 \pm 0.01$ | $0.78 \pm 0.03$ | $0.38 \pm 0.32$ | $\mathbf{0.05} \pm 0.01$ |
| | $0.38 \pm 0.17$ | $0.29 \pm 0.24$ | $0.26 \pm 0.25$ | $\mathbf{0.00} \pm 0.00$ |
| | $0.79 \pm 0.06$ | $0.77 \pm 0.07$ | $0.48 \pm 0.28$ | $\mathbf{0.20} \pm 0.20$ |
| Taxi | $0.91 \pm 0.01$ | $0.91 \pm 0.01$ | $0.87 \pm 0.14$ | $\mathbf{0.66} \pm 0.24$ |
| | $\mathbf{0.86} \pm 0.01$ | $\mathbf{0.86} \pm 0.01$ | $0.90 \pm 0.02$ | $0.87 \pm 0.01$ |
| *Average* | $0.69 \pm 0.30$ | $0.75 \pm 0.28$ | $0.38 \pm 0.21$ | $\mathbf{0.19} \pm 0.22$ |

(b) Episodic ergodic setting.

| | ActorCritic | ActorCriticRNN | BootDQN | DQN |
|---|---|---|---|---|
| DeepSea | $0.46 \pm 0.04$ | $0.48 \pm 0.00$ | $\mathbf{0.23} \pm 0.18$ | $0.25 \pm 0.24$ |
| | $0.01 \pm 0.01$ | $\mathbf{0.00} \pm 0.00$ | $0.07 \pm 0.09$ | $\mathbf{0.00} \pm 0.00$ |
| FrozenLake | $0.67 \pm 0.23$ | $0.88 \pm 0.07$ | $0.34 \pm 0.33$ | $\mathbf{0.09} \pm 0.02$ |
| MG-Empty | $0.86 \pm 0.27$ | $0.96 \pm 0.09$ | $0.24 \pm 0.36$ | $\mathbf{0.04} \pm 0.03$ |
| | $0.96 \pm 0.07$ | $0.99 \pm 0.01$ | $0.42 \pm 0.29$ | $\mathbf{0.20} \pm 0.15$ |
| | $0.98 \pm 0.05$ | $0.98 \pm 0.03$ | $0.35 \pm 0.31$ | $\mathbf{0.27} \pm 0.36$ |
| | $0.83 \pm 0.25$ | $0.96 \pm 0.07$ | $0.35 \pm 0.26$ | $\mathbf{0.08} \pm 0.03$ |
| | $0.77 \pm 0.23$ | $0.95 \pm 0.06$ | $0.55 \pm 0.25$ | $\mathbf{0.29} \pm 0.15$ |
| | $0.71 \pm 0.38$ | $0.65 \pm 0.25$ | $0.24 \pm 0.37$ | $\mathbf{0.02} \pm 0.02$ |
| | $0.27 \pm 0.14$ | $0.69 \pm 0.16$ | $0.28 \pm 0.35$ | $\mathbf{0.04} \pm 0.01$ |
| MG-Rooms | $0.87 \pm 0.22$ | $0.94 \pm 0.16$ | $0.52 \pm 0.43$ | $\mathbf{0.16} \pm 0.28$ |
| | $0.94 \pm 0.16$ | $0.96 \pm 0.08$ | $0.58 \pm 0.40$ | $\mathbf{0.36} \pm 0.47$ |
| | $0.93 \pm 0.16$ | $1.00 \pm 0.00$ | $0.31 \pm 0.35$ | $\mathbf{0.22} \pm 0.37$ |
| RiverSwim | $\mathbf{0.00} \pm 0.00$ | $\mathbf{0.00} \pm 0.00$ | $\mathbf{0.00} \pm 0.00$ | $\mathbf{0.00} \pm 0.00$ |
| | $\mathbf{0.00} \pm 0.01$ | $\mathbf{0.00} \pm 0.00$ | $0.01 \pm 0.01$ | $\mathbf{0.00} \pm 0.00$ |
| SimpleGrid | $0.53 \pm 0.27$ | $0.67 \pm 0.23$ | $0.37 \pm 0.37$ | $\mathbf{0.01} \pm 0.01$ |
| | $0.79 \pm 0.04$ | $0.78 \pm 0.06$ | $0.43 \pm 0.29$ | $\mathbf{0.09} \pm 0.03$ |
| | $0.51 \pm 0.03$ | $0.47 \pm 0.07$ | $0.24 \pm 0.22$ | $\mathbf{0.04} \pm 0.01$ |
| Taxi | $0.77 \pm 0.01$ | $0.77 \pm 0.01$ | $0.76 \pm 0.16$ | $\mathbf{0.54} \pm 0.24$ |
| | $0.34 \pm 0.02$ | $0.34 \pm 0.02$ | $0.36 \pm 0.04$ | $\mathbf{0.32} \pm 0.02$ |
| *Average* | $0.61 \pm 0.32$ | $0.67 \pm 0.34$ | $0.33 \pm 0.18$ | $\mathbf{0.15} \pm 0.15$ |

(c) Continuous communicating setting.

| | ActorCritic | ActorCriticRNN | BootDQN | DQN |
|---|---|---|---|---|
| DeepSea | $\mathbf{0.14} \pm 0.25$ | $0.18 \pm 0.27$ | $0.61 \pm 0.20$ | $0.55 \pm 0.00$ |
| | $\mathbf{0.17} \pm 0.38$ | $0.41 \pm 0.50$ | $0.85 \pm 0.27$ | $0.74 \pm 0.43$ |
| | $\mathbf{0.18} \pm 0.27$ | $0.23 \pm 0.28$ | $0.62 \pm 0.15$ | $0.54 \pm 0.00$ |
| FrozenLake | $\mathbf{0.06} \pm 0.04$ | $0.17 \pm 0.12$ | $0.26 \pm 0.11$ | $0.07 \pm 0.03$ |
| | $0.27 \pm 0.17$ | $0.29 \pm 0.16$ | $0.51 \pm 0.14$ | $\mathbf{0.18} \pm 0.04$ |
| MG-Empty | $0.92 \pm 0.29$ | $0.92 \pm 0.29$ | $0.77 \pm 0.41$ | $\mathbf{0.14} \pm 0.16$ |
| | $0.26 \pm 0.33$ | $0.34 \pm 0.42$ | $0.38 \pm 0.43$ | $\mathbf{0.10} \pm 0.22$ |
| | $\mathbf{0.13} \pm 0.28$ | $0.16 \pm 0.30$ | $0.32 \pm 0.43$ | $0.14 \pm 0.30$ |
| | $0.92 \pm 0.29$ | $0.75 \pm 0.45$ | $0.77 \pm 0.41$ | $\mathbf{0.06} \pm 0.06$ |
| | $1.00 \pm 0.00$ | $1.00 \pm 0.00$ | $0.69 \pm 0.46$ | $\mathbf{0.26} \pm 0.27$ |
| MG-Rooms | $0.47 \pm 0.34$ | $0.61 \pm 0.47$ | $0.74 \pm 0.35$ | $\mathbf{0.26} \pm 0.32$ |
| | $0.34 \pm 0.35$ | $0.71 \pm 0.34$ | $0.62 \pm 0.43$ | $\mathbf{0.24} \pm 0.35$ |
| | $\mathbf{0.62} \pm 0.42$ | $0.77 \pm 0.41$ | $0.77 \pm 0.36$ | $0.69 \pm 0.34$ |
| RiverSwim | $\mathbf{0.00} \pm 0.00$ | $0.07 \pm 0.23$ | $0.14 \pm 0.30$ | $\mathbf{0.00} \pm 0.00$ |
| | $0.07 \pm 0.23$ | $0.07 \pm 0.23$ | $\mathbf{0.00} \pm 0.01$ | $\mathbf{0.00} \pm 0.00$ |
| SimpleGrid | $\mathbf{0.00} \pm 0.01$ | $0.05 \pm 0.08$ | $0.55 \pm 0.48$ | $0.01 \pm 0.01$ |
| | $0.25 \pm 0.26$ | $0.38 \pm 0.23$ | $0.35 \pm 0.22$ | $\mathbf{0.04} \pm 0.11$ |
| | $0.09 \pm 0.27$ | $0.17 \pm 0.28$ | $0.55 \pm 0.45$ | $\mathbf{0.01} \pm 0.01$ |
| | $0.33 \pm 0.25$ | $0.42 \pm 0.19$ | $0.30 \pm 0.25$ | $\mathbf{0.01} \pm 0.02$ |
| Taxi | $0.92 \pm 0.01$ | $0.92 \pm 0.01$ | $0.93 \pm 0.02$ | $\mathbf{0.81} \pm 0.17$ |
| *Average* | $0.36 \pm 0.33$ | $0.43 \pm 0.31$ | $0.54 \pm 0.24$ | $\mathbf{0.24} \pm 0.26$ |

(d) Continuous ergodic setting.

| | ActorCritic | ActorCriticRNN | BootDQN | DQN |
|---|---|---|---|---|
| DeepSea | $0.29 \pm 0.43$ | $0.36 \pm 0.45$ | $0.27 \pm 0.21$ | $\mathbf{0.22} \pm 0.39$ |
| FrozenLake | $\mathbf{0.11} \pm 0.13$ | $0.21 \pm 0.28$ | $0.41 \pm 0.17$ | $\mathbf{0.11} \pm 0.08$ |
| MG-Empty | $0.92 \pm 0.07$ | $0.97 \pm 0.04$ | $0.32 \pm 0.18$ | $\mathbf{0.05} \pm 0.02$ |
| | $0.95 \pm 0.02$ | $0.98 \pm 0.01$ | $0.35 \pm 0.29$ | $\mathbf{0.05} \pm 0.02$ |
| | $0.87 \pm 0.07$ | $0.94 \pm 0.06$ | $0.39 \pm 0.18$ | $\mathbf{0.10} \pm 0.02$ |
| | $0.97 \pm 0.02$ | $0.97 \pm 0.03$ | $0.42 \pm 0.24$ | $\mathbf{0.08} \pm 0.04$ |
| | $0.97 \pm 0.03$ | $0.99 \pm 0.00$ | $0.57 \pm 0.40$ | $\mathbf{0.08} \pm 0.05$ |
| | $0.99 \pm 0.01$ | $0.99 \pm 0.01$ | $0.69 \pm 0.36$ | $\mathbf{0.12} \pm 0.07$ |
| | $0.98 \pm 0.02$ | $0.99 \pm 0.01$ | $0.54 \pm 0.43$ | $\mathbf{0.10} \pm 0.06$ |
| | $0.99 \pm 0.00$ | $0.98 \pm 0.01$ | $0.73 \pm 0.28$ | $\mathbf{0.17} \pm 0.08$ |
| MG-Rooms | $0.97 \pm 0.05$ | $0.99 \pm 0.01$ | $0.59 \pm 0.35$ | $\mathbf{0.24} \pm 0.17$ |
| | $0.99 \pm 0.00$ | $0.99 \pm 0.00$ | $0.59 \pm 0.25$ | $\mathbf{0.38} \pm 0.27$ |
| RiverSwim | $0.08 \pm 0.29$ | $0.33 \pm 0.49$ | $\mathbf{0.00} \pm 0.01$ | $\mathbf{0.00} \pm 0.00$ |
| | $0.07 \pm 0.23$ | $0.07 \pm 0.23$ | $0.01 \pm 0.03$ | $\mathbf{0.00} \pm 0.00$ |
| | $\mathbf{0.00} \pm 0.00$ | $0.13 \pm 0.31$ | $0.02 \pm 0.04$ | $\mathbf{0.00} \pm 0.00$ |
| | $0.33 \pm 0.49$ | $0.17 \pm 0.39$ | $0.01 \pm 0.02$ | $\mathbf{0.00} \pm 0.00$ |
| SimpleGrid | $0.75 \pm 0.01$ | $0.76 \pm 0.01$ | $0.57 \pm 0.27$ | $\mathbf{0.09} \pm 0.04$ |
| | $0.49 \pm 0.01$ | $0.49 \pm 0.01$ | $0.36 \pm 0.16$ | $\mathbf{0.02} \pm 0.01$ |
| | $0.50 \pm 0.00$ | $0.50 \pm 0.00$ | $0.37 \pm 0.18$ | $\mathbf{0.06} \pm 0.04$ |
| Taxi | $0.78 \pm 0.02$ | $0.78 \pm 0.02$ | $0.81 \pm 0.03$ | $\mathbf{0.55} \pm 0.13$ |
| *Average* | $0.65 \pm 0.36$ | $0.68 \pm 0.34$ | $0.40 \pm 0.24$ | $\mathbf{0.12} \pm 0.14$ |

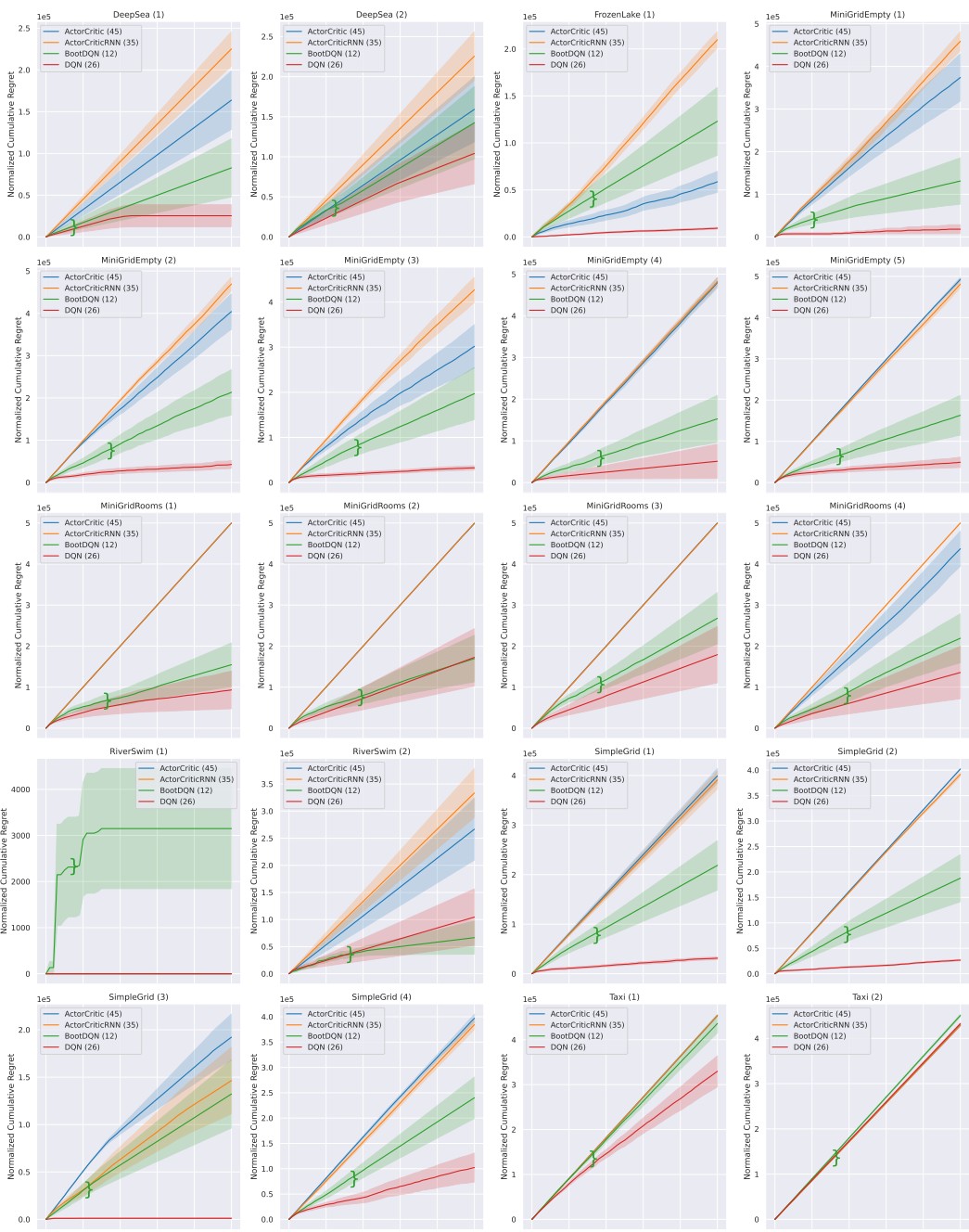

Figure 23: Full interaction performances for the non-tabular baseline agents in the episodic communicating setting.

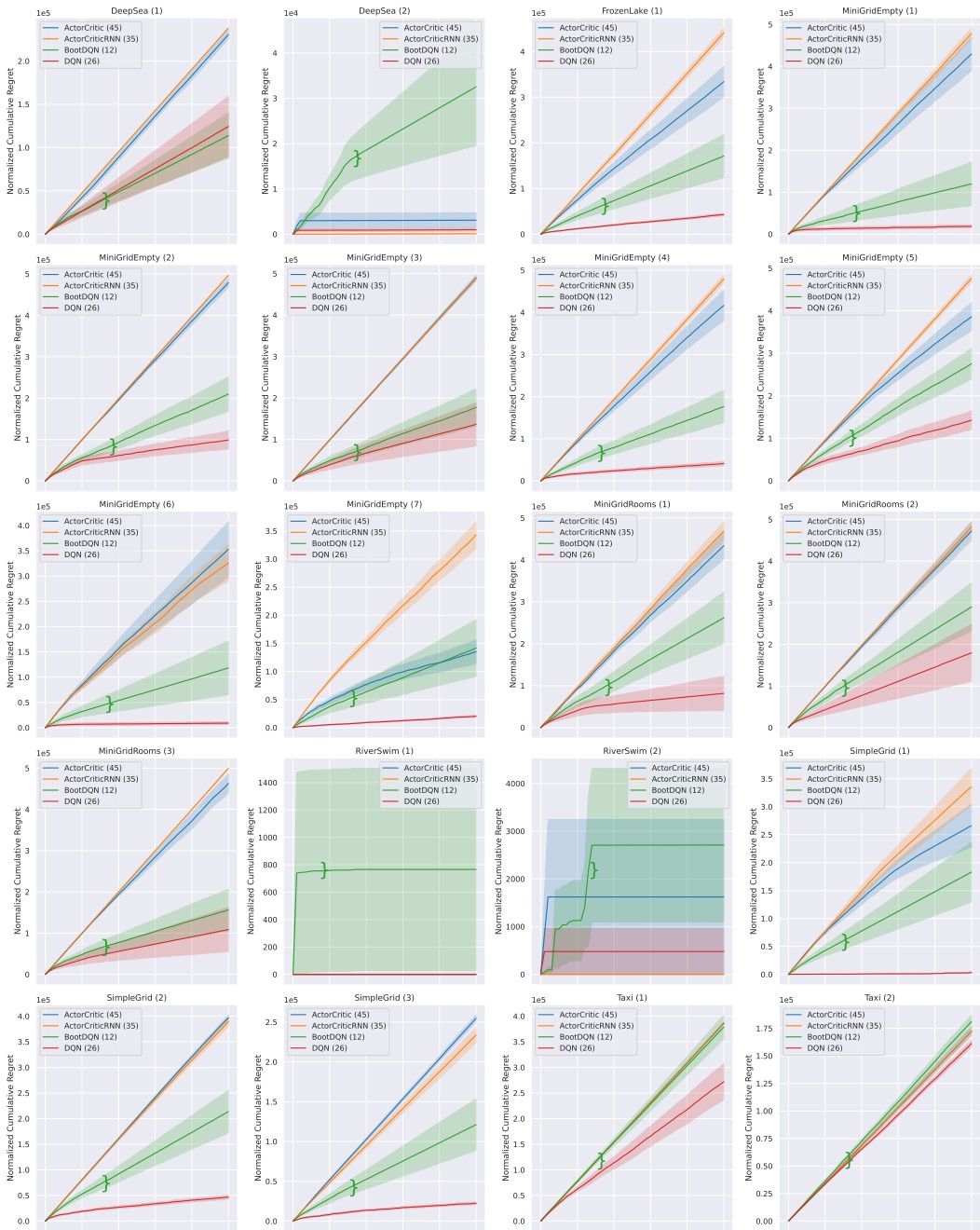

Figure 24: Full interaction performances for the non-tabular baseline agents in the episodic ergodic setting.

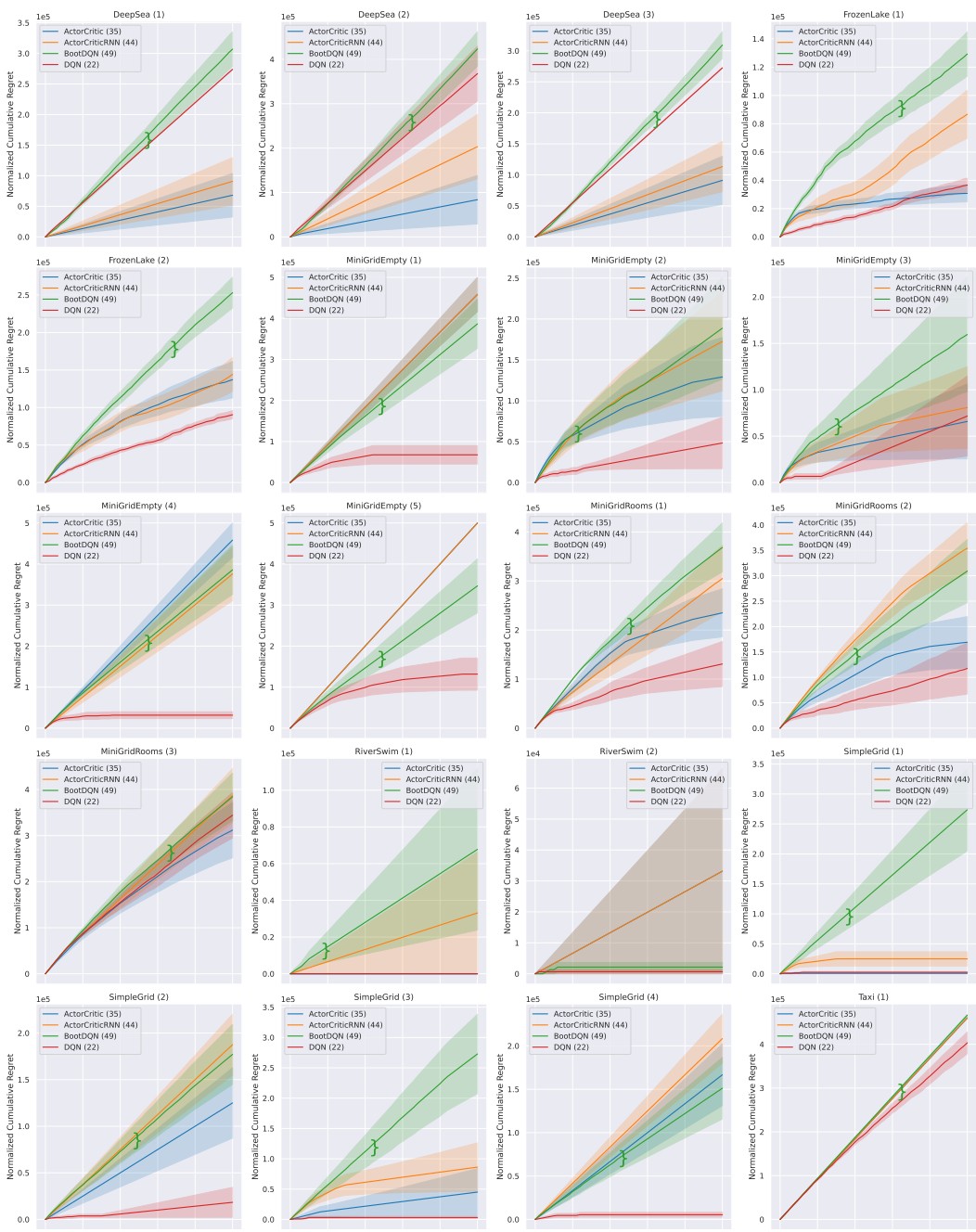

Figure 25: Full interaction performances for the non-tabular baseline agents in the continuous communicating setting.

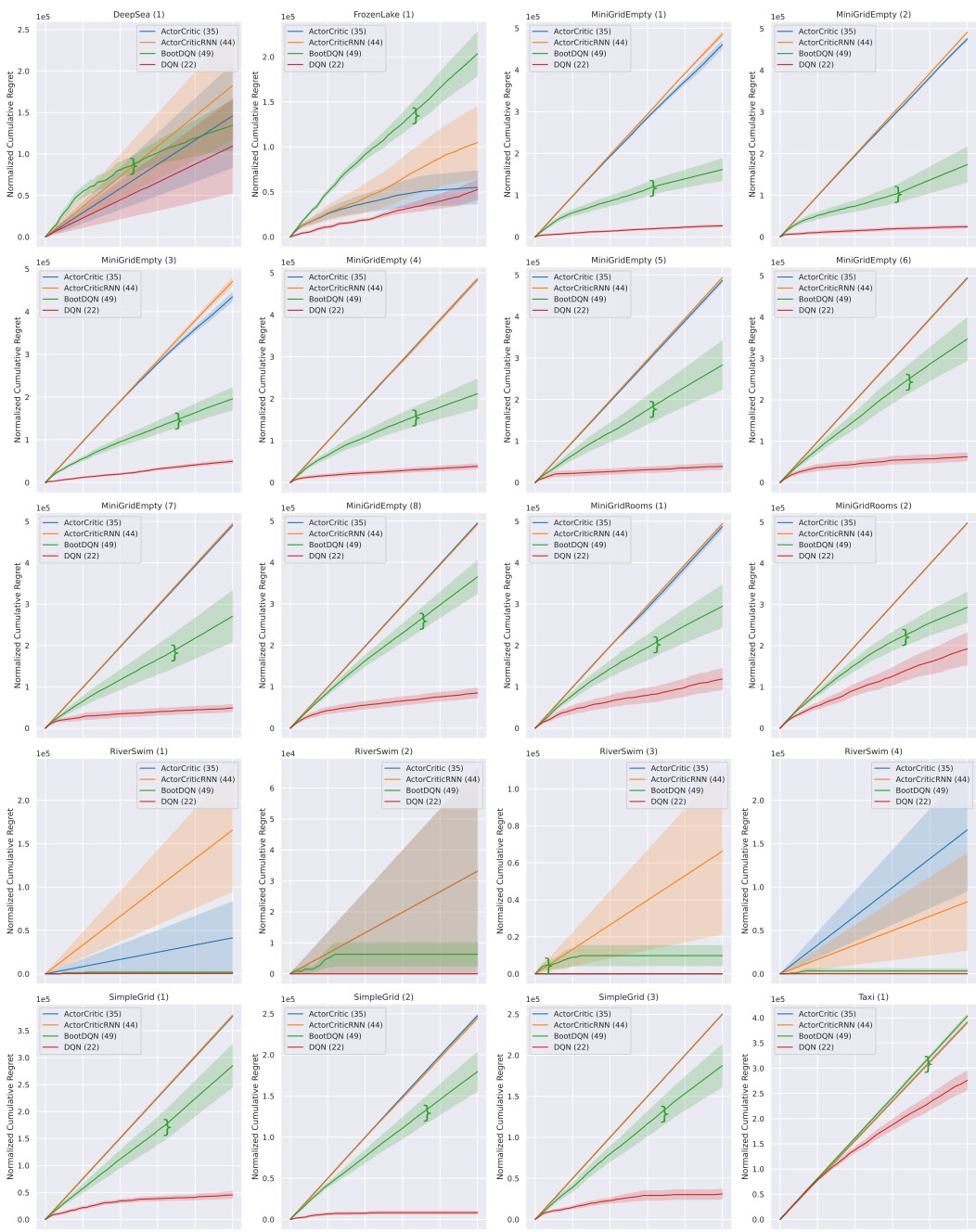

Figure 26: Full interaction performances for the non-tabular baseline agents in the continuous ergodic setting.

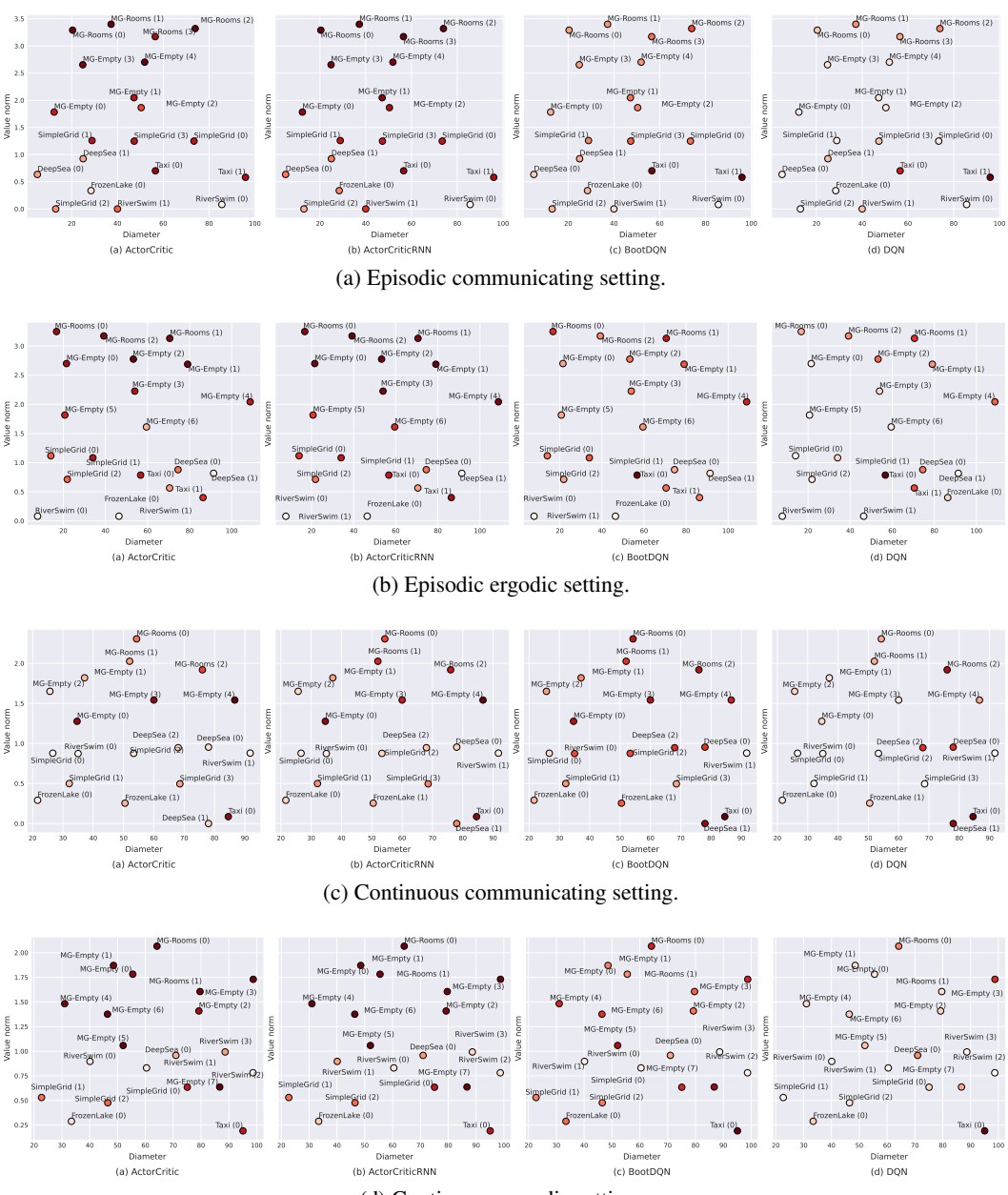

Figure 27: Average cumulative regret obtained by the agents in the continuous ergodic setting placed according to the diameter and the value norm values of the benchmark MDPs.