# OpenReview forum: "Hardness in Markov Decision Processes: Theory and Practice"
_NeurIPS.cc/2022/Conference — NeurIPS 2022 Accept_

### Official Review · Reviewer_cqd5 · 2022-06-14

**Rating:** 6
**Confidence:** 3
**Soundness:** 3 good
**Presentation:** 4 excellent
**Contribution:** 3 good

**Summary:**

This paper reviews and categorizes some hardness measures on tabular MDPs, and proposes a new tabular RL benchmark named "Colosseum" that enables exact computation of these measures in empirical evaluations. Extensive experiments are conducted to assess the performance of existing tabular RL methods on the proposed benchmark that spans diverse environments focusing on different hardness measures.

**Questions:**

1. The authors wrote in the paper "we believe our contributions are important milestones towards the future development of theoretical and empirical measures of hardness for non-tabular reinforcement learning". However, it is not clear how the measures presented in the paper can be extended to non-tabular settings (while still being computationally feasible). Adding more discussion on this point can be helpful.
2. The paper surveys existing hardness measures by dividing them into two categories, namely "Markov chain-based" which focuses on the state visitation complexity, and "value-based" which focuses on the estimation complexity of value functions. It seems like this taxonomy is closely relevant to the well-known "exploitation-exploration" dilemma with state visitation complexity capturing the difficulty in exploration, and estimation complexity capturing the difficulty in exploitation. The authors are encouraged to discuss this relation in more detail.
3. The message conveyed by Definition 2.1 is somewhat vague: it is not clear to me how this "complete measure" connects to the intuition or other theoretical arguments that capture the "hardness" of learning a near-optimal policy on an MDP, e.g., the PAC learnability. In other words, is there any conceptual/theoretical evidence implying that this hardness measure is _sufficient_ in the sense of some criteria? Also, while I can understand that existing measures may be insufficient in measuring all aspects of hardness since they either focus on the state visitation complexity or the estimation complexity, can we use Definition 2.1 to certify this?

**Limitations:**

No concern.

**Strengths And Weaknesses:**

### Strengths
1. The paper suggests an interesting viewpoint of connecting the theory and practice of tabular RL by taking into account the hardness measures that are typically used only in theoretical analysis when evaluating and comparing tabular RL algorithms.
2. The "Colosseum" benchmark is of high quality, flexible, and with sufficient docs and tutorials.
3. The empirical evaluations are thorough.
4. The paper is well organized and presented.

### Weaknesses
1. The benchmark and the hardness measures only apply to tabular RL. In contrast, most of the current empirical work in RL is devoted to the non-tabular setting, and there is also an increasing number of theoretical work that explores non-tabular RL.
2. The theoretical claims made by this paper are somewhat vague (see the following for details).

---

> ### Author Response · Authors · 2022-08-02
> **Response to reviewer cqd5 (Part 1/2)**
>
> Thank you very much for the time dedicated to evaluate our work. We are glad that you found that our paper presents an interesting viewpoint in a well-organized manner. We are also glad that you found the Colosseum benchmark (and associated tools) to be flexible and of high quality and that you are happy with the thoroughness of the empirical evaluation.
>
> In order to address your concern about the benchmark and hardness measures only being applicable to tabular reinforcement learning, we want to emphasize that the principled selection of environments for the Colosseum benchmark can play a fundamental role in the development and analysis of new tabular and non-tabular reinforcement learning algorithms. Although the text has mostly focused on using Colosseum for studying tabular reinforcement learning algorithms, an up-to-date version of the package already allows experimentation using non-tabular representations of the existing environments. Concretely, it is possible to use vector-based state representations that allow Colosseum to benchmark reinforcement learning algorithms that employ both linear and non-linear function approximation. Currently available vector-based state(-action) representations include one-hot encodings, positional representations (for grid-like environments), and a representation that guarantees the linearity of the optimal action-value function [2].
>
> In the early stages of developing a new reinforcement learning algorithm, it is often very useful to test ideas in simple environments, which allows fast iteration. Having a thorough and efficient benchmark equipped with useful analysis tools allows researchers to focus most of their time on the algorithms themselves. Although there is no well-developed theory of hardness for non-tabular reinforcement learning, we argue that success in hard tabular environments is likely a necessary (but not sufficient) condition for success in hard non-tabular environments. The fact that we do not include strictly non-tabular environments stems from our goal of providing a principled benchmark, which is currently not possible for non-tabular reinforcement learning due to hardness being a relatively recent (and active) focus of research in that setting [1]. Similarly, we do not include results of a comparison between state-of-the-art non-tabular reinforcement learning algorithms in the Colosseum benchmark because we believe such comparison would be premature and likely unfair (by not considering all important aspects of hardness in the non-tabular setting). Extending the benchmark to environments that require function approximation in a principled way is one of the main goals of our future work.
>
> Finally, we note that an empirical benchmark may be important even in developing tabular reinforcement learning algorithms with theoretical guarantees. Some choices in those algorithms may have to be made without guidance from the theory, in which case they may be informed by practice. Ultimately, this is important because principled algorithms often inspire scalable counterparts.
>
> In hindsight, we believe that the text does not emphasize the potential of Colosseum for benchmarking non-tabular reinforcement learning algorithms enough, so we plan on changing the text substantially to reflect this. We will do so by highlighting this potential in the abstract and in the introduction, by explaining the corresponding functionalities in the section that introduces Colosseum, and by recalling this potential in the conclusion. Importantly, we will also add a Jupyter notebook tutorial that shows how to use non-tabular reinforcement learning algorithms with vector-based representations of Colosseum environments to the supplementary material.
>
> Part (1/2)

---

> > ### Author Response · Authors · 2022-08-02
> > **Response to reviewer cqd5 (Part 2/2)**
> >
> > Although an analogy between visitation complexity/estimation complexity and the exploration/exploitation dilemma is indeed attractive, we believe that it is potentially misleading. For instance, consider a Markov decision process where it is possible to transition directly from any state to any other state given an appropriate action. According to our Markov-chain based measures of visitation complexity, this environment would be considered easy (note that its diameter is one). However, the exploration/exploitation dilemma still exists. This is because, at every time step, the agent must choose between visiting states for which it has good (value) estimates or less visited states. The more states, the more difficult the trade-off becomes.
> >
> > Indeed, it is not obvious how to extend the hardness measures presented in the paper to the non-tabular setting, and completely novel measures may be required. However, we believe that our effort in surveying the literature for potential hardness measures and introducing desiderata for tabular measures may help and inspire these future developments. We will make these contributions clearer in the conclusion.
> >
> > Note that our definition of a complete measure of hardness connects it directly with a specific performance criterion (such as the sample complexity or the expected cumulative regret). Whenever a complete measure of hardness establishes a "difficulty ordering" between environments in a certain class, every near-optimal agent for that class (according to the same performance criterion) would by definition "agree" with that ordering. Note that a trivial complete measure of hardness ranks every environment equally, which guarantees that a complete measure of hardness exists. However, the existence of a complete measure of hardness is not in itself a guarantee that it is "sufficiently discerning" for all purposes.
> >
> > We hope that these clarifications and the proposed changes will allow you to consider increasing the rating given to our submission. We aim to provide an updated version of our work that incorporates reviewer feedback as soon as possible.
> >
> > Part (2/2)
> >
> > -----
> >
> > References:
> >
> > [1] Dylan J. Foster, Sham M. Kakade, Jian Qian, and Alexander Rakhlin. "The statistical complexity of interactive decision making." arXiv preprint arXiv:2112.13487 (2021).

---

> > > ### Comment · Reviewer_cqd5 · 2022-08-06
> > > **Response**
> > >
> > > Thank you for your response which has addressed most of my concerns. The only remaining part that I found not entirely clear is that while Definition 2.1 is indeed a desideratum that a "good" hardness measure should achieve, its connection with existing hardness measures remains indirect. As you have mentioned in Section 2.3, Definition 2.1 is proposed to "address the issues in the theory of hardness", including the issue of not simultanesouly capturing state visitation complexity and estimation complexity. I would then expect that one or more of the following holds:
> > >
> > > + Since many of existing hardness measures either focus on state visitation complexity or estimation complexity alone, we should be able to show that they do _not_ meet the condition in Definition 2.1 (maybe by constructing some toy MDPs).
> > >
> > > + More generally, we can probably also show that if a measure does not simultaneously consider the two complexities, it _cannot_ meet the condition in Definition 2.1 (or equivalently, if a measure does meet the condition in Definition 2.1, it must simultaneously consider the two complexities).
> > >
> > > Note that I am not asking for a rigorous proof for any of the above arguments since that the visitation complexity and estimation complexity themselves may be hard to define or quantify without a specific RL setting. An intuitive explanation should suffice to show that there indeed exists a connection between "a measure that satisfies Definition 2.1" and "a measure that simultaneously considers state visitation complexity and estimation complexity".

---

> > > > ### Author Response · Authors · 2022-08-09
> > > > **Response to the response of reviewer cqd5**
> > > >
> > > > Thanks for considering our response! We hope to dispel your remaining concerns.
> > > >
> > > > The following examples show how hardness measures that rely solely on visitation complexity or estimation complexity may fail to meet the current definition of a complete measure of hardness. Because visitation complexity and estimation complexity are not formally defined, these examples refer to *reasonable* hardness measures, which attempt to capture intuitive aspects of visitation complexity or estimation complexity.
> > > > - Consider an MDP $\text{M} = (\mathcal S, \mathcal A, P, P_0, R)$ and let $\text{M}' = (\mathcal S, \mathcal A, P', P_0, R')$ be identical to $\text{M}$ except for having an increased probability (prand) $r$ of executing a random action instead of the action selected by the agent and having reward $R'(s,a) = 0$ for every state $s$ and action $a$. Because every policy is optimal for $\text{M}'$, $\psi(\text{M}, A^*) \geq \psi(\text{M}', A^*) = 0$. However, because the agent has less control over transitions in $\text{M}'$, a *reasonable* measure of hardness $\theta_{\text{MC}}$ that focuses solely on visitation complexity may be independent of the reward kernel (which is the case for all Markov chain-based measures discussed in the paper), and therefore assign $\theta_{\text{MC}}(\text{M}) \leq \theta_{\text{MC}} (\text{M}')$. In non-degenerate MDPs, the previous inequalities would also be strict.
> > > > - Consider an MDP $\text{M} = (\mathcal S, \{0,1\}, P, P_0, R)$ where $\mathbb E(R(s, 0)) < \mathbb E(R(s, 1))$ for every state $s$ and let $\text{M}' = (\mathcal S, \{0,1\}, P, P_0, R')$ be identical to $\text{M}$ except for having larger gaps in the rewards means and significantly larger reward variances. Concretely, let $R'(s,0) = R(s, 0) - \mathcal{N}(\epsilon, \sigma^2)$ and $R'(s,1) = R(s, 1) + \mathcal{N}(\epsilon, \sigma^2)$ for every state $s$, where $\epsilon$ is a small constant and $\sigma^2$ is a large constant.
> > > > An agent requires a significantly higher number of samples to find an optimal policy in the presence of larger reward variances, which in turn implies $\psi(\text{M}, A^*) \leq \psi(\text{M}', A^*)$. However, because larger gaps typically make the estimation of the optimal value function easier, a *reasonable* measure of hardness $\theta_{\text{VB}}$ that focuses solely on estimation complexity may be independent of the reward variances (which is the case for all the value-based measures discussed in the paper), and therefore assign $\theta_{\text{VB}}(\text{M}) \geq \theta_{\text{VB}} (\text{M}')$.
> > > >
> > > > Although we have argued that it is possible to create examples where a reasonable hardness measure that relies solely on visitation complexity or estimation complexity may fail to meet the current definition of a complete measure of hardness, we would not argue that a measure that considers both estimation complexity and visitation complexity is necessarily complete. Such an argument would require formalizing the concepts of visitation complexity and estimation complexity, which may be an interesting (but challenging) problem for future work. The argument would also have to consider each reinforcement learning setting independently, since a measure of hardness is complete with respect to a specific criterion lower bound. Finally, the argument would be made even more difficult by the fact that the definition of complete measure of hardness also applies to non-tabular reinforcement learning, which also has known criterion lower bounds in some specific settings [1, 2, 3].
> > > >
> > > > We will try to summarize and include these insights into the paper to help readers understand the implications of the definition of a complete measure of hardness.
> > > >
> > > > **References**
> > > >
> > > > [1] Dongruo Zhou, Jiafan He, and Quanquan Gu. "Provably efficient reinforcement learning for discounted mdps with feature mapping." In International Conference on Machine Learning. PMLR, 2021.
> > > >
> > > > [2] Yuanhao Wang, Ruosong Wang, and Sham Kakade. "An exponential lower bound for linearly realizable mdp with constant suboptimality gap." Advances in Neural Information Processing Systems 34 (2021).
> > > >
> > > > [3] Andrea Zanette. "Exponential lower bounds for batch reinforcement learning: Batch rl can be exponentially harder than online rl." In International Conference on Machine Learning. PMLR, 2021.

---

> > > > > ### Comment · Reviewer_cqd5 · 2022-08-10
> > > > > **Thank you for the response**
> > > > >
> > > > > Thank you for the follow-up response that has addressed all of my remaining concerns. I think including these examples and discussions in the main text or the appendix would benefit future readers for a more clear understanding of the proposed complete hardness measure.

---

### Official Review · Reviewer_QBF7 · 2022-07-10

**Rating:** 6
**Confidence:** 4
**Soundness:** 3 good
**Presentation:** 3 good
**Contribution:** 2 fair

**Summary:**

This paper first presents a survey of existing hardness measures and results from the MDP literature. Their main contribution is the introduction of `Colosseum`, which is a benchmark for empirically validating hardness results, which they use to compare various existing hardness measures.

**Questions:**

1. Is all of section 2 just a review of existing literature? It seems like this is the case, but just wanted to confirm.
2. In line 72 it says "Given an optimization horizon $T$". What is this exactly? Is it related to episodic horizon?
3. In lines 99-100 it says there is an assumption of the existence of an optimal policy $\pi^*$ that "obtains maximum value for every state", but this policy will necessarily be dependent on time as well as state, no?
4. Why do you use the value of $0.25$ in equation (1)? This is not justified in the text.
5. Below line 186, $\gamma$ is a subscript of $C$ on the left-hand side, but it does not appear in the right hand side. Should it? Or should it be removed from $C$?
6. In lines 206-208 it says "approximating the value function ... is particularly easy when every sub-optimality gap is small." This is not necessarily the case in RL: sub-optimality gap is with respect to the optimal policy, but in RL we don't start from the optimal policy.
7. In lines 374-375 you say your empirical investigations "revealed previously undocumented weaknesses of these agents and further validated our choices of environments.". What does this say about existing empirical work evaluating these algorithms?

**Limitations:**

Some limitations are provided (mostly related to future work). It would be nice to have some discussion regarding the significance/impact (or lack thereof) of this work, in line with some of the comments I made above regarding significance.

No discussion of potential negative societal impact was provided.

**Strengths And Weaknesses:**

# Originality
It seems to me that section 2 (Hardness in Theory) is a review of existing literature, so the original contributions of this paper would be limited to the `Colosseum` package and the empirical investigations provided. For these, the work is original as far as I can tell, although some of the claims seem to be a bit inflated (e.g. "a pioneering Python package", "the most exhaustive in tabular reinforcement learning", "invaluable analysis tools", etc.).

# Quality
The authors have done a reasonably thorough survey of hardness literature and evaluated these measures using the various environments in their package. There are a few issues regarding clarity and correctness that I include in the questions below.
The code for `Colosseum` seems to be well-written and well-documented, which I consider to be a core part of this paper's contribution.

# Clarity
The paper is very well written and motivated reasonably well. Some of the plots and tables are hard to digest; specifically, it's often not clear _what_ is being said with them. There's a lot going on in Figure 1 (and even more in Table 1), and even though the main takeaways do seem to be discussed in the text, they're mostly lost in the paragraphs in page 7 (it seems that the last sentence is the main takeaway for each hardness measure). I would suggest highlighting these in a more streamlined manner (to draw the reader's attention directly to the takeaways) and leave the descriptive text until after.
Table 1 is a bit overwhelming, it's not clear what we're supposed to be looking for. I would also suggest rewriting this section so there are clear takeaways and insights for the readers; currently it reads just as a verbal description of the (many) numbers in the table.

Although it is claimed that in Figure 2 "there is generally a positive relationship between both of these hardness measures and the average cumulative regret", it seems almost like points uniformly spread on the plane (i.e. I don't see a clear relationship at all).

There are a few other issues I mention in the questions below.

# Significance
This, to me, is the weakest point of the paper. Although I appreciate the authors' effort to produce a nice package for benchmarking theoretical results, it's not clear how significant this will be. Empirical evaluations on toy environments (for theoretical results) are typically meant to highlight characteristics or subtleties of the theory introduced, but are not the end goal in itself. In particular, whether the empirical results suggest sub-linear or linear growth, say, does not in any way change the theoretical results. Thus, it is not clear what the added value would be to have a "theory benchmark".
Something that I think could make this package more impactful is to try to go beyond tabular. One suggestion would be to look at bsuite (which the authors do cite), as they include both tabular and continuous environments. In particular, it would be interesting to evaluate both as they may allow one to empirically investigate how the hardness measures vary when moving from tabular to larger systems. I acknowledge that in non-tabular systems it may not be possible to compute all of them in closed form, but there may be approximations; alternatively, continuous variants of the tabular systems considered could provide a nice middle ground (e.g. by "smoothing out" each tabular state).
Another aspect that could increase the significance of this work is to evaluate non-tabular methods, for instance with linear function approximators. A lot of RL theory does exist for linear approximators (and tabular, of course), so it would be interesting to evaluate how the dynamics of the empirical evaluations change (or not) when moving from tabular to non-tabular methods.

Along these lines, in line 366 it says "The development of such measures is theoretically and empirically important.". It would be nice to provide some concrete examples, such as a theoretical bound dependent on one of these hardness measures, or something like that. Otherwise it's not clear why these hardness measures are important.

---

> ### Author Response · Authors · 2022-08-02
> **Response to reviewer QBF7 (Part 1/3)**
>
> Thank you very much for the time dedicated to evaluate our work. We are glad that you found both the paper and the code well-written (and well-documented).
>
> We would like to clarify that Section 2 is not limited to reviewing existing literature. Instead, we propose a unifying perspective on what so far have been disparate notions of hardness and provide a qualitative comparison that considers their strengths and weaknesses. Note that most quantities that we consider hardness measures were originally proposed in the context of providing theoretical guarantees for reinforcement learning algorithms. We have re-interpreted these quantities as hardness measures. Our definition of a complete measure of hardness further aims to connect measures of hardness to performance criteria that are actually relevant in reinforcement learning, which hopefully will ground and guide future work on hardness.
>
> Although we understand that calling Colosseum "pioneering" and the benchmark "the most exhaustive" may sound like inflation of claims, Colosseum is indeed the first package to attempt to systematically connect theoretical hardness measures to empirical reinforcement learning. It is also the first package to implement many of the functionalities described in Section 3.1. The Colosseum benchmark also contains the largest (not to mention principled) selection of tabular reinforcement learning environments. However, we agree that calling the analysis tools "invaluable" is not appropriate, and so we will change that in the text.
>
> We agree that some of the plots and tables can take a long time to interpret, even though we spent a significant amount of time trying to convey the (large amount of complex) information as simply as possible. Our relatively long descriptive analysis tries to guide the reader through the most important findings. We believe that highlighting the summary of the descriptive analysis for each hardness measure on page 7 (lines 285-307) is a great suggestion. We will incorporate this in the text by moving each summary before the corresponding descriptive analysis. Hopefully, this will help readers focus on the most important results.
>
> Regarding Figure 2, the points are indeed spread somewhat uniformly on the plane, which shows how the continuous ergodic MDPs in the benchmark are diverse with respect to hardness measures. The claim that there is a generally positive relationship between both the measures and the average cumulative regret is related to the color of each point. The claim predicts that the farther along the main diagonal a point is, the darker it should be. In other words, the most difficult (according to the hardness measures) the MDP, the larger the (average cumulative) regret. This pattern is most clear in Figure 2a, but can also be seen in the other figures. We hope this eliminates your concern.
>
> In order to address your concerns about significance, we would like to emphasize that Colosseum is not intended mainly as a benchmark for empirically validating hardness measures or providing empirical guidance for theoretical developments, although it could certainly be used for those purposes.
>
> Most importantly, we believe that the principled selection of environments for the Colosseum benchmark can play a fundamental role in the development and analysis of new tabular and non-tabular reinforcement learning algorithms. Although the text has mostly focused on using Colosseum for studying tabular reinforcement learning algorithms, an up-to-date version of the package already allows experimentation using non-tabular representations of the existing environments. Concretely, it is possible to use vector-based state representations that allow Colosseum to benchmark reinforcement learning algorithms that employ both linear and non-linear function approximation. Currently available vector-based state(-action) representations include one-hot encodings, positional representations (for grid-like environments), and a representation that guarantees the linearity of the optimal action-value function [2].
>
> Part (1/3)

---

> > ### Author Response · Authors · 2022-08-02
> > **Response to reviewer QBF7 (Part 2/3)**
> >
> > In the early stages of developing a new reinforcement learning algorithm, it is often very useful to test ideas in simple environments, which allows fast iteration. Having a thorough and efficient benchmark equipped with useful analysis tools allows researchers to focus most of their time on the algorithms themselves. Although there is no well-developed theory of hardness for non-tabular reinforcement learning, we argue that success in hard tabular environments is likely a necessary (but not sufficient) condition for success in hard non-tabular environments. The fact that we do not include strictly non-tabular environments stems from our goal of providing a principled benchmark, which is currently not possible for non-tabular reinforcement learning due to hardness being a relatively recent (and active) focus of research in that setting [1]. Similarly, we do not include results of a comparison between state-of-the-art non-tabular reinforcement learning algorithms in the Colosseum benchmark because we believe such comparison would be premature and likely unfair (by not considering all important aspects of hardness in the non-tabular setting). Extending the benchmark to environments that require function approximation in a principled way is one of the main goals of our future work.
> >
> > Finally, we note that an empirical benchmark may be important even in developing tabular reinforcement learning algorithms with theoretical guarantees. Some choices in those algorithms may have to be made without guidance from the theory, in which case they may be informed by practice. Ultimately, this is important because principled algorithms often inspire scalable counterparts.
> >
> > In hindsight, we believe that the text does not emphasize the potential of Colosseum for benchmarking non-tabular reinforcement learning algorithms enough, so we plan on changing the text substantially to reflect this. We will do so by highlighting this potential in the abstract and in the introduction, by explaining the corresponding functionalities in the section that introduces Colosseum, and by recalling this potential in the conclusion. Importantly, we will also add a Jupyter notebook tutorial that shows how to use non-tabular reinforcement learning algorithms with vector-based representations of Colosseum environments to the supplementary material.
> >
> > We believe that the development of complete hardness measures is important for two main reasons. Empirically, it would allow the creation of principled benchmarks for different performance criteria. Theoretically, it could elucidate what makes a problem hard for a specific performance criterion. However, the lack of (non-trivial) complete measures of hardness does not allow us to relate them to existing performance bounds. We will make this clearer in the conclusion.
> >
> > Part (2/3)

---

> > > ### Author Response · Authors · 2022-08-02
> > > **Response to reviewer QBF7 (Part 3/3)**
> > >
> > > Here are the answers to the remaining questions, organized by occurrence in the text:
> > >
> > > Line 72: The optimization horizon is the total number of time steps for interaction between the agent and the environment. It may be infinite in some cases, but some algorithms depend on a known horizon.
> > >
> > > Lines 99-100: For the sake of simplicity, we consider the standard formulation of a Markov decision process where an optimal policy is independent of the time due to the Markov assumption (which guarantees that any information useful to predict next states and rewards would be encoded in the current state). For time-dependent transition and reward kernels, if the current time is not somehow encoded in the current state, the optimal policy is indeed time-dependent.
> > >
> > > Line 131: The value of 0.25 used in Equation (1) is a convention in the Markov chain literature. We will clarify this in the text.
> > >
> > > Line 186: We sometimes drop the corresponding subscript when the episodic or discounted setting is clear from the context. However, there is no reason for dropping the subscript in this case, so we will include it on the right side of the equation as suggested.
> > >
> > > Lines 206-208: When the sub-optimality gap is small for every state and action, taking a non-optimal action necessarily has a small impact on the expected value obtained by an agent. In the most extreme case, when the sub-optimality gap is zero for every state and action, every policy is optimal. We will try to clarify our claim that identifying near-optimal policies (and thus approximating the optimal value function) is particularly easy when every sub-optimality gap is small.
> > >
> > > Lines 374-375: Prior to our work, very little empirical work was done on evaluating these algorithms, which is why we believe the results presented in Table 1 are noteworthy.
> > >
> > > Finally, please let us know if you believe a discussion of potential negative societal impact is necessary.
> > >
> > > We hope that these clarifications and the proposed changes will allow you to consider increasing the rating given to our submission. We aim to provide an updated version of our work that incorporates reviewer feedback as soon as possible.
> > >
> > > Part (3/3)
> > >
> > > -----
> > >
> > > References:
> > >
> > > [1] Dylan J. Foster, Sham M. Kakade, Jian Qian, and Alexander Rakhlin. "The statistical complexity of interactive decision making." arXiv preprint arXiv:2112.13487 (2021).

---

> > ### Comment · Reviewer_QBF7 · 2022-08-04
> > **Acknowledgement of rebuttal**
> >
> > Thank you for your detailed response! You have done a good job at addressing many of my concerns and I am inclined to increase my score.
> >
> > However, I am still a little unsure on the point of significance; but this may be in part because I do not follow the literature for algorithmic developments that focus solely on tabular environments.
> >
> > To be a bit more concrete, two points in your rebuttal that I'd love a little more detail on are the following:
> > 1. `we argue that success in hard tabular environments is likely a necessary (but not sufficient) condition for success in hard non-tabular environments`
> >    * it would be great if you could provide examples (ideally from the literature) that demonstrates this point. it seems relatively intuitive, but i can also imagine cases where it is not the case (e.g. tabular algoriths can't handle continuous state spaces, but function approximation can).
> > 1. `we note that an empirical benchmark may be important even in developing tabular reinforcement learning algorithms with theoretical guarantees`
> >    * in making this point, it would be useful if you could provide references to existing works that focus exclusively on tabular RL algorithms for people like me who are not as familiar with this literature.
> >
> > One final point which would be great to have, but not as critical as the ones I listed above:
> >
> > Ultimately, I feel, the goal of our community's research is to provide algorithms that can be of practical use in the world. So far, there is plenty of evidence that shows that deep networks with RL can provide remarkable real-world success; it would be great if you could provide examples where purely-tabular RL algorithms have had real-world (or at least non-toy) impact.

---

> > > ### Author Response · Authors · 2022-08-05
> > > **Response to acknowledgement of rebuttal from reviewer QBF7 (Part 1/2)**
> > >
> > > Thanks for considering our response!
> > >
> > > We are quite confident in our claim that success in hard tabular environments is likely a necessary (but not sufficient) condition for success in [general] hard non-tabular environments. For example, consider a large grid-world with obstacles, objects, and sparse-rewards, which can be implemented as a MiniGrid environment in Colosseum. Such a grid-world can be very challenging for tabular reinforcement learning. For non-tabular reinforcement learning, if the state representation enables generalization (such as through encoding the grid-world as an image), the task should become easier than the task of the tabular agent, which cannot rely on generalization. For instance, upon observing that an action always takes the agent to the left in many states, a non-tabular agent may predict the outcome of an action in a state that it has never visited. If the state representation does not enable generalization, such as when the state employs a one-hot encoding, the advantage of the non-tabular agent is removed. However, such kind of representation can test whether a non-tabular agent is able to deal with environments where generalization is difficult. Generalization is naturally difficult in many settings, so a generally successful non-tabular reinforcement learning algorithm would have to explore as efficiently as possible even in that situation. Therefore, in both cases, tabular environments can be used to test the limits of non-tabular algorithms to a large extent. However, as we mentioned, only a theory of hardness in non-tabular reinforcement learning would enable completely principled and thorough testing, which is why we believe it is so important. This is why we claim that success in tabular reinforcement learning is not a sufficient condition for success in non-tabular reinforcement learning. In case that was not clear, we never intended to claim that a non-tabular algorithm that solves a specific difficult task necessarily needs to be successful in every hard tabular environment. The claim only refers to non-tabular agents that intend to be successful across a wide range of environments.
> > >
> > > Regarding the importance of developing tabular reinforcement learning algorithms (with or without theoretical guarantees), DQN [1] is arguably the most influential work in the entire deep reinforcement learning literature. In the paper, although Q-learning with function approximation has been explored in many previous works, the authors explicitly acknowledge the Q-learning algorithm, which was developed by Watkins [2] for tabular reinforcement learning. It may be tempting to think that insights from tabular reinforcement learning are no longer relevant, so we would like to mention an even more recent development. Posterior sampling for reinforcement learning, whose theoretical properties in the tabular setting have been studied by Osband et al. [3], promises principled and scalable exploration, and has subsequently led to several extensions to the non-tabular setting, such as Bootstrapped DQN [4], randomized least-squares value iteration [5,6], successor uncertainties [7], and several others.
> > >
> > > The goal of Colosseum is to provide the community with a benchmarking tool able to nurture principled reinforcement algorithms that can later be scaled up to the non-tabular setting.
> > >
> > > Part (1/2)

---

> > > > ### Author Response · Authors · 2022-08-05
> > > > **Response to acknowledgement of rebuttal from reviewer QBF7 (Part 2/2)**
> > > >
> > > > It is less important but still worth mentioning that tabular reinforcement learning algorithms have had an impact in areas where prior knowledge is readily available and can be used to formulate a precise model of the problem. Although we do not consider ourselves experts in applications of tabular reinforcement learning, we can mention the work of Jalalimanesh et al. [8], which employs an agent-based model to simulate vascular tumour growth based on biological evidence with the objective of optimizing dose calculation in radiotherapy, and the work of Govindaiah and Petty [9], which relies on Sarsa(0) to produce increasingly efficient plans for material handling in manufacturing facilities.
> > > >
> > > > Part (2/2)
> > > >
> > > > ----------------------
> > > >
> > > > [1] Volodymyr Mnih, Koray Kavukcuoglu, David Silver, Andrei A. Rusu, Joel Veness, Marc G. Bellemare, Alex Graves et al. "Human-level control through deep reinforcement learning." nature 518, no. 7540 (2015): 529-533.
> > > > [2] Christopher Watkins. "Learning from delayed rewards". King's College (United Kingdom), 1989.
> > > > [3] Ian Osband, Daniel Russo, and Benjamin Van Roy. "(More) efficient reinforcement learning via posterior sampling." Advances in Neural Information Processing Systems 26 (2013).
> > > > [4] Osband, Ian, Charles Blundell, Alexander Pritzel, and Benjamin Van Roy. "Deep exploration via bootstrapped DQN." Advances in neural information processing systems 29 (2016).
> > > > [5] Andrea Zanette, David Brandfonbrener, Emma Brunskill, Matteo Pirotta, and Alessandro Lazaric. "Frequentist regret bounds for randomized least-squares value iteration." In International Conference on Artificial Intelligence and Statistics, pp. 1954-1964. PMLR, 2020.
> > > > [6] Ian Osband, Benjamin Van Roy, and Zheng Wen. "Generalization and exploration via randomized value functions." In International Conference on Machine Learning, pp. 2377-2386. PMLR, 2016.
> > > > [7] David Janz, Jiri Hron, Przemysław Mazur, Katja Hofmann, José Miguel Hernández-Lobato, and Sebastian Tschiatschek. "Successor uncertainties: exploration and uncertainty in temporal difference learning." Advances in Neural Information Processing Systems 32 (2019).
> > > > [8] Ammar Jalalimanesh, Hamidreza Shahabi Haghighi, Abbas Ahmadi, and Madjid Soltani. "Simulation-based optimization of radiotherapy: Agent-based modeling and reinforcement learning." Mathematics and Computers in Simulation 133 (2017): 235-248.
> > > > [9] Swetha Govindaiah, and Mikel D. Petty. "Applying reinforcement learning to plan manufacturing material handling." Discover Artificial Intelligence 1, no. 1 (2021): 1-33.

---

> > > > > ### Comment · Reviewer_QBF7 · 2022-08-05
> > > > > **Tabular envs**
> > > > >
> > > > > Thank you for your responses. These are all very fair points and do address a lot of my reservations.
> > > > >
> > > > > I am leaning to increase my score, but will wait to see what comes of the discussions with the other reviewers.

---

### Official Review · Reviewer_XAUr · 2022-07-11

**Rating:** 7
**Confidence:** 3
**Soundness:** 3 good
**Presentation:** 2 fair
**Contribution:** 4 excellent

**Summary:**

The paper introduces Colosseum, a Python package that allows empirical investigation of MDP hardness along estimation complexity and visitation complexity for tabular reinforcement learning. It surveys various existing harndess measures and argues why many of these do not capture the above mentioned complexities properly. The ones which come closest to capturing these are Environmental value norm for estimation complexity and Diameter for visitation complexity as I understand from the paper. It also implements agents and implements a benchmark for what the authors claim are the four most widely studied tabular reinforcement learning settings - which I believe are: (a) Episodic ergodic. (b) Episodic communicating. (c) Continuous ergodic. (d) Continuous communicating. They also perform experiments examining the hardness measures under various changes to the MPDs (Fig. 1) as also with the agents in the mentioned settings (Tab. 1 and fig. 2).


**Questions:**

>For instance, consider an MDP where a set of lowly rewarding states stands between an agent and a set of highly rewarding states. A regret minimization agent is heavily discouraged from visiting the lowly rewarding states, in contrast to a PAC-RL agent. Therefore, the state visitation complexity should differ across settings.

I do not quite think this is the best argument here because if, let's say, the only way to the high-rewarding states is through the lowly rewarding states, then regret would still be minised by going through them and so the agent would not be discouraged.


>The fact that current measures disregard this distinction is concerning, since they should quantify the difficulty of a specific optimization task.

What is "this distinction" here?

What is "the criterion lower bound of class M up to logarithmic factors"?

In line no. 274: "the probability p_lazy that an MDP stays in the same state instead of executing the action selected by an agent"

How can the MDP remain in the same state if an action was taken? Is there a "no-op" action as well in the action space? I saw only 3 actions in the action space and no no-op.

"which solely allows comparing trends"

why is solely italicised here?

How did the authors choose twelve and five random seeds for the experiments?

The paper mentions both cumulative regret and sample efficiency for exploration as possible performance criteria/metrics but then only experiments with cumulative regret agents are shown. Is there a reason sample efficiency for exploration was not considered for the experiments?



**Limitations:**

Yes.

**Strengths And Weaknesses:**

I believe they motivate the approach well and the benchmark is more comprehensive than existing benhcmark for tabular RL. The quality of the woek also seems high. They provide code and collect it in a single package which should be of great significance to the community, esp. but not just the tabular RL communtiy. I assume the code quality is also good based on a quick walk through the Jupyter notebook: Colosseum_main_tutorial.ipynb. The analysis plots also look good. However, in respect to the clarity, I feel like the paper would be much more suited to the journal format. I feel like many of the questions I ask below are probably clarified in the Appendix (which I only glanced through), however, I feel like some of these rather belong in the main paper.

The motivations for the different MDP families was not described in the main paper

There were some statements such as:
>The ergodic settings seem generally slightly easier than the communicating settings.

But the evidence was not explicitly pointed out. I felt like this in many places and it feels this was done to save space.

Because of such space-saving measures, I feel the paper is better suited to a journal.


In various places the clarity can be improved, e.g.:

>The diameter also increases almost linearly with the number of states. When p_rand is relatively small, an approximately linear relationship can still be observed.

This was a little confusing because sub-figures being referenced are not mentioned. esp. for the 2nd sentence, I can't see the linear relationship when  p is "small" in Fig. 1a because it's not zoomed in enough. What values of p were meant "small"?


>"Sum of the reciprocals of the sub-optimality gaps": This measure of hardness is not particularly apt at capturing estimation complexity, since it focuses solely on optimal policy identification. It also underestimates the increase in hardness induced by an increase in visitation complexity.

I understand the arguments. However, in Fig. 1a-1d, tbh, "Sum of the reciprocals of the sub-optimality gaps" actually seems the closest to the cumulative regret of the tuned near-optimal agent. Could the authors please add some reasoning as to why this measure ends up being seemingly the closet to the cumulative regret of the tuned near-optimal agent even though it is not suited either for visitation complexity or estimation complexity.


>For Q-learning and PSRL (Figs. 2b and 2c), the diameter seems to have a generally smaller influence on the average cumulative regret

I cant quite see this in the figures.

>(efficiently computable) hardness measures

Adding a table with computational complexity of calculating the measures would be highly appreciated.

---

> ### Author Response · Authors · 2022-08-02
> **Response to reviewer XAUr (Part 1/2)**
>
> Thank you very much for the time dedicated to evaluate our work. We are glad that you found our work well motivated and of potentially great significance to the community.
>
> In case this is not clear, we would like to emphasize that Section 2 is not limited to reviewing existing literature. Note that most quantities that we consider hardness measures were originally proposed in the context of providing theoretical guarantees for reinforcement learning algorithms. We have re-interpreted these quantities as hardness measures.
>
> Indeed, we had to make difficult choices to meet the page requirements, and a journal would allow us to include much more detail in the main paper. However, we believe a conference submission is likely to attract more feedback from the community, which is very important for the widespread adoption of the Colosseum benchmark. Although some explanations in the main paper are indeed brief, we hope the appendices present every detail needed to understand and replicate our work. We will consider extending the work on Colosseum and submitting it to a journal in the future.
>
> We also agree that the motivation for choosing the different environment families belongs in the main paper instead of the appendix, and so we will move it to the text. During our selection, we tried to balance between traditional environment families (such as RiverSwim, Taxi, and FrozenLake) and unconventional environment families (such as MiniGid, which is not widely used in tabular reinforcement learning). The Deep Sea family was included since it was recently proposed and developed as an example of a hard exploration problem. The SimpleGrid acts as a simplified version of the MiniGrid-Empty environment.
>
> Our claim that "ergodic settings seem generally slightly easier than the communicating settings" is supported by the last line of Table 1 (the average normalized cumulative regret across ergodic environments is lower than the same quantity for the communicating environments). The claim that "diameter also increases almost linearly with the number of states. When prand is relatively small, an approximately linear relationship can still be observed" does not refer to Fig. 1a, which corresponds to scenario 1, but to Fig 1.d, which corresponds to scenario 4, so the relatively small value of p is 0.1. We will clarify both claims in the paper.
>
> The measures of hardness analysed in Fig. 1 are not originally on a comparable scale. In order to present a meaningful comparison, they have to be normalized. As a consequence, plots can only be used to compare trends (growth rates). Noting this is quite important to interpret the figures, which is why the word "solely" is italicized. We will update the paper to explain this more thoroughly before we present the figures. In the case of the sum of the reciprocals of the sub-optimality gaps, although the values of such measures are close in the plots to the cumulative regret of the tuned near-optimal agent, the trends are not similar. Trends are similar in Fig. 1a and dissimilar in all the other scenarios. In Fig. 1b, although the normalized values of the sum of the reciprocals of the sub-optimality gaps may appear close to the cumulative regret of the tuned near-optimal agent, the growth rate of the diameter more closely reflects the trend of the cumulative regret of the tuned near-optimal agent. In Figures 1c and 1d, the sum of the reciprocals of the sub-optimality gaps presents the same identical trends even though setting the value of prand to $0.1$ significantly impacts the challenge posed by the MDPs.
>
> Regarding the fact that "for Q-learning and PSRL (Figs. 2b and 2c), the diameter seems to have a generally smaller influence on the average cumulative regret", note that it is possible to observe values of the cumulative regret higher than $0.8$ for values of the diameter in the interval $[20, 60]$. This is not the case for UCRL, for which high values of regret only appear after a diameter value of $60$. We will improve the clarity of this statement, which does not make the relativeness to UCRL clear.
>
> As requested, we will add details about the computational complexity of the (efficiently computable) hardness measures.
>
> Part (1/2)

---

> > ### Author Response · Authors · 2022-08-02
> > **Response to reviewer XAUr (Part 2/2)**
> >
> > We agree that our explanation of why a regret minimization agent would generally behave differently from a PAC-RL agent is somewhat unclear. The distinction in behavior encouraged by the different performance criteria is best explained by the fact that the regret minimization agent must be cautious not to incur large regret during learning (for example, by not revisiting lowly rewarding states that are not followed by highly rewarding states), whereas a PAC-RL agent has the flexibility to incur a large regret as long as it ends up with a near-optimal policy.
> >
> > Regarding "the criterion lower bound of class M up to logarithmic factors", for each environment class M, it is possible to obtain a lower bound on the cumulative regret or the sample complexity. The lower bound quantifies the minimum amount of cumulative regret or sample complexity that a generic agent will incur when attempting to solve an environment from the class M. When a known upper bound on the cumulative regret or the sample complexity of a specific agent matches this lower bound of an environment class up to a logarithmic factor, we say that the agent is near-optimal. For example, Jaksch et al. (2010) prove a lower bound of $\Omega(\sqrt{DSAT})$ for the class of continuous communicating MDPs, which implies that the criterion lower bound of the class of communicating MDPs up to logarithmic factors is $\sqrt{DSAT}$. We will clarify this in the paper.
> >
> > Regarding "the probability plazy that an MDP stays in the same state instead of executing the action selected by an agent", we emphasize that the agent does not have a "no-op" action. Instead, the transition kernel of an original MDP is transformed by the introduction of the probability plazy. No matter the action chosen by the agent, there is a probability plazy that the original transition kernel will be ignored and that the transformed MDP will remain in its current state.
> >
> > The number of seeds for the experiments was determined during preliminary experiments where we observed the variability of the results.
> >
> > In practice, it is very computationally expensive to compute the sample efficiency criterion without making approximations that make it hard to interpret and compare, which is why it is currently not measured by Colosseum. However, we are currently developing an approximation to implement in Colosseum.
> >
> > Because other reviewers were concerned that Colosseum could be limited to studying tabular reinforcement algorithms, we would like to note that an up-to-date version of the package already allows experimentation using non-tabular representations of the existing environments. Concretely, it is possible to use vector-based state representations that allow Colosseum to benchmark reinforcement learning algorithms that employ both linear and non-linear function approximation. Currently available vector-based state(-action) representations include one-hot encodings, positional representations (for grid-like environments), and a representation that guarantees the linearity of the optimal action-value function [3]. In hindsight, we believe that the text does not emphasize the potential of Colosseum for benchmarking non-tabular reinforcement learning algorithms enough, so we plan on changing the text substantially to reflect this. We will do so by highlighting this potential in the abstract and in the introduction, by explaining the corresponding functionalities in the section that introduces Colosseum, and by recalling this potential in the conclusion. Importantly, we will also add a Jupyter notebook tutorial that shows how to use non-tabular reinforcement learning algorithms with vector-based representations of Colosseum environments to the supplementary material.
> >
> > We hope that these clarifications and the proposed changes will allow you to consider increasing the rating given to our submission. We aim to provide an updated version of our work that incorporates reviewer feedback as soon as possible.
> >
> > Part (2/2)
> >
> > --------
> >
> > References:
> >
> > [1] Dylan J. Foster, Sham M. Kakade, Jian Qian, and Alexander Rakhlin. "The statistical complexity of interactive decision making." arXiv preprint arXiv:2112.13487 (2021).
> >
> > [2] Thomas Jaksch, Ronald Ortner, and Auer Peter. "Near-optimal regret bounds for reinforcement learning." Advances in neural information processing systems 21 (2008).
> >
> > [3] Ian Osband, Benjamin Van Roy, and Zheng Wen. "Generalization and exploration via randomized value functions." In International Conference on Machine Learning, pp. 2377-2386. PMLR, 2016.

---

### Official Review · Reviewer_XzpS · 2022-07-22

**Rating:** 6
**Confidence:** 4
**Soundness:** 4 excellent
**Presentation:** 4 excellent
**Contribution:** 3 good

**Summary:**

This work provides a unifying review of hardness measures for MDPs that have appeared in previous RL theory bounds in tabular settings. The authors have also developed a benchmark with easily estimable values for these hardness measures to be used for empirical investigation of RL theory. Performance of four standard algorithms are measured in the environments.

**Questions:**

How were environments selected from the literature? Was any attempt made to construct novel MDPs with particular relationships between hardness measures? What limited this process if so?

**Limitations:**

As suggested above, the insight here is mainly unifying, with little specific novel contributions, either in terms of environment or in terms of analysis. New measures of hardness, or previously unseen environments with useful properties would make the paper very strong.

**Strengths And Weaknesses:**

*Originality:*
The work provides a unifying perspective on what has thus far been seen as disparate notions of hardness, with qualitative comparison of the strengths and weaknesses of each. I think that, while no new notion of hardness is investigated here, which would make this a very strong paper, the perspective offered is novel and worthwhile. The development of a standard tabular benchmark for RL theory is again a work of synthesis from previous papers. While valuable for RL theory practitioners, there is less novel insight here, as these environments are well known.

*Quality:*
The paper is clearly well-thought through and well constructed, with unifying insight provided. I think if the environments weren't chosen from the literature, but instead chosen such that the different aspects of hardness described in the paper were easier to control independently with respect to the different policy and environment parameters described in the paper, the benchmark would lead to even more meaningful insight.

*Clarity:*
The paper is easy to follow, and plots and charts are easy to read. The visualisations of the environments give good intuition as to their structure, and the experiments give good characterisation of their hardness.

*Significance:*
The paper is significant, and well-poised to enable future work in unified hardness and empirical evaluation of RL theory results.

---

> ### Author Response · Authors · 2022-08-02
> **Response to reviewer XzpS (Part 1/2)**
>
> Thank you very much for the time dedicated to evaluate our work. We are glad that you found that our perspective is novel and worthwhile. We are also glad that you found our paper well constructed behind a unifying insight and significant for future work.
>
> In case it was not clear, we would like to emphasize that Section 2 is not limited to reviewing existing literature. Note that most quantities that we consider hardness measures were originally proposed in the context of providing theoretical guarantees for reinforcement learning algorithms. We have re-interpreted these quantities as hardness measures.
>
> Regarding the criteria used to select environments from the literature, we tried to balance between traditional environment families (such as RiverSwim, Taxi, and FrozenLake) and unconventional environment families (such as MiniGid, which is not widely used in tabular reinforcement learning). The Deep Sea family was included since it was recently proposed and developed as an example of a hard exploration problem. The SimpleGrid acts as a simplified version of the MiniGrid-Empty environment. We will move the motivation for choosing the different environment families from the appendix to the main text to make this clearer.
>
> We believe that constructing novel MDPs with desired relationships between hardness measures is made straightforward by Colosseum. In fact, this process can be automated to some extent by employing a strategy similarly to how we selected the MDPs that compose the benchmark. We will make note of this in the text.
>
> We also want to emphasize that we expect Colosseum to have an impact beyond the reinforcement learning theory community. The principled selection of environments for the Colosseum benchmark can play a fundamental role in the development and analysis of new tabular and non-tabular reinforcement learning algorithms. Although the text has mostly focused on using Colosseum for studying tabular reinforcement learning algorithms, an up-to-date version of the package already allows experimentation using non-tabular representations of the existing environments. Concretely, it is possible to use vector-based state representations that allow Colosseum to benchmark reinforcement learning algorithms that employ both linear and non-linear function approximation. Currently available vector-based state(-action) representations include one-hot encodings, positional representations (for grid-like environments), and a representation that guarantees the linearity of the optimal action-value function [2].
>
> In the early stages of developing a new reinforcement learning algorithm, it is often very useful to test ideas in simple environments, which allows fast iteration. Having a thorough and efficient benchmark equipped with useful analysis tools allows researchers to focus most of their time on the algorithms themselves. Although there is no well-developed theory of hardness for non-tabular reinforcement learning, we argue that success in hard tabular environments is likely a necessary (but not sufficient) condition for success in hard non-tabular environments. The fact that we do not include strictly non-tabular environments stems from our goal of providing a principled benchmark, which is currently not possible for non-tabular reinforcement learning due to hardness being a relatively recent (and active) focus of research in that setting [1]. Similarly, we do not include results of a comparison between state-of-the-art non-tabular reinforcement learning algorithms in the Colosseum benchmark because we believe such comparison would be premature and likely unfair (by not considering all important aspects of hardness in the non-tabular setting). Extending the benchmark to environments that require function approximation in a principled way is one of the main goals of our future work.
>
> (Part 1/2)

---

> > ### Author Response · Authors · 2022-08-02
> > **Response to reviewer XzpS (Part 2/2)**
> >
> > Finally, we note that an empirical benchmark may be important even in developing tabular reinforcement learning algorithms with theoretical guarantees. Some choices in those algorithms may have to be made without guidance from the theory, in which case they may be informed by practice. Ultimately, this is important because principled algorithms often inspire scalable counterparts.
> >
> > In hindsight, we believe that the text does not emphasize the potential of Colosseum for benchmarking non-tabular reinforcement learning algorithms enough, so we plan on changing the text substantially to reflect this. We will do so by highlighting this potential in the abstract and in the introduction, by explaining the corresponding functionalities in the section that introduces Colosseum, and by recalling this potential in the conclusion. Importantly, we will also add a Jupyter notebook tutorial that shows how to use non-tabular reinforcement learning algorithms with vector-based representations of Colosseum environments to the supplementary material.
> >
> > We hope that these clarifications and the proposed changes will allow you to consider increasing the rating given to our submission. We aim to provide an updated version of our work that incorporates reviewer feedback as soon as possible.
> >
> > (Part 2/2)
> >
> > ---------
> >
> > References:
> >
> > [1] Dylan J. Foster, Sham M. Kakade, Jian Qian, and Alexander Rakhlin. "The statistical complexity of interactive decision making." arXiv preprint arXiv:2112.13487 (2021).
> >
> > [2] Ian Osband, Benjamin Van Roy, and Zheng Wen. "Generalization and exploration via randomized value functions." In International Conference on Machine Learning, pp. 2377-2386. PMLR, 2016.

---

### Author Response · Authors · 2022-08-02
**Update: new version of the Colosseum package**

We have updated the Colosseum GitHub repository, which can be anonymously accessed through Anonymous GitHub using [this link](https://anonymous.4open.science/r/Colosseum-B725/README.md), with the following improvements.
#### *Clarity*
- We have refactored part of the code to improve clarity and ease the implementation of novel reinforcement learning agents and environments.
- We have reached 100% documentation coverage.
#### *Efficiency*
- We have improved the computational efficiency of the dynamic programming algorithms by optimizing the sparse matrices product operations and of the Markov chain-related computations by implementing sparse matrices-based algorithms to compute stationary distributions.
#### *New features*
- We have added automatic computations of state representations for environments implemented in Colosseum. Note that this means that these representations will be already available for every new environment that will be implemented in Colosseum in the future. We make note of the available representations below.
- We have implemented an automatic hyperparameters selection procedure when running the benchmark to ensure a fair comparison between reinforcement learning agents with respect to the hyperparameter optimization.
#### *Tutorials*
- We have added a tutorial on the non-tabular representation available for the Colosseum environments.
- We have added a tutorial on how to implement new reinforcement learning agents in Colosseum.

The available non-tabular state representations are:
- A feature vector with the same dimensionality as the state space that contains a one-hot encoding.
- A feature vector $\phi(s)$ that enables linear representation of the optimal value function. In other words, there is a $\theta$ such that $\hspace{1mm}V^*(s) = \theta^T\phi(s)$.
- A feature vector $\phi(s)$ that enables linear representation of the value function of the randomly acting policy policy. In other words, there is a $\theta$ such that $\hspace{1mm}V^\pi(s) = \theta^T\phi(s)$, where $\pi$ is the randomly acting policy.
- A feature vector that contains uniquely identifying information about the state (for instance, coordinates for the DeepSea environment).
- A feature matrix that encodes a visual representation of the environment.
- A 3d matrix that one-hot encodes a visual representation of the environment.

We will update the supplementary material with the new version of the code when we finish updating the paper according to your feedback.

---

### Meta-Review · Area_Chair_U1aA · 2022-08-26

**Recommendation:** Accept
**Confidence:** Less certain

**Metareview:**

The reviewers' opinions are quite consistent towards a weak accept.
I'm not confident with the big title "Hardness in Markov Decision Processes: Theory and Practice". This paper is more like a survey + benchmark review instead of a research article. Neither the theory part or the practice part is novel enough as a research article. It's a bit thin as a survey paper.
I personally tend to weak reject but I respect the reviewers' weak accept.

**Award:**

No

---

### Decision · Program_Chairs · 2022-09-14

Accept